# Dynamic actuation enhances transport and extends therapeutic lifespan in an implantable drug delivery platform

William Whyte [1,9], Debkalpa Goswami [1,9], Sophie X. Wang [1,2,9], Yiling Fan[3], Niamh A. Ward [1,4], Ruth E. Levey[5], Rachel Beatty [5], Scott T. Robinson[5,6], Declan Sheppard[7], Raymond O'Connor[5], David S. Monahan [1,5], Lesley Trask [4], Keegan L. Mendez[8], Claudia E. Varela[8], Markus A. Horvath[8], Robert Wylie[5], Joanne O'Dwyer[1,4], Daniel A. Domingo-Lopez [5], Arielle S. Rothman [1], Garry P. Duffy[5,6], Eimear B. Dolan [4] ✉ & Ellen T. Roche [1,3,8] ✉

Fibrous capsule (FC) formation, secondary to the foreign body response (FBR), impedes molecular transport and is detrimental to the long-term efficacy of implantable drug delivery devices, especially when tunable, temporal control is necessary. We report the development of an implantable mechanotherapeutic drug delivery platform to mitigate and overcome this host immune response using two distinct, yet synergistic soft robotic strategies. Firstly, daily intermittent actuation (cycling at 1 Hz for 5 minutes every 12 hours) preserves long-term, rapid delivery of a model drug (insulin) over 8 weeks of implantation, by mediating local immunomodulation of the cellular FBR and inducing multiphasic temporal FC changes. Secondly, actuation-mediated rapid release of therapy can enhance mass transport and therapeutic effect with tunable, temporal control. In a step towards clinical translation, we utilise a minimally invasive percutaneous approach to implant a scaled-up device in a human cadaveric model. Our soft actuatable platform has potential clinical utility for a variety of indications where transport is affected by fibrosis, such as the management of type 1 diabetes.

Our immune system has evolved to acquire a robust defence mechanism against foreign body invasion. In the presence of a 'foreign object', neutrophil infiltration initiates a cascade of inflammatory and wound healing processes, which precipitates the formation of a dense, encapsulating fibrous capsule (FC)[1,2]. The foreign body response (FBR) minimises exposure to potential toxins and is often advantageous; for

example, soldiers with bullet wounds rarely develop clinical symptoms of lead poisoning[3,4].

This protective response, however, is detrimental to the long-term durability of implantable biomedical devices such as breast implants[5,6], heart valves[7], and pacemakers[8]. These devices have transformed modern patient care, but the immune infiltration and fibrotic

[1]Institute for Medical Engineering and Science, Massachusetts Institute of Technology, Cambridge, MA, USA. [2]Department of Surgery, Beth Israel Deaconess Medical Center, Boston, MA, USA. [3]Department of Mechanical Engineering, Massachusetts Institute of Technology, Cambridge, MA, USA. [4]Department of Biomedical Engineering, National University of Ireland Galway, Galway, Ireland. [5]Anatomy and Regenerative Medicine Institute (REMEDI), National University of Ireland Galway, Galway, Ireland. [6]Advanced Materials and BioEngineering Research Centre (AMBER), Trinity College Dublin, Dublin, Ireland. [7]Department of Radiology, University Hospital, Galway, Ireland. [8]Harvard-MIT Program in Health Sciences and Technology, Cambridge, MA, USA. [9]These authors contributed equally: William Whyte, Debkalpa Goswami, Sophie X. Wang. ✉e-mail: eimear.dolan@nuigalway.ie; etr@mit.edu

response can negate device function over time, necessitating painful revision or replacement surgery. This fibrous barrier is particularly deleterious for biosensors, such as continuous glucose monitors, and controlled drug release devices, such as insulin pumps, which rely on interactive communication with their local tissue environment[9–11]. In such cases, the formation of a hypopermeable capsule can impede transport of molecules, both to[12] and from[13,14] the implant, and lead to therapy failure.

One pertinent example is the management of type 1 diabetes, a chronic disease affecting 18 million people worldwide, with an annual economic burden of greater than $90 billion USD (Study: Disease-Modifying Therapies Needed to Offset Costs of Type 1 Diabetes - Juvenile Diabetes Research Foundation). Successful implementation and clinical adoption of an artificial pancreas combining continuous glucose monitoring with the rapid, responsive, release of insulin (or glucagon) would vastly improve outcomes and quality of life for this patient population. The development of a fully automated closed loop insulin delivery system would reduce user burden, remove the need for multiple daily injections, and increase time spent in the optimal blood glucose range, which is imperative for the prevention of long-term diabetic complications. Unfortunately, current efforts at developing such a device have been hindered by the dynamic and unpredictable FBR, leading to glucose sensing inaccuracy, inhibition of insulin release, and gradual loss of functionality in the weeks to months following implantation[4,15–17]. Looking towards the future, living implants containing stem cell-derived pancreatic β-cells represent a potential cure for diabetes. However, the attenuation of oxygen and molecular transport due to the FC barrier still constitutes a major hurdle to successful clinical translation of these implants[9,10,18,19]. It is evident that a method to (i) mitigate the FBR or (ii) improve transport across the FC could transform the management of this pervasive disease. Furthermore, such a method could have broader implications for a range of diseases and device-based treatments affected by the FBR.

Conventional strategies to mitigate the FBR have focused on changing the attributes of the implant material itself, such as its size, shape, topography and surface coating[20–28], or involved the concomitant delivery of FBR modifying drugs, such as steroidal anti-inflammatory, anti-fibrotic, and anti-proliferative agents[29]. While these strategies have shown promise, they have not succeeded in completely disarming the FBR and possess several limitations. Firstly, materials are generally pre-designed to target only one component or timepoint of an immune response that is multifaceted and temporally dynamic. Secondly, the use of FBR modifying therapeutics presents safety concerns due to untargeted adverse effects and local toxicity[30–32]. Sustained, systemic delivery of therapeutics such as non-steroidal anti-inflammatories is associated with a range of toxicities in the liver, kidneys, heart, and gastrointestinal tract[32]. Local targeted delivery can reduce off-target effects but may still adversely affect the underlying tissue or interfere with the mechanism of action of the implantable device. For example, local delivery of dexamethasone can mitigate the FBR, but not without suppressing underlying tissue regeneration[33]. Furthermore, the effect of long-term immunosuppression on the behaviour and secretome of cell-based therapeutics is unclear. Finally, a local depot of drug is finite, often lasting 1–2 months, while many needs are lifelong[32]. Thus, depending on drug pharmacology and clinical context, the immune and fibrotic response may rebound once the residual effects of drug inhibition dissipate. A long-term, drug-free method that can modulate and adapt to the FBR over time would, therefore, be highly desirable to address these limitations.

Dynamically altering the local biomechanical environment at the implant site is one such promising, yet underexplored, drug-free approach[34]. Cells in our body are exquisitely sensitive to their mechanical environment, with loading playing a pivotal role in cell functions such as differentiation[35], proliferation[36], and migration[37].

Historically, studies have observed biomechanical stress as a pro-fibrotic or regenerative stimulus[1], demonstrating that application of stretch[38–40], fluid flow[41,42], or compression[43] to cells can lead to increased deposition of collagenous matrix. Accordingly, many anti-FBR strategies have focused on minimising the mechanical mismatch, interfacial stress, and movement between the implant and local tissue. Our research seeks to challenge this status quo and reveals the potential for a dynamic mechanotherapeutic that uses low-magnitude, atraumatic, tissue strain and convective flow as a defence mechanism against the invading cellular FBR. Interestingly, some studies have observed that small magnitude, dynamic loading has anti-inflammatory and pro-regenerative effects. Previous work applying dynamic loading to tissue has used daily mechanical, pneumatic, or magnetic stimuli, either internally or externally, to apply cyclic loads, inducing strains ranging from 4 to 50%, with each cycle lasting between 1 s and 10 min[44–50]. These studies have demonstrated beneficial effects in terms of vascularisation[44,45,50], functional tissue regeneration[46,49], and anti-inflammatory gene expression[47]. These prior works on mechanical loading have indicated the presence of a therapeutic threshold, beyond which tissue damage and inflammation occurs[47,49].

Preceding work from our group demonstrated a fibrosis attenuating effect elicited by a dynamic soft reservoir following acute implantation[34]. Here, we build on this work, and introduce a soft transport augmenting reservoir (STAR) which can persistently mitigate the dynamic FBR and maintain long-term, rapid, molecular communication with its tissue environment using two distinct and synergistic soft robotic actuation strategies: intermittent actuation (IA) and actuation-mediated rapid release (RR). Importantly, we shed light on the mechanistic underpinnings of IA and reveal an immunomodulatory effect in the acute phase of implantation, with a significant reduction in neutrophil infiltration at the pericapsular site, followed by multiphasic temporal capsular changes with chronic implantation. Lastly, in a step towards clinical translation, we demonstrate minimally invasive percutaneous delivery of a human-scale STAR device.

## Results

### Design of a soft transport augmenting reservoir (STAR)

Our lab previously demonstrated the fibrotic attenuating potential of a dynamic device during the initial stages of the FBR (2 weeks)[34]. Based on this foundational work, we propose that application of intermittent, cyclical, low amplitude actuation can act as an oscillating shield against the invading, multiphasic FBR, induce local immunomodulatory effects, and create a favourable environment for the rapid long-term transport of macromolecular drug therapy (Fig. 1a).

To test this hypothesis, we first designed a reservoir suitable for long-term tissue implantation and the precise repeatable delivery of both drug and actuation therapy. Figure 1b shows the multi-layered composition of STAR, with a low-profile design that minimises the presence of sharp angles or edges which may exacerbate the FBR[51,52]. A therapeutic chamber lies in direct contact with underlying tissue and is separated by a membrane with an array of 10 μm pores (Supplementary Fig 1). A connected indwelling catheter line allows for delivery of drug therapy with temporal control (Fig. 1b, c). Superimposed on the therapeutic chamber is an actuation chamber that can be pressurised to elicit controlled oscillation of the porous, tissue-contacting membrane (Fig. 1c, d; Supplementary Movie 1). Imbalances between the mechanical properties of the implant and the surrounding tissue are also known to exacerbate FC formation, with stiffer implants eliciting a heightened immune response[52]. For this reason, STAR was manufactured from thermoplastic polyurethane (TPU) with an elastic modulus of ~15 MPa (Fig. 1d), similar to that of extracellular matrix[53,54]. STAR can easily be scaled between animal models using 3D printed moulds and a simple thermoforming/heat-sealing process (Supplementary Fig 2).

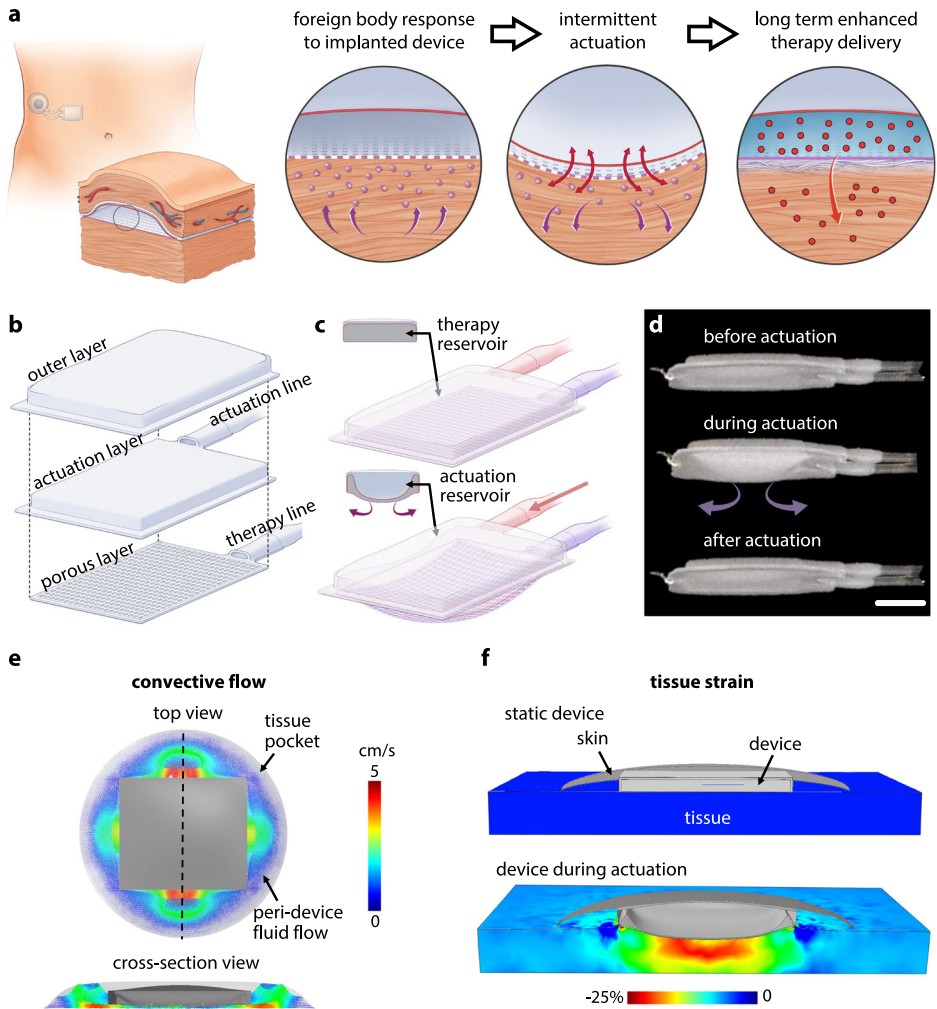

**Fig. 1 | Design of soft transport augmenting reservoir (STAR). a** Proposed mechanism of STAR: intermittent actuation attenuates the foreign body response, creating a favourable environment for the rapid long-term transport of macromolecular drug therapy. **b** Exploded view showing the different layers comprising STAR. **c** Deflection of the actuation and porous layers during actuation. **d** A prototype of STAR showing the deflection of the porous layer during an actuation cycle. Scale bar is 5 mm. **e** FE model showing peri-implant fluid velocity of convective flow during actuation. **f** FE model estimating maximum principal tissue strain induced by actuation.

As part of device design and optimisation, we performed finite element (FE) simulations to understand the biomechanical changes mediated by actuation, particularly the relationships between membrane deflection, convective flow, and tissue strain (Fig. 1e, f; Supplementary Fig 3). Based on recently reported results[49], we designed our soft robotic actuation strategy to induce tissue strain that would fall within the atraumatic range (<40%), and hypothesise that this regimen would mitigate the FBR by creating convective flow disruptive to the cellular immune response.

## Insulin transport test (ITT): a longitudinal, in vivo method to study the effect of the FBR on therapy transport

Following STAR design and manufacture, we next developed a method to longitudinally monitor the detrimental and progressive effect of the FBR on therapy transport (Fig. 2a). Insulin was chosen as our model macromolecular drug to allow for a real-time, dose-dependent measurement of functional response as insulin crosses the FC and enters the bloodstream.

First, we implanted static STAR devices (without IA) on the subcutaneous dorsal aspect of C57BL/6 mice (Supplementary Fig 4). Next, we injected short-acting human insulin into the device and monitored diffusion-based release across the FC into the systemic circulation via serial blood glucose measurements at day 3 (baseline, BL), 2 weeks, and 3 weeks post implantation (Fig. 2b).

The functional efficacy of an equivalent dose of insulin decreased with implantation time and with progression of the FBR, as indicated by the maximum blood glucose (BG) drop (Fig. 2c) and the area under the BG curve (AUC; Supplementary Fig 5a, b). To corroborate these results, we analysed FC thickness longitudinally using 2D micro-computed tomography (μCT; Fig. 2d) and related it to these functional results. As expected, thickness of the capsule increased with time (Fig. 2e). Importantly, we observed an inverse linear relationship ($r = -0.929$) between FC thickness and insulin efficacy metrics (Fig. 2f, Supplementary Fig 5c).

In a final validation step, we examined the effect of FC thickness on therapy release using a multiphysics computational diffusion model. Our simulations corroborate our experimental results, also indicating that increasing FC thickness has a pronounced effect on drug transport (Fig. 2g), introducing a time lag for the desired therapeutic concentration to cross the capsule and elicit a functional effect (Fig. 2h).

In summary, this data demonstrate the development and validation of a pre-clinical model that can detect real-time changes in FC formation via its effect on macromolecular transport and track these changes over time.

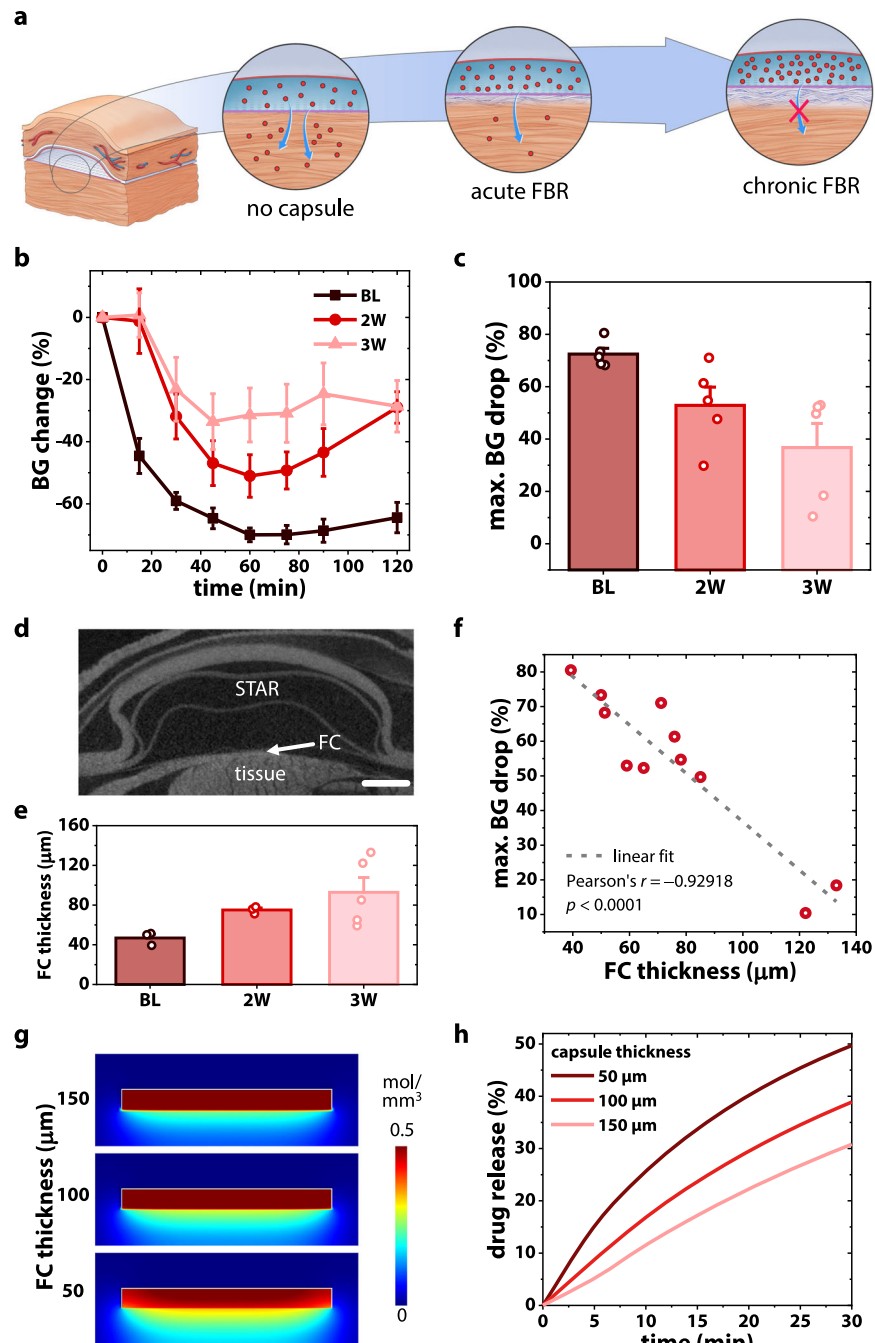

**Fig. 2 | Development of a pre-clinical model to monitor the effect of FBR on therapy transport longitudinally. a** Schematic demonstrating detrimental effect of fibrous capsule (FC) formation on therapy delivery with time. **b** Blood glucose (BG) response to human insulin delivered via STAR, measured over 120 min at baseline (BL, day 3), week 2 and week 3. $n = 5$ mice at each time point. **c** Temporal evolution of the maximum BG % drop (denoting functional effect), calculated from **b**. **d** Representative 2D μCT slice of STAR with fibrous encapsulation. Scale bar is 1 mm. **e** Average FC thickness encapsulating STAR at baseline (day 3), week 2 and week 3 following implantation. $n = 3$ mice at baseline and 2 weeks, 5 mice at 3 weeks. Data are means ± standard error of mean. **f** Relationship between FC thickness and maximum effect of insulin measured by reduction in blood glucose level. **g** COMSOL Multiphysics simulations showing spatial drug diffusion through FCs of varying thicknesses. **h** Temporal evolution of drug release percentage for varying FC thicknesses.

## Dynamic intermittent actuation (IA) extends therapeutic lifespan of STAR

Following device and in vivo model development, we designed an 8-week longitudinal preclinical study to test the ability of STAR to modulate the FBR and improve macromolecular delivery across the formed FC.

We implanted STAR devices (without drug) on the dorsal subcutaneous aspect of three groups of mice (Fig. 3a). In two experimental groups, we performed STAR-enabled IA with cyclic pressure input of

2 psi at 1 Hz for 5 min every 12 h using a custom-made pneumatic control system (Supplementary Fig 6). One group (8W IA) was intermittently actuated for the total study duration of 8 weeks, while the second group (3W IA) received 3 weeks of IA followed by no actuation for the remainder of the study. A third group that did not receive IA served as the control. We then injected short-acting human insulin (2 IU/kg) into the device at various time points post-implantation: 2, 3, 4, 5 and 8 weeks as well as day 3, which served as a BL. We monitored passive, diffusion-based transport across the formed FC and into the

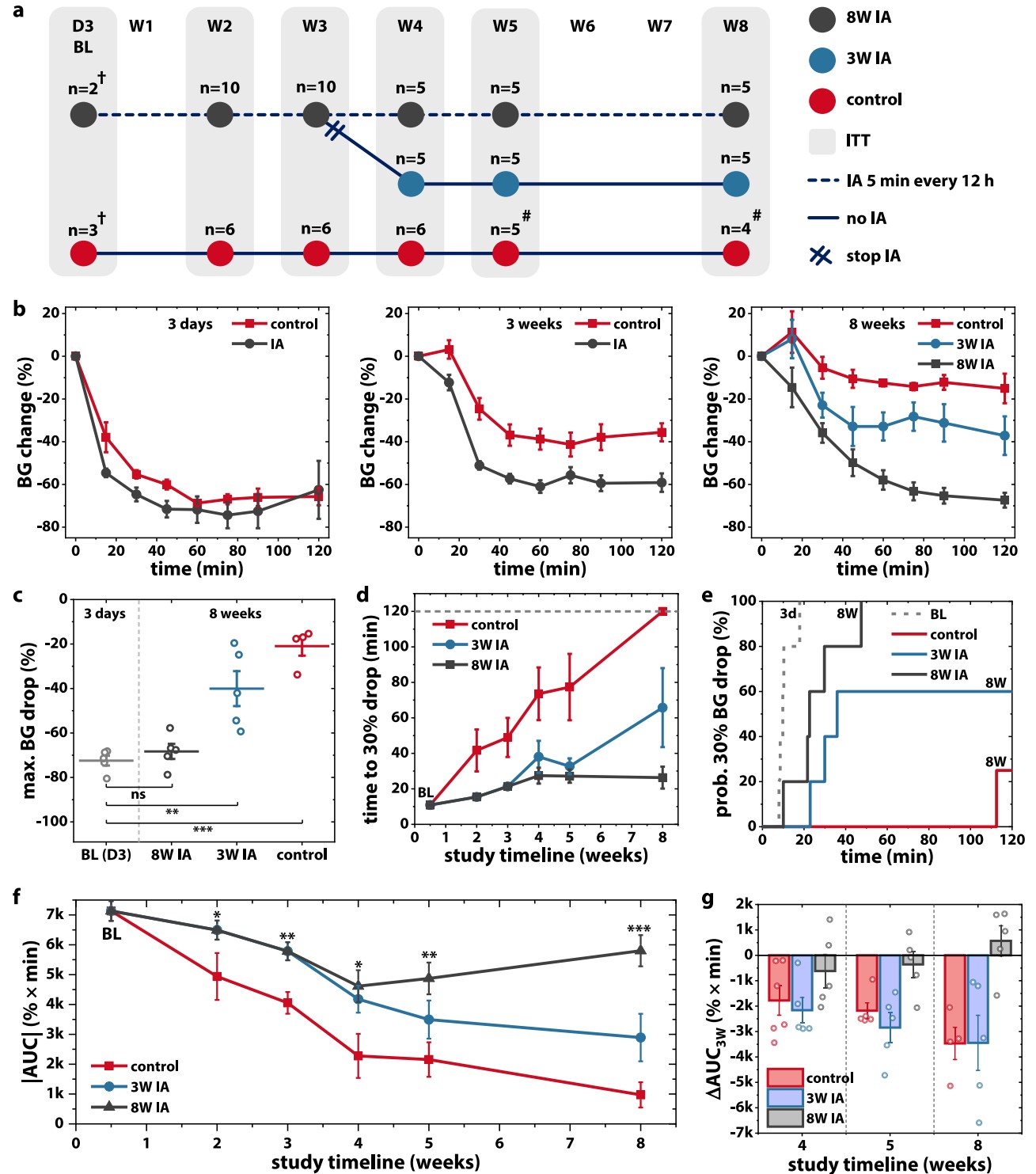

**Fig. 3 | Intermittent actuation (IA) improves long-term macromolecule delivery. a** Preclinical study timeline used to evaluate the effect of IA on insulin transport through a fibrous capsule. **b** Blood glucose (BG) response to human insulin, measured over 120 min at day 3 (baseline), week 3 and week 8. **c** Maximum BG% change at the 8-week timepoint. **d** Time to achieve a 30% drop in BG level, measured longitudinally. **e** Cumulative incidence curves demonstrating probability of achieving a 30% BG drop over 120 min for all groups at both baseline (3 days) and 8-week timepoints. **f** Area under BG % curve (AUC) denoting overall functional effect mapped over study duration. Statistical comparisons w.r.t. control group. **g** Change in AUC from 3-week timepoint. Data are means ± standard error of mean; *$p < 0.05$, **$p < 0.01$, ***$p < 0.001$. See Supplementary Note 1 for detailed statistical analyses. † Baseline study was performed post-hoc with separate mice. # Control mice removed from study at intermediate time points due to self-inflicted device damage with subsequent decrease in $n$.

bloodstream via serial blood glucose measurements at these timepoints.

At baseline, insulin administration produced a similar drop in blood glucose in both IA and control groups (Fig. 3b, Supplementary Note 1). Over 8 weeks, the blood glucose curves separate, with decreased insulin responsiveness in the control and 3W IA groups, compared to the 8W IA group. Impressively, the 8W IA group maintained its rapid drop in blood glucose over the entire study duration

(Fig. 3b). Despite chronic implantation and FC development, there was no statistical difference in the maximum blood glucose drop at 3 days (BL; 72.5 ± 2.2%) and 8 weeks (68.3 ± 3.4%) post-implantation in the 8W IA group (Fig. 3c, Supplementary Note 1). In contrast, the control group achieved only a 20.9 ± 4.3% maximum drop in blood glucose at the 8-week timepoint, reflecting nearly complete loss of drug delivery functionality due to implant isolation by the FC. The 3W IA group had a similar loss of function, with a maximum blood glucose drop that was not significantly better than control at the end of the study (see Supplementary Note 1 for details).

The mean time to achieve a physiologically relevant response (30% drop in blood glucose) was preserved at less than 30 min in the 8W IA group over the entire study duration (Fig. 3d, Supplementary Note 1), with all mice achieving this 30% drop within 48 min at 8 weeks (Fig. 3e). In contrast, the mean time to effect for the control and 3W IA groups progressively increased with implantation time. The mean time to therapeutic effect increased to >65 min in the 3W IA group (with 2 out of 5 mice not responding) by week 8, and was not detectable within the 120 min experimental timeframe in the control group (Fig. 3d, e, Supplementary Note 1). Notably, 8W IA devices were able to achieve a therapeutic blood glucose drop twice as fast as control devices at 4 weeks (mean time to 30% drop: 27.43 ± 4.48 min vs 73.55 ± 14.85 min) and four times as fast at 8 weeks (26.33 ± 6.16 min vs >120 min).

When time and magnitude were integrated by calculating AUC (Supplementary Fig 5a), 8W IA produced a robust treatment effect with significant benefits in drug delivery at all time points in comparison to the control group (Fig. 3f, Supplementary Note 1). Stopping actuation at 3 weeks in the 3W IA group led to a worsening of insulin response, with an AUC progression that paralleled the control group (Fig. 3g). Though there appears to be a trend towards better functional effect in the 3W IA compared to control at the later time points, this was not statistically significant, which implies that continued mechanical dosing will be needed for robust long-term beneficial effects on transport (Fig. 3f, g). Overall, this data suggests that IA can mitigate the FBR and extend the therapeutic lifespan of implantable drug delivery devices.

## Multiphasic temporal effects of intermittent actuation

Following the completion of our pre-clinical study, we next set out to analyse differences in FC composition at evolving timepoints to better understand the multiphasic cellular changes and key drivers of enhanced drug transport caused by IA (Fig. 4a).

First, we investigated the initial, acute phase of the inflammatory FBR. Using immunofluorescent Ly-6G + staining, we examined the pericapsular region for the presence of neutrophils, the first responders of the immune defence. We found that IA significantly reduces the presence of neutrophils at day 5 in comparison to the control (Fig. 4b, c). This result indicates that the application of IA can mediate a localised immunomodulatory effect.

We next assessed the activation of matrix-producing cells into a myofibroblast phenotype, a key contractile cell in fibrosis progression. The IA group exhibited a significant reduction in αSMA expression when compared to the control at 2 weeks (Fig. 4d, e). Despite observing differences in individual cell populations (neutrophils, myofibroblasts), we did not detect differences in overall cell number at equivalent timepoints (Fig. 4f, g).

Next, we investigated the macroscale capsular changes responsible for improved therapy transport. We examined evolving capsule thickness with longer periods of STAR implantation. IA mitigated capsule growth in the first 2 weeks following implantation, with a significant reduction in thickness observed w.r.t. the control at 2 weeks (Fig. 4h, i). This result aligns with the enhanced blood glucose

responsiveness of the IA group at early timepoints (Fig. 3) and suggests that FC thickness is an important contributor to initial improvements in macromolecular transport.

By 8 weeks, however, capsule thickness had equalised between the IA and control groups (Fig. 4i). This result suggests that additional mechanisms are responsible for the sustained improvement in functional response to insulin in the IA group at later timepoints (Fig. 3). To investigate this further, we examined capsule vascularity, density, and the maturity of the collagen fibres. However, we did not observe differences that would account for improvements in macromolecular transport and functional effect (Supplementary Fig 7, Supplementary Note 1). By optical coherence analysis of polarised light microscopy images, we found that collagen fibres exhibited higher alignment in the IA group compared to control at week 8 (Fig. 4j, k). IA appeared to increase collagen alignment over time from 2 weeks to 8 weeks, whereas there was no temporal change in alignment observed in the control group (Fig. 4k). We posit that the lower degree of alignment, and therefore greater degree of fibre entanglement, in the control group creates steric hindrance which potentially slows or immobilises transport of macromolecules through the collagenous matrix[55,56].

Finally, we examined the ability of IA to protect against cellular invasion and blockage of the porous membrane of STAR. Scanning electron microscopy demonstrated clear differences in cellular infiltration between the control and IA group at the 8-week timepoint (Fig. 4l). This effect could be attributed to convective flow generated by STAR upon actuation (Supplementary Fig 3g). These capsular analyses reveal the pleiotropic role of IA in modulating the FBR and highlight cellular and structural changes that lead to improved transport of macromolecular therapy.

## On-demand, actuation-mediated rapid release of drug using STAR

In addition to intermittent immunomodulatory actuation, we demonstrate another soft robotic actuation-based mechanism of augmenting drug transport. Actuation-mediated RR of a drug-loaded STAR device consists of a few (~1-5) cycles of on-demand actuation at the same magnitude as IA (2 psi, 1 Hz). This strategy can accelerate mass transport of drug from the device reservoir (Fig. 5a, Supplementary Movie 2) into surrounding tissue (Fig. 5b, Supplementary Movie 3). Using this on-demand, convective flow-based approach, we investigated if RR could overcome a diffusion-limiting FC barrier by inducing higher concentration and pressure gradients (Fig. 5c, d).

First, we developed a multiphysics computational model comparing passive diffusion-based transport to RR (Fig. 5d–f). RR enhances drug transport across the capsule and thus higher concentrations can reach the therapeutic target in a temporally controlled manner (Fig. 5d, e). Furthermore, multiple actuation cycles can increase transcapsular transport in comparison to a single cycle, and thus dosing can be adapted to the specific clinical scenario (Fig. 5f). Péclet number (Pe) calculations[57] estimate Pe = 2.35 for passive diffusion and Pe = 70.18 for actuation-mediated RR, suggesting that for a given dose of drug, the time required for passive drug delivery through a diffusion dominated process far exceeds that of actuation-mediated drug delivery, which is convection dominated.

To substantiate these simulations, we next investigated the utility of RR in vivo, following long-term implantation and development of a FC. We implanted two STAR devices in a Sprague Dawley rat model to evaluate the spatial distribution of drug with and without RR. On day 24 following implantation, we monitored the distribution area of a fluorescent small molecule drug analogue (Genhance 750) using an in vivo imaging system (IVIS) (Fig. 5g). While passive diffusion of Genhance was slow, RR led to a sharp increase (~7 fold) in drug distribution, despite presence of a FC (Fig. 5h).

In a final example, we demonstrated enhanced mass transport and downstream functional effect using RR in our ITT model at 2 weeks

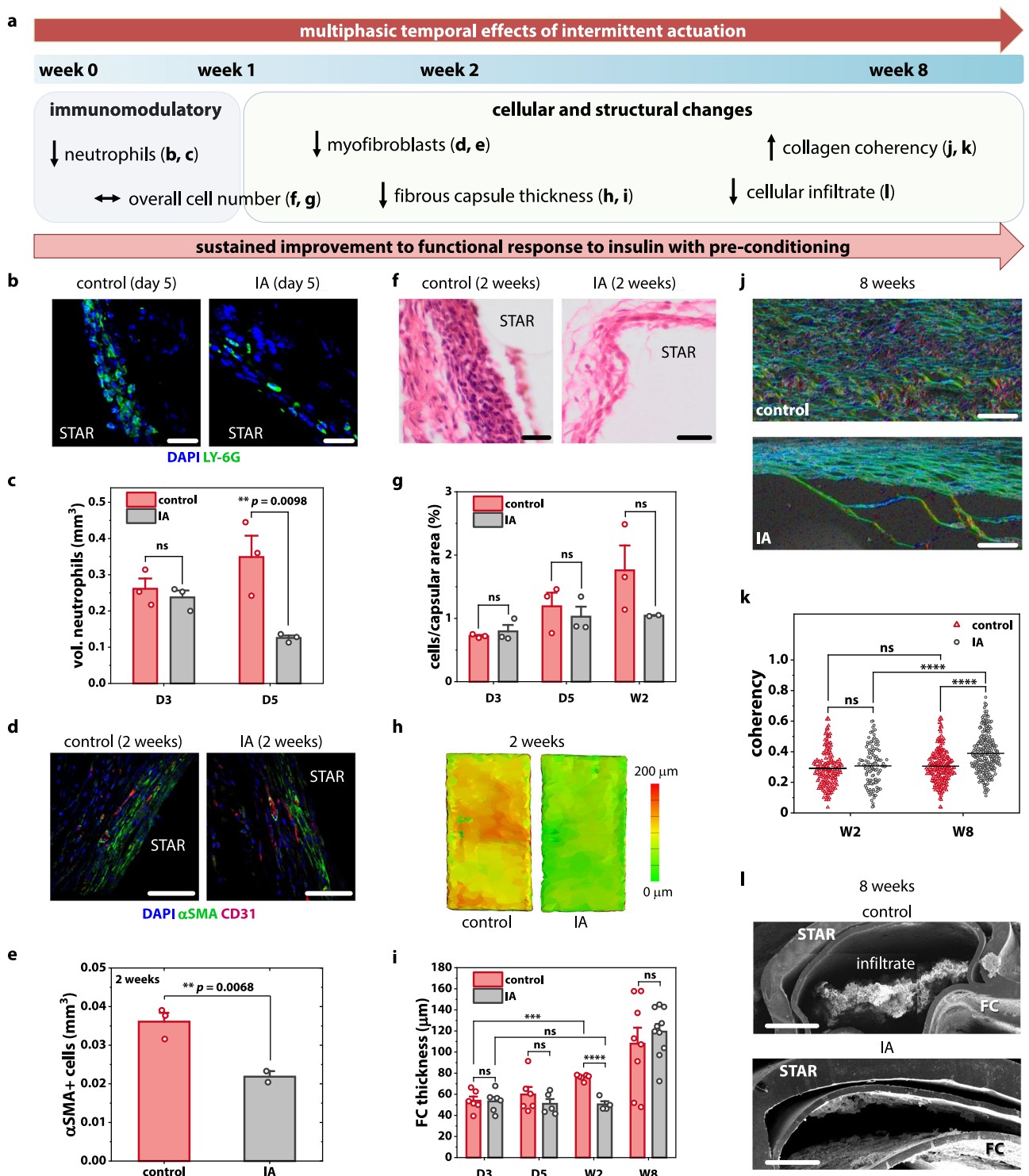

**Fig. 4 | Multiphasic temporal effects of intermittent actuation (IA). a** Timeline of multiphasic cellular and fibrous capsule (FC) changes induced by IA.
**b** Representative fluorescent images of the FC stained with Ly-6G + (green) and DAPI (blue). Scale bars are 20 μm. **c** Quantification of neutrophils present within FC +/− IA at day 3 and 5. **d** Representative fluorescent images of the FC stained with α-SMA (green) and CD31 (red). Scale bars are 50 μm. **e** Quantification of myofibroblasts present within FC+/− IA at 2 weeks. **f** Representative histologic images of the FC stained with haematoxylin and eosin. Scale bars are 20 μm. **g** Quantification of total cells/capsular area +/− IA at day 3, day 5, and 2 weeks. **h** Representative topographical reconstructions of μCT images showing the differences in FC thickness+/− IA at 2 weeks. **i** Average FC thickness of the control and actuated groups at day 3, day 5, 2 weeks, and 8 weeks with two measurements taken per animal. **j** Representative polarised light microscopy images of the FC obtained after picrosirius red staining at 8 weeks. Scale bars are 100 μm. **k**, Quantification of the FC collagen fibre orientation by optical coherency with 60 ROIs per animal.
**l** Representative SEM images demonstrating reduced cellular invasion with actuation at the 8-week timepoint. Scale bars are 500 μm. $n = 2–6$ animals per group; data are means ± standard error of mean; $*p < 0.05$, $**p < 0.01$, $***p < 0.001$, $****p < 0.0001$. See Supplementary Note 1 for detailed statistical analyses.

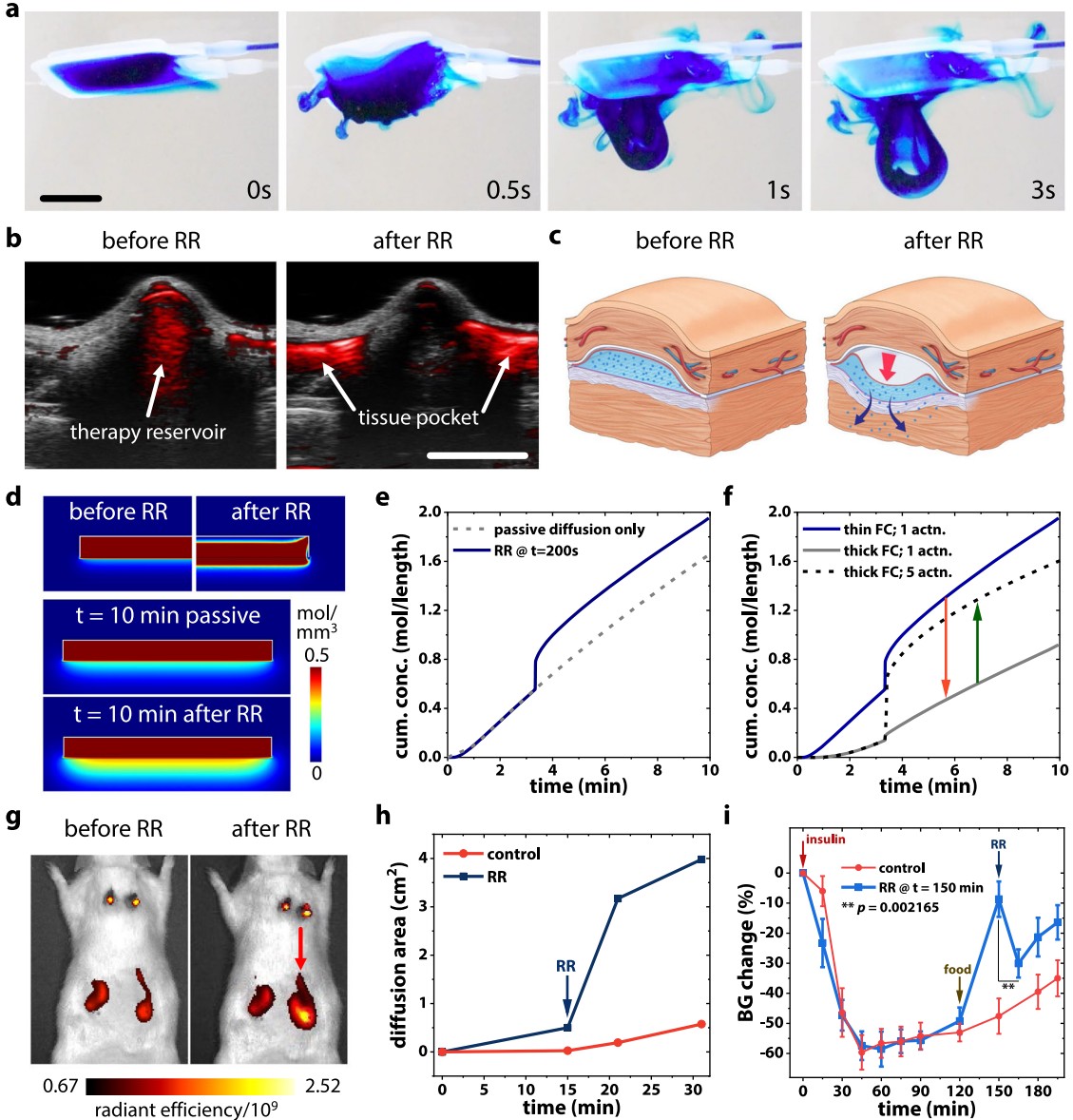

**Fig. 5 | On demand, actuation-mediated rapid release (RR) of drug using STAR. a** RR enables convective flow of a model drug, methylene blue, from the therapy reservoir of STAR. Scale bar is 5 mm. **b** Photoacoustic images showing subcutaneously implanted STAR in a rat model: RR enables convective flow of drug analogue (red) from the therapy reservoir into the surrounding tissue pocket. **c** Schematic showing actuation-mediated RR overcoming fibrous encapsulation. **d** Snapshots from COMSOL Multiphysics simulation showing the rate-limiting diffusion barrier created by a FC and the ability to improve transport using RR. **e** Concentration of drug outside the FC comparing passive diffusion alone to RR at 200 s. **f** Concentration of drug outside the FC for a thin (100 μm) or thick (200 μm) FC with 1 or 5 RR actuation cycles. **g** In vivo images of rat model with two STAR devices implanted. Fluorescence shows the distribution of drug analogue Genhance 750. Red arrow indicates the device after undergoing RR actuation. **h** Temporal evolution of the drug diffusion area of Genhance 750 in passive (control) and RR actuated STAR, quantified by fluorescent IVIS imaging. **i** Blood glucose response to insulin in control (passive diffusion only) and in RR actuated (at $t = 150$ min) STAR devices, at 2 weeks following implantation. $n = 4$ animals per group; data represents means ± standard error of mean. $p$ value calculated from paired one-tailed $t$-test.

after STAR implantation (Fig. 5i). Passive diffusion of insulin led to a drop in blood glucose over 120 min in all animals. At this point, food was given to one group to allow recovery of blood glucose levels towards baseline. At 150 min, this group was subjected to 5 cycles of actuation (with the same parameters of 2 psi at 1 Hz). No additional insulin was administered after the initial dose given at the start of the ITT. Despite a reduced insulin concentration gradient across the device and attenuated insulin sensitivity in the post-prandial animals, actuation-mediated RR led to a significant reduction in blood glucose levels over 15 min due to augmented release of insulin from STAR (Fig. 5i).

These results support the development of an on-demand actuation-based method to enhance transport across a diffusion limiting FC.

## Minimally invasive surgical implantation and vision for clinical translation

We demonstrate in a human cadaver model that the soft and foldable characteristics of STAR lend themselves to minimally invasive implantation, establishing feasibility of clinical translation. Our choice of material (TPU) enables scalability to clinically relevant dimensions (80 mm × 120 mm) and integration of additional elements such as deployment and adhesive channels, without changing the

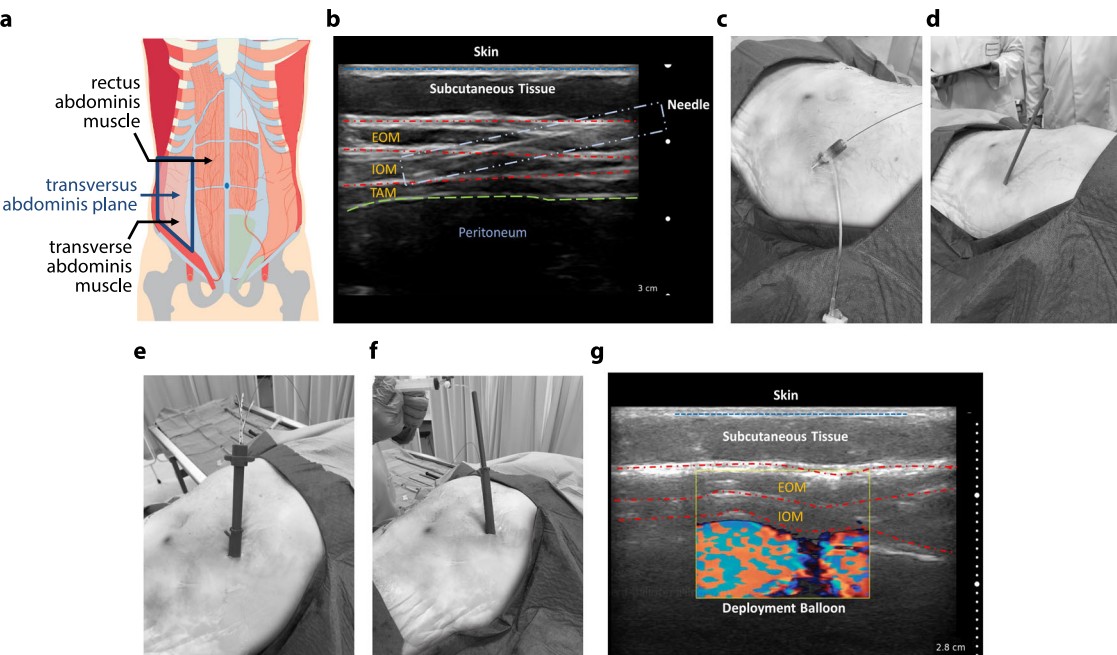

**Fig. 6 | Minimally invasive surgical implantation in a human cadaver model.**
**a** Location of the *transversus abdominis* plane in the anterior abdominal wall.
**b** Ultrasound guided needle access to the desired tissue intermuscular plane in the anterior abdominal wall and hydro-dissection to generate a potential space (EOM: external oblique muscle, IOM: internal oblique muscle, TAM: transversus abdominis muscle). **c** The Seldinger technique was used to get needle access to the *transversus abdominis* plane and a 5 Fr sheath was exchanged over a guide wire to maintain durable access to the tissue plane. **d** A commercially available dilator set is used to expand the space to accommodate positioning of the deployment sheath. **e** STAR advancement through sheath into tissue space. **f, g** An echogenic contrast agent was used to inflate the deployment channel to ensure complete unfolding of the STAR device within the plane using ultrasound guidance.

manufacturing process (Supplementary Fig 8). We designed a deployment system and surgical plan (Supplementary Fig. 9) to allow for minimally invasive implantation of STAR to an intermuscular delivery site through a 1 cm incision. The deployment system consists of a delivery sheath, space creating balloon, and delivery cartridge containing STAR. We selected the *transversus abdominis* plane, lying in the anterior abdominal wall between the internal oblique and *transversus abdominis* muscles, as the implant site (Fig. 6a). This potential space is well vascularised and is an established tissue plane that is frequently accessed by healthcare providers, e.g., for the delivery of analgesia during abdominal surgery[58,59].

Guided by ultrasound imaging, we first accessed the intermuscular *transversus abdominis* space with an 18-gauge needle in the anterior abdominal wall and used hydro-dissection to separate the tissue plane between the internal oblique and *transversus abdominis* muscles (Fig. 6b). Next, we utilised the Seldinger technique to exchange the needle over a wire[60] and verify correct tissue placement with ultrasound guidance (Fig. 6c). We then made a 1 cm skin incision to facilitate device delivery and used a commercially available dilator set to expand the space and accommodate positioning of our bespoke delivery sheath (Fig. 6d). Finally, we completed separation of the tissue planes with a space creating balloon, which could be visualised with ultrasound during filling. The scaled-up STAR device was then pre-loaded into the delivery cartridge, advanced easily through the sheath to the submuscular plane (Fig. 6e) and deployed. Pressurisation of the deployment channel with echogenic contrast enabled opening of the device under ultrasound visualisation (Fig. 6f, g). Postprocedural dissection of the tissue indicated that the device was successfully delivered to the appropriate space.

## Discussion

We present STAR, an implantable platform which can evade and overcome the diffusional barrier of the FC to achieve long-term enhanced therapy transport using two synergistic soft robotic strategies: (1) intermittent immunomodulatory actuation, and (2) actuation-mediated rapid release.

Prior to testing these mechanotherapeutic strategies, we developed a robust in vivo method (ITT) of detecting the effect of the FBR on macromolecular transport of insulin and monitoring this outcome over time (Fig. 2). The ITT has several advantageous features which aid technology development. Firstly, its measurement in real-time allows for agile feedback and iterative development. Secondly, the method allows for precise, quantitative assessment of the intervention using clinically relevant parameters including time to effect, maximum effect, or an integration of both time and magnitude via AUC. Finally, the repeated, non-invasive measurements enable longitudinal studies of complex, multiphasic phenomena with the ability to track individual animals and treatment groups over time.

IA is the first element in STAR's armamentarium. By inducing strain at the tissue-contacting membrane and perturbing peri-device fluid flow, STAR acts as an oscillating mechanical shield against the invading cellular FBR. These localised mechanical effects create a favourable environment for the long-term transport of macromolecules. IA is able to preserve the functional effect of STAR over 8 weeks at the same level as that seen immediately following implantation, i.e., prior to formation of a significant inhibitory FC (Fig. 3c). In sharp contrast, the insulin responsiveness of the control group decreased with longer implantation time, until near complete FC isolation and implant failure. The extension of therapeutic effect using a simple 5 min, twice-daily actuation regimen represents an attractive and innovative strategy for mitigating the FBR.

By combining ITT blood glucose results with a range of ex vivo capsular analytical techniques, we were able to unravel the multiphasic, temporal effects of IA on cell infiltration and capsule formation. When examining the inflammatory response in the IA and control groups, we detected significant differences in distinct cell populations at times of expected peak infiltration, despite there being no difference in total pericapsular cell number (Fig. 4).

We demonstrated that IA produces a localised immunomodulatory effect by clearing neutrophils from the pericapsular site (Fig. 4b, c). The infiltration of neutrophils is an important first step in the FBR, initiating and propagating the inflammatory process, which causes the subsequent recruitment of cell populations known to develop the FC (e.g., macrophages). Early modulation of the neutrophil response may have important long-term consequences on the FBR. Indeed, we observed a trend towards better functional effect in the 3W IA group compared to control even after stopping actuation. Though this effect did not reach statistical significance, there may be some long-term benefits in therapy transport even with initial periods of actuation; however, it is clear that maximal anti-inflammatory benefit comes with continued actuation (Fig. 3f). A recent study by Seo et al[49]. corroborates the mechanosensitive nature of neutrophil cell populations following dynamic loading for skeletal muscle regeneration. The authors postulated that mechanical flushing of chemoattractants was responsible for the reported decrease in neutrophils[49]. In this context, our work motivates further study on the effect of IA on pro-inflammatory chemoattractant gradients such as interleukin 1, 6 and 8 compared with direct mechanical effects on cell attachment, orientation, and function.

We noted a significant decrease in myofibroblast cell number with IA compared to control (Fig. 4d, e). The activation of matrix-producing cells into a myofibroblast phenotype, characterised by αSMA expression, is a critical step in fibrosis progression. Increased expression leads to heightened contractile activity, formation of stress fibres, and synthesis of extra-cellular matrix. Furthermore, activated fibrogenic cells can produce cytokines responsible for additional cell recruitment and propagation of the deleterious fibrotic response[43]. It is becoming increasingly evident that stiffness precedes, or is an important contributor to fibrosis[61]. Thus, reducing myofibroblast expression and its effect on matrix stiffness may be a key strategy to modifying this self-perpetuating fibrotic effect.

In addition to multiphasic cellular changes with time, we also observed multiphasic differences in macro-capsular architectural evolution, with distinct differences between the IA and control groups. At early timepoints (2 weeks) following implantation, we observed differences in capsule thickness between groups (Fig. 4h, i), which aligns well with improved diffusion-based transport at these timepoints (Fig. 3b, f). FC thickness equalises between groups after 8 weeks of implantation, indicating that other mechanisms are responsible for the improved transport at later timepoints. We rejected several relevant hypotheses for improved transport in the IA group including vascularity, capsule density, and collagen maturity (Supplementary Fig. 7). However, we noted differences in collagen architecture and cellular infiltration into the device reservoir which may contribute to late differences in transport and functional effects between the groups (Fig. 4j–l). Further study will be necessary to fully elucidate and understand the multiple mechanisms at play. Note that 2 out of 6 mice in the control group had to be withdrawn from the study at later time points due to self-inflicted damage to their dorsal tissue and subcutaneous implant. Interestingly, this did not occur in any IA group mice. Capsular contracture mediated by excessive FBR is a painful condition[62] that could rationalise this group difference, and future work could investigate this observation further.

In addition to IA, STAR possesses a second transport-augmenting strategy of RR. Actuation of a drug-loaded STAR can induce higher concentration and pressure gradients[31,32], and thus improve drug transport across a formed FC with temporal control (Fig. 5). RR can be particularly advantageous for potent drugs where accurate dosing and a rapid time to functional effect are important, or macro-drugs such as proteins, where convective flow enhances diffusion-based flux, principally governed by molecular weight and concentration gradient[33,34]. Convection-enhanced delivery has been successfully used to improve distribution of chemotherapy in deep brain tumour targets, albeit with modest clinical outcomes[26,27,32,63].

We envision several clinical use scenarios for IA and RR technologies demonstrated in this work. RR could be used independently of IA, such as for the rapid, on-demand delivery of therapeutics in response to a clinical emergency. Some relevant examples include the delivery of adrenaline for the treatment of anaphylaxis or glucagon for the treatment of hypoglycaemic coma. In both cases, impedance of delivery due to FC formation would have grave consequences.

An optimal FC mitigating strategy could also combine both IA and RR. For example, short daily bursts of IA (without drug) could attenuate the FBR to extend device lifespan and improve performance in long-term implantation. A temporally controlled and modular RR actuation regime could then be used to make rapid, precise dosage adjustments in accordance with the patient-specific clinical scenario and/or FBR severity. IA and RR are complementary strategies that elegantly make use of a single device design and pump, making STAR well suited to a variety of clinical scenarios.

The management of type 1 diabetes is one relevant clinical area where STAR could provide synergistic benefit. For example, IA could be applied to extend the lifespan of an artificial pancreas[64], preventing unnecessary FBR-mediated blockages, linked hyperglycaemic events, and ultimately simplifying the dosing regimen and patient experience. In synergy, actuation-mediated RR could make rapid insulin adjustments and maintain blood glucose levels in the narrow window necessary to prevent long-term complications[65]. Looking further into the future, application of STAR could enable translation of next-generation bioartificial technologies utilising human-derived insulin-producing islet cells by modifying the transport-limiting FC, which has been a major barrier to the viability of cell-based therapeutics[18,66–68]. Considering the need for daily IA for long-term efficacy, it is likely that embodiments will require a wearable pump, similar to those used in existing insulin pumps[69].

STAR's soft material endows it with biocompatibility advantages over rigid implantable drug delivery systems[70], and lends itself to minimally invasive catheter implantation. In a step towards clinical translation, we scaled up the STAR device and developed a bespoke delivery tool as well as a minimally invasive procedure that is congruent with conventional interventional radiology techniques (Fig. 6). We demonstrated delivery of STAR to a clinically accessible intermuscular space in the anterior abdominal wall in a human cadaver model. This approach allowed for a short procedure time (<20 min) for implantation of a human-scale device through a modified sheath using ultrasound guidance in the hands of an experienced interventional radiologist. Additional design features demonstrated correct deployment and adhesive delivery to maintain device position in the tissue plane. Thus, STAR can be quickly implanted by interventionalists in an outpatient setting under local anaesthesia using an established imaging modality.

Though we have demonstrated robust preclinical results in a long-term mouse model, there are several limitations and barriers to clinical translation. Our findings from implantation in the dorsal subcutaneous space of mice may not directly predict similar results in humans at different anatomical locations (e.g., abdominal intermuscular space), and further work is needed to understand how these anatomic and microenvironmental differences impact the effects of STAR. Although rodent models have been extensively used to study the FBR[9,26], rodents have been shown to have different tissue collagen content in the subcutaneous space surrounding an implant and different metabolites in the interstitial fluid at implant interfaces, as compared to humans[71]. Moreover, differences in rodent skin, fur, and behaviours may subject implanted devices to different biomechanical forces than in humans[51]. Encouragingly, thus far, we have observed similar FBR mitigating effects with an IA regimen that is agnostic of species and device design[34]. In addition to this, the presence of conserved inflammatory pathways in FC formation across species[72] suggests that STAR may have a similar benefit in prolonging implant lifespan in humans.

Although there have been a number of prior studies that examine the effect of mechanical loading on inflammation and tissue regeneration, there is significant heterogeneity with regards to actuation methods, regimens, resulting deformations, target tissues, and animal models used[44–50]. Only a few studies have attempted to address the effect of varying tissue strain[47,49,50] and loading frequency[50]; therefore, significant work is needed to define the optimal loading parameters that maximise the anti-inflammatory effects of mechanical actuation, which may differ with the type of tissue and mechanical stimulus.

We can draw six conclusions from this study: (1) The ITT represents a robust, longitudinal method to monitor macromolecular therapy transport across a developing FC in vivo. (2) The FBR can negate insulin transport over time from a static STAR device until complete implant isolation and therapy failure. (3) Intermittent actuation can preserve therapy transport at baseline levels and extend the therapeutic lifespan of STAR, even with long-term implantation. (4) IA can mediate immunomodulatory changes in the neutrophil inflammatory response and elicit downstream multiphasic temporal changes in cellular infiltration and capsule formation. (5) Actuation-mediated RR of a drug-loaded STAR device can synergistically enhance mass transport and therapeutic effect with tunable, temporal control, despite the presence of a FC. (6) Minimally invasive catheter implantation of STAR was possible in a human cadaver model, showing clinical translatability of our approach.

In summary, the STAR platform represents a new mechanotherapeutic approach to both mitigate and overcome the FBR, extending the lifespan and efficacy of implantable drug delivery devices. It holds vast clinical utility for a variety of indications where transport is affected by fibrosis, such as the management of type 1 diabetes.

## Methods

### Device manufacture for pre-clinical mouse model
One and two-channel positive, and corresponding negative moulds were 3D printed using VeroBlue resin (Stratasys Objet30) (Supplementary Fig 2a). Thermoplastic urethane (TPU; 0.3 mm, XGD0385, QING GEN) was vacuum thermoformed (Yescom Dental) over the positive two-channel mould (Supplementary Fig 2b). This process was then repeated with the positive one-channel using a thinner TPU (0.076 mm, HTM-8001-M, polyether, American Polyfilm) (Supplementary Fig 2b). Pores with 10 μm diameter were laser cut in a TPU membrane (0.076 mm, HTM-8001-M, polyether, American Polyfilm) using a UV Nanosecond laser (National Centre for Laser Applications, National University of Ireland Galway).

The thermoformed and laser-cut membranes were assembled in a negative mould (Supplementary Fig 2c). Mandrels of outer diameter 0.21 mm were inserted into the channels to retain patency. The assembly was heat sealed together using a heat transfer machine (330QXAi, PowerPress). The mandrel was removed and a TPU catheter tubing (0.037″ × 0.023″; MRE037, Micro-Renathane, Braintree Scientific) was inserted and heat sealed to the device using heat shrink tubing. The final assembled devices measured 15 mm (width) × 18 mm (length) × 2 mm (height) and consisted of two chambers—the larger measuring 12 × 6 mm and the smaller measuring 3 × 12 mm (Supplementary Fig 3a, b).

### Device manufacture for pre-clinical rat model
Rat scale devices were produced as previously described[34]. Final devices measured 12 mm in length with the hemispherical reservoir measuring 3.9 mm in height and 3.5 mm in diameter, with variable lengths of 3Fr TPU catheter tubing.

### Device manufacture for human-scale prototype
Human scale devices measuring 120 × 80 mm were produced as previously described (Supplementary Fig. 8)[73]. An additional deployment

channel and actuation chamber were included to allow for minimally invasive delivery through a sheath and dynamic actuation following implantation (Supplementary Fig. 9).

### Electropneumatic actuation and control system
A custom-made electropneumatic system to deliver actuation to the implanted device was developed as described in Supplementary Fig 6. The system consisted of pre-programmed electrical signalling to control pneumatic power sources. Pneumatic components included a positive pressure and vacuum generator, a pressure regulator, and electropneumatic (solenoid) valves. A programmable microcontroller board (Arduino Uno) along with a power source were used to establish an open loop control of the pneumatic power. Positive pressure was guided through an electropneumatic pressure regulator (ITV1030; SMC Inc.) which was controlled via the microcontroller board to adjust the precise actuation pressure. Actuation of the implanted device was then achieved by alternating the positive pressure for device expansion and negative pressure for device deflation. The delivery of this pneumatic actuation pattern was ensured by two electropneumatic solenoid valves (NVKF333; SMC Inc.) for positive and negative pressure which were controlled using the same microcontroller and two MOSFETs coupled to the electrical supply power (Supplementary Fig. 6a). A manifold was used to actuate multiple devices in separate animals simultaneously, ensuring that the set pressure level was consistently achieved on all the manifold channels (Supplementary Fig. 6b, c).

### STAR membrane deflection characterisation
STAR devices were manufactured and placed in a custom-made 3D printed holder (Objet30 Prime, Stratasys). Devices were pneumatically inflated from 1 to 9 psi using the electropneumatic actuation and control system described above. Images of membrane deflection were captured using a digital camera (Nikon DLSR) and tripod positioned in the side view. Deflection magnitude was subsequently analysed using ImageJ. Based on this strategy, a bi-chambered configuration was selected to investigate the effect of two distinct deflection magnitudes (0.58 and 1.3 mm) in our pre-clinical mouse model. It should be noted that the lower deflection magnitude was closely matched to our previous work.

### STAR membrane pore characterisation
5 mm × 5 mm pieces of laser-cut porous TPU membranes were characterised by scanning electron microscopy (SEM) using a Hitachi S2400 microscope operating at 20 kV electron accelerating potential in backscatter electron imaging mode and a sample working distance between 8 and 10 mm. After imaging, pore diameters were measured from the images using the Hough circle transform function in Fiji 2.0.0 (ImageJ) (Supplementary Fig. 1).

### Computational modelling
Fluid-structure interaction (FSI) simulations using smoothed particle hydrodynamic (SPH) method were conducted to investigate the peri-implant fluid flow and the dynamics of drug transport under active delivery. All FSI simulations were created using Abaqus/Explicit 2018 (Dassault Systèmes, Vélizy-Villacoublay, France). The device was modelled as a 3D surface geometry and meshed with 14,636 four-node shell elements (Abaqus node type S4R). A Dirichlet boundary condition, where nodal displacement in all directions were fixed to zero, was applied on the edge of the bottom porous membrane to prevent rigid body motion. A pressure loading which linearly ramped up to 2 psi in 500 ms and ramped down to 0 psi in the next 500 ms was applied to the internal surface of the outer and middle membranes. The membranes were modelled using an Ogden 3rd order hyperelastic material with parameters, $\mu_1 = -8.31$ MPa, $\mu_2 = -0.36$ MPa, $\mu_3 = 17.89$ MPa, $\alpha_1 = 0.46$, $\alpha_2 = 3.62$, $\alpha_3 = -3.10$. The fluid domains, drug and outer fluid, were meshed with linear tetrahedral elements and each element was

converted to an SPH particle located at its centroid. The drug domain and the outer fluid domain contained 107,406 and 174,516 particles, respectively. The particles were assigned with the following properties: density of 9.96E-7 kg/mm$^3$, bulk modulus of 2.094 GPa and dynamic viscosity of 3.56E-8 MPa-s.

A structural finite element (FE) model was constructed to investigate the deformation of the tissue underneath the device. The device was modelled with same geometry and material model as the FSI simulations. To model the skin constraint when the device is implanted subcutaneously, a dome shape shell structure with a linear elastic property ($E = 1$ MPa) was added on top of the device. The tissue was also modelled as a linear elastic material ($E = 15$ kPa). In terms of boundary conditions, the edge of the skin and the bottom face of the tissue were both fixed in all directions. Tie constraints were used to model the suture attachments between the device and the tissue. A general contact with friction coefficient of 0.5 was applied throughout the entire model. The inner chamber of the device was subjected to different linearly increasing pressure loadings (1 psi, 2 psi and 3 psi). From the FE simulations, strain contour plots and downward displacement contour plots of the tissue were extracted. The average strain and deflection were calculated at the interface between the device and the tissue.

Mass transport simulations were conducted using COMSOL Multiphysics software version 5.6 (COMSOL, Burlington, MA). The 2D model contains three domains: the drug reservoir, the FC, and the outer fluid domain representing the body. The FC is a thin layer surrounding the device with thickness ranging from 50 μm to 200 μm. The outer fluid domain is a rectangular region with height and width of 5 mm and 20 mm, respectively. Mass transport was modelled using the transient diffusion-convection equation in the 'Transport of Diluted Species' module in COMSOL. Diffusivity in the reservoir and outer fluid domain was given as 855 μm$^2$/s and the diffusivity in the FC was given as 50 μm$^2$/s[74]. Initial concentration of 1 mol/mm$^3$ was applied in the drug reservoir domain. Fluid velocity between pure fluid domain and porous media was calculated using the Brinkman Equations in COMSOL. The drug reservoir and the outer fluid were modelled as pure fluid domain with density of 997 kg/m$^3$ and viscosity of 8.9E-4 Pa-s. The FC was modelled as porous media with permeability of 8.9E-16 m$^2$ and porosity of 0.8. Pressure inlet of 2 psi ramped up and down within 1 s was applied on the boundary of the reservoir when device actuation was triggered; and zero outlet pressure was applied on the boundary of the outer fluid domain. The model showed the transient changes of drug distribution profile within 30 min under diffusion and convection. In all simulations, both local concentration and regionally accumulated concentration were extracted over time as quantitative outputs.

### Péclet number calculations

The Péclet number for mass transfer, for a characteristic length $L$, is defined as $Pe_L = uL/D$, where $u$ is the local flow velocity and $D$ is the diffusivity[57]. For representative calculations, $L = 1$ mm was assumed. This assumption is based on experimental and computational data showing that the maximum device membrane deflection is ~1.5 mm and the estimated tissue deflection is ~0.73 mm (Supplementary Fig. 3). A diffusivity of $D = 855$ μm$^2$/s, which is the same value used in the COMSOL Multiphysics models above, was used[74]. For the passive diffusion scenario, a reasonable estimate for $u$ is the velocity of interstitial fluid, which has been reported widely to be in the range 0.1–2 μm/s[75]. A flow velocity of 0.06 mm/s adjacent to the porous membrane immediately following actuation was calculated from the COMSOL simulations above.

### Pre-clinical studies

Animal procedures were reviewed and approved according to ethical regulations by the Institutional Animal Care and Use Committee at Massachusetts Institute of Technology. Animals were housed in a facility with 12 h on/off light cycle, at 20–22 °C with a relative humidity ranging between 30 and 70%. Animals were singly housed with standard bedding and food for the duration of the study. All devices were sterilised using ethylene oxide before implantation.

**Mouse pre-clinical surgery.** Male C57BL/6 mice (25–30 g) were placed under anaesthesia using inhalable isoflurane (1–3%). A single dose of sustained-release buprenorphine (Bup-SR, 1 mg/kg) was administered subcutaneously to control pain. A STAR device was implanted subcutaneously in the mouse as depicted in Supplementary Fig. 4. To prepare the surgical site, the hair on the back of the mouse was removed using a clipper and topical depilatory cream, and the site was sterilised with three washes of Povidone-iodine and 70% ethanol. Medial dorsal incisions were made at the base of the neck and 1 cm from the tail (Supplementary Fig. 4a). A blunt dissection was made at the incision sites, and a curved haemostat was used to tunnel subcutaneously from the superior to the inferior sites. A transcutaneous self-sealing port available from Instech laboratories (VABM2B/22R22) was connected to the dorsal end of the therapy and actuation catheter of each STAR device. The 15 × 18 × 2 mm STAR device (Supplementary Fig. 3a, b) was then inserted under the skin via the superior incision site and tunnelled inferiorly into position. The device was secured to the underlying fascia with one suture at either side (7-0 monofilament). The port was then inserted under the skin at the superior incision site (Supplementary Fig. 4b). The skin at each incision site was then closed with interrupted sutures (5-0 Maxon monofilament) (Supplementary Fig. 4c), and the animal was allowed to recover on a heated pad. To replete intraoperative fluid losses, 0.2 mL of warm saline was administered subcutaneously.

**Insulin transport test (ITT).** A kinetic measure of insulin release from STAR devices and subsequent effect on blood glucose levels was measured through an ITT. The mouse was weighed and a solution containing a dose of 1 IU/kg/150 μL was prepared from a stock solution of Humulin R U-100 short-acting human insulin. Animals were fasted for 4 h prior to the start of the ITT, and were kept in a clean cage without food and bedding for the duration of the test. An initial blood glucose measurement was taken to establish a baseline. An insulin preparation at a total dose of 2 IU/kg was then administered into the device via the transcutaneous self-sealing port available from Instech laboratories (VABM2B/22R22) at time = 0 min. A PNP3M connected to a 1 mL Luer Lock disposable syringe (BD) was used to administer the dose (Supplementary Fig 4e). Following administration, blood was sampled from the lateral tail vein of the mouse and serial blood glucose measurements were performed over 120 min using a Bayer Contour Next Blood Glucose Monitoring System at time = 15, 30, 45, 60, 75, 90, and 120 min. The animal was restrained using a commercial restrainer (TV-RED 150-STD, Braintree Scientific). The tail was warmed using a HotHand warmer for 10 s prior to blood sampling. The area was then disinfected with a Kimwipe soaked in 70% ethanol. Finally, venepuncture was performed using a 27-gauge needle (BD) and the measurement was recorded.

**Intermittent actuation.** Intermittent actuation was performed by connecting a custom-made electropneumatic actuation and control system (described above) to the self-sealing transcutaneous actuation port using a PNP3M connecter (Instech) (Supplementary Fig. 4d). The device was then cyclically actuated at a controlled input pressure of 2 psi at 1 Hz for 5 min every 12 h as previously described[34,46]. No drug was present in the device during IA throughout the study.

**Long-term mouse preclinical study.** Devices were implanted into male C57BL/6 mice as described above. ITTs as described above were performed in all mice at 2, 3, 4, 5 and 8 weeks following

device implantation, after which point animals were euthanized by $CO_2$. Six devices were static controls, and ten devices were dynamically actuated for 5 min every 12 h. The actuation group was split at 3 weeks post-actuation, with one group ($n = 5$) stopping actuation and remaining passive thereafter, and another group ($n = 5$) continuing dynamic actuation for the entire 8-week study period (Fig. 3a).

**Actuation-mediated rapid release of insulin.** At 2 weeks following implantation, serial blood glucose measurements were performed over 120 min after insulin injection as described above. Food was given to the RR group at 120 min to allow recovery of blood glucose levels. Actuation-mediated RR was then performed by actuating the STAR device at 150 min. Note that no additional insulin was administered after the initial dose given at time = 0 min. The STAR actuation reservoir was connected to a custom-made pneumatic control unit, via the transcutaneous self-sealing access port, using a PNP3M connecter (Instech) and pneumatically activated for 5 cycles of 2 psi cyclical pressure at 1 Hz. The duration of activation and evacuation were equivalent. Blood glucose levels were measured by tail vein sampling at four additional 15 min increments in 4 mice per group.

**Rat pre-clinical study.** Two female Sprague Dawley rats (250–300 g) were placed under anaesthesia using inhalable isoflurane (1–3%). Bup-SR (1 mg/kg) was administered subcutaneously to control pain. To prepare the surgical site, the hair on the back of the rats was removed, and the sites were sterilised with three washes of Betadine and 70% ethanol. A superior incision was made at the base of the neck for the port, and two inferior incisions were made 9 cm from the original incision along the back of the rat and 1 cm lateral of the spine. A blunt dissection was made at all incisions, and a pair of forceps was used to tunnel subcutaneously from the anterior to the posterior sites. A transcutaneous self-sealing port available from Instech laboratories (VABM2B/22R22) was connected to the dorsal end of the therapy and actuation catheter of each STAR device. The ports were placed in position at the base of the neck, and the devices were tunnelled posteriorly into position. Each port was secured to the underlying fascia using at least one interrupted suture (5-0 monofilament). Each STAR device was secured to the underlying fascia with one suture at either side (7-0 monofilament). The skin was closed with interrupted sutures (5-0 monofilament), and the animal was then allowed to recover on a heated pad. Warm saline (300 μL) was administered subcutaneously to mitigate fluid loss and dehydration caused by surgery. Animals were euthanized by $CO_2$.

**In vivo imaging.** On day 24 of the rat preclinical study, release of a small molecule fluorescent drug analogue, Genhance 750 (Perkin Elmer), was assessed using IVIS Spectrum in vivo imaging system (PerkinElmer). An excitation filter of 745 nm and an emission filter of 800 nm was used to acquire the images. First, the animals were anaesthetised using inhalable isoflurane. The hair was removed above the subcutaneous STAR reservoir, and the circumference of each reservoir was marked on the skin, to aid identification of the region of interest (ROI). After preparation, a control image was acquired. 35 μL of the Genhance drug analogue was then injected via the transcutaneous port and refill line into each STAR reservoir using a syringe pump (Harvard Apparatus). The refill line and reservoir were first cleared by applying negative pressure. Images were acquired every 3 min in a sequence. After 14 min following injection of Genhance into STAR, imaging was paused, and the intervention reservoir was pneumatically activated. Imaging was then restarted. Using a consistent image threshold, a custom ROI was used to delineate the area of diffusion in each image (Living Image 4.5.4, PerkinElmer). The initial area of diffusion at $t = 0$ following injection was subtracted from all subsequent readings for analysis.

**Photoacoustic ultrasound.** Methylene blue solution (1 mg/mL) was loaded into the STAR reservoir, and the actuation line of the device was filled with water using a three-way valve (Qosina), as air could cause interference with the ultrasound signal. The device was implanted subcutaneously in one Sprague Dawley rat just after euthanasia. Ultrasound and photoacoustic imaging (PAI) were performed using the Vevo LAZR-X Photoacoustic and micro-ultrasound Imaging System (FUJIFILM VisualSonics, Toronto, Canada) using a MX550 transducer at a frequency of 40 MHz, and a resolution of 40 μm. The narrow Vevo Optical Fibre, composed of high-efficiency fused silica, and surrounded by a MX550 Fibre Jacket was used to perform Multispectral PAI imaging.

The probe was moved across the skin, and the subcutaneous device, using a 3D stepper motor to obtain a 3D data set. Using Nanostepper mode, each slice of the 3D scan consisted of images taken at 680, 730, 750, 800, 850 and 900 nm wavelengths. In addition, a Spectro mode acquisition was performed at a single slice, where images were taken between 680 and 970 nm in 5 nm increments. Images were rendered using VevoLAB 3.2.0 software (FUJIFILM VisualSonics, Toronto, Canada).

### Human cadaveric studies
The protocol for this study was approved by the National University of Ireland Galway (NUI Galway) Research Ethics Committee. All cadaveric material was bequeathed to the Medical School, NUI Galway, for further advancement of medical knowledge. Informed consent was obtained from next of kin as part of the bequeathment process. This is covered by legislation governing the practice of Anatomy in the Republic of Ireland (Medical Practitioners Act 2007). Whole adult male human cadavers were fixed with embalming fluid containing 21% methanol, 21% glycerine, 5.6% phenol and 3.1% formaldehyde. A Seldinger technique was used to access the *transversus abdominis* plane under ultrasound guidance[60]. The delivery system consisted of a delivery sheath, a space-creating balloon and the STAR delivery cartridge (Supplementary Fig 9). The delivery sheath and STAR delivery cartridge were directly 3D printed (Form 2, Formlabs, Somerville, MA). The space-creating balloon consisted of a 3D printed shaft connected to a TPU balloon. A 4–14 MHz linear ultrasound transducer (Clarius) was used to visualise the muscular layers of the anterior abdominal wall in the cadaver. An 18-gauge needle was advanced into the transversus abdominis plane and physiological saline as injected to separate the muscle planes with hydro-dissection. The needle was exchanged over a 5 Fr, 10 cm sheath, and a 1 cm skin incision was made using a scalpel. An 0.035″ Amplatz Super Stiff wire (Boston Scientific) was then advanced into the space, and the serial dilation was performed using an Amplatz-type renal dilator set (Boston Scientific), followed by the advancement of the custom-made delivery sheath into the tissue plane. The space-creating balloon was used to fully separate the tissue planes, and then exchanged for the delivery cartridge. STAR was then deployed using the cartridge into the intermuscular space. The cartridge and delivery sheath were removed, and reservoir filling was demonstrated with infusion of an ultrasound contrast agent (SonoVue) and visualised under ultrasound. Proper deployment of the device and positioning was confirmed with dissection.

### Histology and immunohistochemistry
After euthanizing the animals with $CO_2$, each device and the immediate surrounding tissue were extracted. Tissues were fixed for 24 h using 10% formalin (pH 7.4). The tissue was then washed and stored in PBS. Fixed tissue samples were transected in half, oriented, and embedded in paraffin wax blocks for histological and immunohistochemical analyses. Each block was assigned a code for randomisation and blinding purposes. Sections of 7 μm were cut, deparaffinized in xylene, and rehydrated through a series of graded alcohols. For assessment of FC collagen maturity and arrangement, sections were stained in 0.1%

Fast Green and then in 0.1% Sirius red in saturated picric acid. Slides were then dehydrated through graded alcohols and cleared in two changes of xylene. The slides were cover slipped using DPX mounting medium and left to dry. Slides were imaged using Ocular 2.0 Imaging Software on an Olympus BX4 polarised light microscope (Mason Technology Ltd. Dublin, Ireland) at 20× magnification.

To determine total cell number per area, samples were stained with haematoxylin and eosin following widely established protocols[76]. Ocular 2.0 Imaging Software on an Olympus microscope (Mason Technology Ltd. Dublin, Ireland) at 40× magnification. For immuno-histochemical analysis, primary antibodies of CD31 (1:200; Ab182981, Abcam) and αSMA (1:500; ab7817, Abcam) were incubated for 1 h at 37 °C. Secondary antibodies of Alexa Fluor 594 goat anti-mouse immunoglobulin G (IgG; 1:200 Thermo Fisher Scientific), Alexa Fluor 594 goat anti-rabbit IgG (1:200; Thermo Fisher Scientific), and Alexa Fluor 488 goat anti-mouse IgG (1:200; Thermo Fisher Scientific) were incubated for 1 h at room temperature, respectively. Primary antibody Ly-6G (1:100; Biolegend 127602) was incubated for 1 h at 37 °C after Tris-EDTA (pH 9) antigen retrieval. A ready probes mouse-on-mouse (Invitrogen, R37621) blocking solution was performed to block endogenous binding. Secondary antibody Alexa Fluor 488 goat anti-rat 488 (1:100; Thermo Fisher Scientific) was incubated for 1 h at room temperature. Sections were stained with Hoechst and cover slipped using fluoromount. Immunofluorescence-stained slides were observed using a spinning disc inverted confocal microscope (CSU22, Yokagawa) combined with Andor iQ 2.3 software. For blood vessel analysis, CD31 (1:200; Ab182981, Abcam) was incubated for 1 h at 37 °C after sodium citrate antigen retrieval (pH 7.2). A Dako EnVision+ System−HRP (DAB) secondary and counterstain (haematoxylin) was applied following manufacturer instructions.

**Analysis of total cell number.** Five random fields of view were acquired from two sections using light microscopy. Images were cropped to ensure only the capsule was included in the analysis. Sections were converted to 8-bit images and nuclei were manually thresholded from background tissue. Particles were outlined and analysed using ImageJ (Fiji version 2.0.0) software.

**Analysis of myofibroblasts and neutrophils.** Ten random fields of view were acquired from two sections using confocal microscopy. The volume fraction of αSMA+ cells and Ly-6G + cells within the FC were estimated by an unbiased stereological counting technique using ImageJ (Fiji version 2.0.0) software. The same unbiased stereological method was used to count both myofibroblasts and neutrophils. A random offset stereological square grid (10,000 μm²) was superimposed onto the images to provide test points. To calculate area fraction, intersections falling on positively stained cells were counted and expressed as a ratio of total grid intersections within the FC. To estimate the relative volume of myofibroblast cells per FC and assess whether the presence of myofibroblasts was perturbed by actuation, the volume fraction of αSMA + cells was normalised to a volume defined by FC thickness multiplied by unit area using the following equation: Relative Volume of αSMA+ cells (mm³) = (Volume Fraction of αSMA+ cells)/FC Thickness (mm) × 1 mm × 1 mm.

**Numerical density of blood vessels per unit area.** Quantification of vascularisation of the FC was performed using a previously reported technique[5,77,78]. A systemic random sampling strategy was used. From each tissue section, ten non-overlapping images were taken of the FC in the area of interest. Number of blood vessels per area (Na) were calculated using an unbiased counting frame (grid size = 2000 μm²). The numbers of points which coincided with blood vessels were counted. The volume fractions of blood vessels were then calculated by expressing the proportion of points hitting blood vessels as a fraction of the total number of points observed in the tissue. The

forbidden line rule was followed to ensure blood vessels were only counted once. The application of this counting rule generates an unbiased estimate of the number of blood vessels per unit area ($N_a = C_N \times C_{pts} \times A$; where $C_N$ is the cumulative number of blood vessels counted and $C_{pts}$ is the cumulative number of points in the area of interest).

**Analysis of collagen orientation.** Quantification of the collagen content was performed using a previously reported technique[5,34,79–81]. In ImageJ, six ROIs were manually selected for each image obtained by polarised light microscopy to completely capture all collagen fibres imaged while minimising noise from background artefact. Using the OrientationJ 2.0.5 plugin, the coherency of each ROI was calculated. Each device sample generated 60 total datapoints (6 ROIs per section, 10 sections per sample) (Fig. 4k).

**Micro-computed tomography analysis**
Following μCT imaging of implanted devices, FC thickness was quantified using Materialise MIMICS Research 18.0.0.525 and Materialise 3-matic Research 10.0.0.212 software. DICOM files generated from μCT scans were imported to MIMICS, and a threshold mask was applied to the field of view allowing the FC to be visually identified by a change in signal intensity superior to the fascia layer. The threshold region was cropped to include only the region of the FC, and manual segmentation of the FC in the sagittal view was performed. This was repeated on every five DICOM slices, with interpolation between slices performed to mask the FC in the intermediate regions. Upon completion of the masking process, a rectangular section was isolated from the centre of the mask. An STL file of this rectangular section was exported for thickness analysis using 3-matic Research. In 3-matic, the fix wizard was used to fix errors in the STL file and a smoothing factor of 0.8 was applied, following which the model was remeshed using the auto-remesh feature. A wall thickness analysis was generated, and data were exported for statistical analyses.

Wall thickness of the isolated FC under both the small and large chambers was measured (Supplementary Fig 10a, b), which demonstrated no difference. Similarly, thickness measurements were taken at device edges which did not differ significantly from thickness values underneath device chambers (Supplementary Fig 10c, d).

**Density analysis.** The density of tissue in the fibrotic capsule was measured using the 'Density in Rectangle' function in Materialise MIMICS Research 18.0.0.525 on μCT scans of explanted STAR devices. Three radiodensity measurements of rectangular sections (0.01 mm²) of fibrotic capsule were measured under each chamber, at five locations across the span of the STAR device. Background radiodensity measurements were obtained from non-sample regions of the scan and were subtracted from the average radiodensity reading for the FC under each chamber.

**Scanning electron microscopy**
Following μCT imaging, devices with associated surrounding tissue were processed for SEM. Tissue samples were first trimmed of excess tissue and hair was removed to prevent unwanted interaction with the electron beam. Samples were first dehydrated through graded alcohols (70%, 95%, 100%, 100%) before being placed into a LEICA EM CPD300 critical point dryer for 3 h. Dried samples were then mounted onto aluminium stubs with carbon adhesive tabs and gold sputter-coated using a Quorum Q150R ES plus. Samples were observed and imaged at 40× magnification 15 kV on a HITACHI S-26OON Scanning Electron Microscope.

**Statistical analysis**
Unpaired, one-tailed, two-sample t-tests were used to assess the differences in blood glucose levels and FC thicknesses between

groups. Unpaired, two-tailed, two-sample *t*-tests were used to assess the histological, immunohistochemical, vascularity, and radiodensity differences between the control and IA groups. Before performing *t*-tests, equality of variances was verified between groups using Levene's test. The analyses were performed in OriginPro 2018b (OriginLab Corp.) and the same software was used to generate all plots. Data are represented as mean ± standard error of mean. Between-group differences were evaluated at a significance level of 95% ($\alpha = 0.05$). The Bonferroni correction was applied for multiple comparisons by dividing $\alpha$ by $m$, where $m$ is the number of comparisons. $p$ was deemed significant at $p < \alpha/m$. Exact $p$ values for all statistical comparisons are presented in Supplementary Note 1.

### Reporting summary
Further information on research design is available in the Nature Research Reporting Summary linked to this article.

### Data availability
All data supporting the findings of this study are available within the article and the Supplementary Information. Data is available from the corresponding author(s) upon request.

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

## Acknowledgements

W.W., S.T.R., R.B., and G.P.D. acknowledge support from Science Foundation Ireland under grant SFI/12/RC/2278, Advanced Materials and Bioengineering Research (AMBER) Centre, Royal College of Surgeons in Ireland, National University of Ireland Galway, and Trinity College Dublin, Ireland. W.W., E.B.D., and E.T.R. acknowledge a Pilot and Feasibility Grant from the Juvenile Diabetes Research Fund (1-PNF-2019-778-S-B). N.A.W. and E.B.D. acknowledge funding from the Science Foundation Ireland Royal Society University Research Fellowship (URF\R1\191335). S.X.W. acknowledges funding from the National Institutes of Health training grant T32 HL007734. S.T.R. has received funding from the European Union's Horizon 2020 research and innovation program under the Marie Skłodowska-Curie Actions Grant Agreement No. 713567. D.S.M. acknowledges funding from the Irish Research Council Government of Ireland Postgraduate Scholarship (GOIPG/2017/927) and a Fulbright Enterprise Ireland Award. E.B.D. and G.P.D. acknowledge the DRIVE project which has received funding from the European Union's Horizon 2020 Framework Programme under Grant Agreement No. 645991. E.T.R. acknowledges departmental funding from the Institute for Medical Engineering and Science and the Mechanical Engineering Department at the Massachusetts Institute of Technology and funding from NSF EFRI grant 1935291. We thank the Centre for Microscopy and Imaging (NUI Galway) and Ciaran Weldon for assistance with SEM imaging. We thank Dr. Bo Ri Seo for providing the staining protocol for the neutrophil immune response.

## Author contributions

W.W., D.G., S.X.W., G.P.D., E.B.D., and E.T.R. designed the study. W.W., D.G., S.X.W., N.A.W., R.E.L., R.B., S.T.R., D.S., R.O'C., D.S.M., K.L.M., C.E.V., M.A.H., J.O'D., and A.S.R. performed the experiments. Y.F. performed computational modelling. W.W., D.G., S.X.W., Y.F., N.A.W., R.E.L., R.B., L.T., D.A.D-L, R.W., and E.B.D. analysed and reviewed the data. W.W., D.G., S.X.W., Y.F., N.A.W., R.E.L., S.T.R., G.P.D., E.B.D., and E.T.R. wrote the paper. All authors reviewed and edited the paper.

## Competing interests

W.W., S.T.R., K.L.M., C.E.V., G.P.D., E.B.D., and E.T.R. are inventors on a pending patent application related to the device described here. The other authors declare no competing interests.
