## [Peer Review File · Nature Communications]

Dynamic actuation enhances transport and extends therapeutic lifespan in an implantable drug delivery platformREVIEWER COMMENTS

Reviewer #1 (Remarks to the Author):

Thank you for the opportunity to review this excellent work. The authors describe STAR, an implantable system that incorporates mechanotherapeutic methods to modify and overcome the foreign body response which forms around implantable devices. The problem of FBR is quite significant and an implantable device that could successfully overcome this would be remarkable and quite impactful. The authors elegantly describe experiments that seek to answer mechanistic questions about how mechanical modulation might impact FBR and the capacity for drug delivery from the implant. However, the experiments fall short of answering a few fundamental questions that this reviewer believes are vital to enable the impact of these experiments to be fully realized. Some of my comments and feedback are below.

Major Comments:

Effect of mechanical preconditioning on fibrosis. Figure 2 shows how preconditioning seems to enable improved drug delivery. However, the trend seems to suggest that over time (Day 23 vs day 15) the effect is reduced. Two questions remain unanswered: 1) Is an initial mechanical preconditioning period sufficient to have long-term effects on FC even when stopped? If so, how long an initial period is necessary? 2) What happens if mechanical preconditioning continues to be applied?

What are the optimal loading parameters for reducing FBR? In the introduction the authors mention that previous studies exceeded the threshold for reducing FBR. This is an interesting aspect that is not fully addressed. What are the ideal forces, strain, or stresses to significantly reduce FBR, and how did this analysis shape the experiments performed here?

The manuscript fails to provide justification as to why 2psi, 1Hz, and 12 hours/day were chosen as preconditioning parameters. These parameters also appear to be distinct from authors' previous papers (Dolan et al. and Cezar et al.) which seemed to use 2psi at 1Hz for 5 minutes every 12 hours. Some additional experiments may be necessary here to at least show what the effect of varying these parameters may be.

The animal experiments showing insulin delivery through burst actuation are unclear. Were both groups given food? If so, why did the experimental group see their BG rise more than control. If not, why not?

The authors do not mention any effects of burst actuation on FC formation. The model proposed and the experimental data suggest that the FC is 'expanded' by the burst actuation – does this affect FC growth over time?

I commend the authors on developing a clinical context and implementation for their device. However, what is lacking is a convincing explanation of how preconditioning will be applied in a clinical context. Would a pump be worn by the patient or could it be implanted?

The authors do not include any limitations or drawbacks of the experiments here in the discussion.

Please add and include what future steps would be necessary for clinical translation.

Finally, the pairing of preconditioning and burst actuation strategies is unclear. These two strategies are presented as distinct by the authors, but in fact the same device could be used to achieve both. Is there an optimal FC-reduction strategy that combines both of them, and what is it? What is the ultimate framework or guidance that other groups seeking to implement these techniques in their implants should follow to best reduce FC?

Minor Comments:

Given the focus on insulin and reference to closed loop management of T1DM, it would be helpful to discuss FC formation impeding current chronic insulin pump functionality

In the introduction the authors mention that past work has not identified clear design rules for physical properties of implanted devices to modify FBR. This is not entirely true. For example, Veisheh et al. does make conclusions about shape and size. Also it is unclear if the authors in this paper provide any fundamental design rules either.

N values for various experiments unclear. Please clarify where plots show n greater than number of animals and specify how many values are taken per animal. For example, in figure 4c it is not clear how many values came from each animal, and how many animals were used in each group.

Reviewer #2 (Remarks to the Author):

This paper describes a soft, transport augmenting reservoir (STAR) that can modulate and adapt to the foreign body response, and can improve passive macromolecular transport. This advanced soft robotic anti-FBR strategy may attract researches in the field of implantable materials and drug delivery devices. This is a great work in terms of novelty and overall quality. I would recommend acceptance of this manuscript for publication in Nature Communications after necessary revision.

1. In the introduction part, the author should elaborate the differences and advantages of STAR compared to recent advanced milliscale dynamic soft reservoir (Sci. Robot. 2019. 4, eaax7043.) that was reported by the author.
2. Why STAR can modulate the FBR? It helps reader to understand the work better if the authors comment more on the mechanism?
3. Several important and representative advances relevant to the material strategies to resist the FBR is missing. Related references should be added, such as those advances reported in Nat. Commun. 2019, 10 (1), 5262; Nat. Commun. 2021, 12 (1), 5327; Angew. Chem. Int. Ed. 2020, 59 (24), 9586; Adv. Funct. Mater. 2019, 29, 1900140.
4. Why are the n values of several different groups different in some experiments such as in Fig. 3.
5. How were the number of CD31+ blood vessel and volume of cells stained with α -SMA counting and quantification performed? What identification criteria, counting area were used?

6. The examination of the inflammatory response in the early stage is important evaluation criteria to the FBR. I suggest that the author examine the inflammatory response after 7 days or 15 days of implantation.

Reviewer #3 (Remarks to the Author):

The authors describe their soft, transport augmenting reservoir (STAR) implant that builds upon recent evidence of reduction of FBR using a dynamic implant motion (see Ref 44, 45). STAR is an actuable, implantable drug delivery pouch that uses externally percutaneously applied pneumatic pressure to induce convective fluid flow to modulate the FBR, reduce fibrosis, and improve the transport of small- and macromolecular therapeutics.

Unique and impacting features of this report include:

1. They leverage a fibrosis-attenuating externally activated percutaneous oscillatory inflation/deflation implant preconditioning regimen to alter the foreign body capsule (FC), and improve passive macromolecular transport in rodent subcutaneous models.
2. They show a direct inverse relationship between STAR drug release and FC capsule thickness. Two discrete preconditioning deflection regimens (0.58 mm and 1.3 mm, respectively) are applied to affect local FC formation.
3. They further demonstrate that STAR can adapt to increasing capsule thicknesses at later timepoints by utilizing bursts of STAR inflatable membrane external percutaneous actuation to drive convective drug transport through the FC at the implant site in vivo.
4. Importantly, in both these drug transport-augmenting strategies, they claim to demonstrate functionality by delivering an active therapeutic (insulin) that elicits a known therapeutic effect clinically.
5. They demonstrate how such a device is feasibly implanted in human abdominal locations using a human cadaver surgical implant procedure.

Notably, STAR preconditioning-mediated differences observed in macromolecular transport through the FC are not attributed to changes in FC vascularity or FC collagen content or organization. They did not observe any within-group differences for FC thicknesses between the smaller and larger chamber wall implant in situ membrane deflections. Is the same mechanism also operable in human sub-muscular abdominal compartment as proposed for translation with the same membrane deflection amplitudes and timings?

The approach is validated by experimental design; the quality results/data follow logically from the experiments, and their study conclusions are reasonable from the reported results. While well-written with an important creative and impacting message to the medical device and implant community, the following considerations would improve the paper's clarity and delivery:

Introduction: please consider these clarifications/details:

1. Authors write: "heart valves⁷, pacemakers⁸, artificial joints⁹, orthopedic implants¹⁰, hernia meshes¹¹, and stents¹²." but CGMs represent a relevant additional implant class for the diabetes theme of this manuscript. (see FBR-relevant CGM implant issues in book: *Implantable Glucose Sensing*, eds. J. Stenken, D. Cunningham, Wiley, NY, USA, ISBN 978-0470112960, 2009.)

2. Authors write: “and delivery of anti-inflammatory drugs^{30,31} have been developed to modify the FBR”: many different drugs have been delivered locally than merely anti-inflammatories (anti-histamines, antimicrobials, steroids, angiogenic agents, anti-fibrotics (this is reviewed in Avula and Grainger, “Addressing medical device challenges with drug/device combinations,” Chapter 3, in Drug-Device Combinations for Chronic Diseases, R. Siegel., S.P. Lyu, eds., Wiley, New York, USA ISBN: 978-1-118-12000-2, 280 pp., 2015.)

3. Authors write: “and need to be replenished for long term efficacy^{31–33}.”: the DES taught the community that long-term on-going drug release wasn't necessary if the initial acute host response was effectively attenuated. Reloading and continual release may not be necessary for the effective approach for all implants. This depends on the specific pharmacology.

4. “Altering the biomechanics at the implant site”: local tissue-implant biomechanics issues described in these papers ought to be considered here to fully describe the in situ situation: Helton et al., J Diabetes Sci Technol 2011;5(3):647-656; Helton et al., J Diabetes Sci Technol 2011;5(3):632-646.

Results: clarifications requested here:

1. “by thickness measurements of the isolated FC under both the small and large chambers”: what is the specific physical location of the FC thickness measurements? STAR corners or edges or simply flat STAR surfaces? Corners and edges exacerbate FBR thickness.

2. “and the downstream clinical effects” and “elicit a clinical effect”: no clinical effects are shown since no human therapeutic treatments are shown. Please modify.

3. “used to improve distribution of chemotherapy in deep brain tumor targets 28–31”: though therapeutic outcome in this case is relatively modest, even perhaps clinically insignificant.

Methods: improved details are requested a specified here:

1. “The STAR device was then inserted”: dimensions of this implanted device?

2. “controlled input pressure of 2 psi at 1 Hz every 12 hours for 14 days”: what STAR wall deflection amplitude does this create in the STAR device wall against local tissue?

3. “pneumatically activated using a pressure input of 2 psi.”: duration of this activation? constant isobaric stimulus, no cycling?

4. “and actuation catheter of each STAR device.”: dimensions of this implanted device?

5. “Genhance 750”: chemical description of this substance should be briefly included.

Limitations: Some expert comments on limitations of the the SD Rat and C57BL6 mice STAR dorsal pocket implant fibrosis models reported in predicting actual human FBR and STAR implant fibrotic responses in the different anatomical location (abdominal sub-muscular/intermuscular) would help balance the discussion.

Weakness: Mechanism for how the FC around the implant is modulated by exogenously induced wall deflections is missing. This limitation is mentioned but remains to be addressed. As such then, the reporting is empirical and observational, producing intriguing effects but without a rational explanation. Phenomenologically, the cause-effect description is intriguing as described.

Typos to revise:

“This potential space it is well”

“Devices was pneumatically”

“baseline. 300 µl of the”: shouldn't start a sentence with a numeral.

“heated pad. 3 mL of warm”: shouldn't start a sentence with a numeral.

“Data is represented”

Response to Reviewers' Comments

Summary of Revision for All Reviewers

We would like to thank the reviewers for their insightful and detailed comments as the referee report was very helpful and led to a significant improvement of the work.

The reviewers' questions pushed us to build on our foundational work and more comprehensively study the effect of our actuation strategies on drug delivery and the underlying mechanism of action. To do so, we have conducted the following new experiments and analyses since the initial submission, with a number of new findings:

(1) An 8-week pre-clinical study longitudinally examining the effect of intermittent actuation on macromolecular delivery (new Figure 3), which now demonstrates that actuation over 8 weeks is able to preserve the therapeutic function of our implantable device at levels comparable to those seen immediately following implantation.

(2) An accompanying assessment of FC formation at longitudinal timepoints over 8 weeks (new Figure 4). This was conducted to better understand the multiphasic cellular changes and key drivers of enhanced drug transport caused by intermittent actuation. We now include a longitudinal analysis of neutrophils, myofibroblasts, total cell number, capsule thickness, capsule vascularity, capsule density, collagen maturity, and collagen orientation.

(3) An evaluation of the effect of device membrane deflection on underlying tissue strain using a computational finite element model (new Figure 1f, and Supplementary Figure 3).

We have also altered some of the terminology and conventions from the previous submission to be more descriptive and easier to understand for the reader:

- The term mechanical preconditioning has been replaced with the term intermittent actuation (IA)
- The term burst actuation has been replaced with the term actuation-mediated rapid release (RR)
- Blood glucose curves are now expressed in terms of change in % blood glucose where a drop in blood glucose is represented by a negative number. Therefore, higher values of area under the curve (AUC) now represent greater therapeutic effect.

To facilitate reassessment of the manuscript, we have provided a point-by-point response to all the reviewers' comments in the following pages. **Our responses are in green text, and the resultant changes in the manuscript noted in blue text.**

Reviewer #1:

Summary Comment: Thank you for the opportunity to review this excellent work. The authors describe STAR, an implantable system that incorporates mechanotherapeutic methods to modify and overcome the foreign body response which forms around implantable devices. The problem of FBR is quite significant and an implantable device that could successfully overcome this would be remarkable and quite impactful. The authors elegantly describe experiments that seek to answer mechanistic questions about how mechanical modulation might impact FBR and the capacity for drug delivery from the implant. However, the experiments fall short of answering a few fundamental questions that this reviewer believes are vital to enable the impact of these experiments to be fully realized. Some of my comments and feedback are below.

Response: We thank the reviewer for their review. As mentioned above, these comments have pushed us to significantly expand on our foundational work and ask important questions using our developed methodology.

Major Comments:

R1C1: Effect of mechanical preconditioning on fibrosis. Figure 2 shows how preconditioning seems to enable improved drug delivery. However, the trend seems to suggest that over time (Day 23 vs day 15) the effect is reduced. Two questions remain unanswered: 1) Is an initial mechanical preconditioning period sufficient to have long-term effects on FC even when stopped? If so, how long an initial period is necessary? 2) What happens if mechanical preconditioning continues to be applied?

Response: To address these questions, we designed and conducted an 8-week long-term implantation animal study, where we longitudinally assessed insulin transport across the formed FC and its ability to elicit a change in blood glucose response.

The study groups were as follows:

- (a) 3 weeks of intermittent actuation and then stop actuation until week 8: 3W IA
- (b) 8 weeks of intermittent actuation: 8W IA
- (c) Static control for 8 weeks: Control

Diffusion-based transport of insulin was assessed via serial blood glucose measurements at 3 days and 2, 3, 4, 5 and 8 weeks.

Our main finding from this study is the ability of intermittent actuation (when applied for the entire study duration) to maintain drug transport at baseline levels and extend the therapeutic lifespan of our device. In sharp contrast, the insulin responsiveness of the static control group decreased with longer implantation time, until near complete isolation by the FBR, and resultant implant failure (Figure 3). In addition, longitudinal capsular analysis revealed the temporal pleiotropic role of intermittent actuation in combating the foreign body response and thus improving transport. This included modulation of distinct cell populations (neutrophils, myofibroblasts) along with structural changes (thickness, fibre alignment) and degree of cellular infiltration (Figure 4).

Our results suggest that an initial period of actuation (3 weeks) can also mediate positive long-term benefits in therapy transport. Stopping actuation at 3 weeks in the 3W IA group led to a worsening of insulin response, with an AUC progression that paralleled the control group (Figure 3g). Though there appears to be a trend towards better functional effect in the 3W IA compared to control at the later time points, this was not statistically significant. To support this, we demonstrated that intermittent actuation causes a localized immunomodulatory effect through clearance of neutrophils at the pericapsular site at an early timepoint (5 days after implantation; Figure 4). Neutrophil infiltration following implantation is an important first step in the initiation and propagation of the inflammatory process, and the subsequent recruitment of cell populations known to develop the FBR. Early resolution of the neutrophil response may have important long-term consequences on the FBR.

In summary, an initial period of mechanical conditioning is not sufficient to have long-term effects on STAR function after actuation is stopped. However, if actuation is continued, there is a dramatic benefit in terms of therapeutic lifespan of the device. Overall, the results suggest that continued mechanical actuation will be needed to maximize long term beneficial effects on transport (Figure 3f,g).

The complete study data have been added to the manuscript as new Figures 3 and 4. We have reproduced the relevant text and figures below.

Results (page 12, line 14 – page 19, line 9)

Dynamic intermittent actuation (IA) extends therapeutic lifespan of STAR. Following device and *in vivo* model development, we designed an 8-week longitudinal preclinical study to test the ability of STAR to modulate the FBR and improve macromolecular delivery across the formed FC.

We implanted STAR devices on the dorsal subcutaneous aspect of 3 groups of mice (Figure 3a). In two experimental groups, we performed STAR-enabled IA with cyclic pressure input of 2 psi at 1 Hz for 5 min every 12 hours using a custom-made pneumatic control system (Supplementary Figure 6). One group (8W IA) was intermittently actuated for the total study duration of 8 weeks, while the second group (3W IA) received 3 weeks of IA followed by no actuation for the remainder of the study. A third group which did not receive IA served as the control. We injected short-acting human insulin (2 IU/kg) into the device at various time points post-implantation: 2, 3, 4, 5, and 8 weeks as well as day 3, which served as a baseline (BL). We monitored diffusion-based transport across the formed FC and into the bloodstream *via* serial blood glucose measurements at these timepoints.

At baseline, insulin administration produced a similar drop in blood glucose in both IA and control groups (Figure 3b, Supplementary Note 1). Over 8 weeks, the blood glucose curves separate, with decreased insulin responsiveness in the control and 3W IA groups, compared to the 8W IA group. Impressively, the 8W IA group maintained its rapid drop in blood glucose over the entire study duration (Figure 3b). Despite chronic implantation and FC development, there was no statistical difference in the maximum blood glucose drop at 3 days (BL; $72.5 \pm 2.2\%$) and 8 weeks ($68.3 \pm 3.4\%$) post-implantation in the 8W IA group (Figure 3c, Supplementary Note 1). In contrast, the control group achieved only a $20.9 \pm 4.3\%$ maximum drop in blood glucose at the 8-week timepoint, reflecting nearly complete loss of drug delivery functionality due to implant isolation by the FC. The 3W IA group had a similar loss of function, with a maximum blood

glucose drop that was not significantly better than control at the end of the study (see Supplementary Note 1 for details).

The mean time to achieve a physiologically relevant response (30% drop in blood glucose) was preserved at less than 30 min in the 8W IA group over the entire study duration (Figure 3d, Supplementary Note 1), with all mice achieving this 30% drop within 48 min at 8 weeks (Figure 3e). In contrast, the mean time to effect for the control and 3W IA groups progressively increased with implantation time. The mean time to therapeutic effect increased to > 65 min in the 3W IA group (with 2 out of 5 mice not responding) by week 8, and was not detectable within the 120 min experimental timeframe in the control group (Figure 3d,e, Supplementary Note 1). Notably, 8W IA devices were able to achieve a therapeutic blood glucose drop twice as fast as control devices at 4 weeks (mean time to 30% drop: 27.43 ± 4.48 min vs 73.55 ± 14.85 min) and four times as fast at 8 weeks (26.33 ± 6.16 min vs >120 min).

When time and magnitude were integrated by calculating AUC (Supplementary Figure 5a), 8W IA produced a robust treatment effect with significant benefits in drug delivery at all time points in comparison to the control group (Figure 3f, Supplementary Note 1). Stopping actuation at 3 weeks in the 3W IA group led to a worsening of insulin response, with an AUC progression that paralleled the control group (Figure 3g). Though there appears to be a trend towards better functional effect in the 3W IA compared to control at the later time points, this was not statistically significant, which implies that continued mechanical dosing will be needed for robust long-term beneficial effects on transport (Figure 3f,g). Overall, this data suggests that IA can mitigate the FBR and extend the therapeutic lifespan of implantable drug delivery devices.

Fig. 3 | Intermittent actuation (IA) improves long-term macromolecule delivery. **a**, Preclinical study timeline used to evaluate the effect of IA on insulin transport through a fibrous capsule. **b**, Blood glucose (BG) response to human insulin, measured over 120 min at day 3 (baseline), week 3 and week 8. **c**, Maximum BG% change at the 8-week timepoint. **d**, Time to achieve a 30% drop in BG level, measured longitudinally. **e**, Cumulative incidence curves demonstrating probability of achieving a 30% BG drop over 120 min for all groups at both baseline (3 days) and 8-week

timepoints. **f**, Area under BG % curve (AUC) denoting overall functional effect mapped over study duration. Statistical comparisons w.r.t. control group. **g**, Change in AUC from 3-week timepoint. Data are means \pm standard error of mean; * $p < 0.05$, ** $p < 0.01$, *** $p < 0.001$. See Supplementary Note 1 for detailed statistical analyses. † Baseline study was performed post-hoc with separate mice. # Control mice removed from study at intermediate time points due to self-inflicted device damage with subsequent decrease in n .

Multiphasic temporal effects of intermittent actuation. Following the completion of our pre-clinical study, we next set out to analyse differences in FC composition at evolving timepoints to better understand the multiphasic cellular changes and key drivers of enhanced drug transport caused by IA (Figure 4a).

First, we investigated the initial, acute phase of the inflammatory FBR. Using immunofluorescent Ly-6G+ staining, we examined the pericapsular region for the presence of neutrophils, the first responders of the immune defence. We found that IA significantly reduces the presence of neutrophils at day 5 in comparison to the control (Figure 4b,c). This result indicates that the application of IA can mediate a localised immunomodulatory effect.

We next assessed the activation of matrix producing cells into a myofibroblast phenotype, a key contractile cell in fibrosis progression. The IA group exhibited a significant reduction in α SMA expression when compared to the control at 2 weeks (Figure 4d,e). Despite observing differences in individual cell populations (neutrophils, myofibroblasts), we did not detect differences in overall cell number at equivalent timepoints (Figure 4f,g).

Next, we investigated the macroscale capsular changes responsible for improved therapy transport. We examined evolving capsule thickness with longer periods of STAR implantation. IA mitigated capsule growth in the first 2 weeks following implantation, with a significant reduction in thickness observed w.r.t. the control at 2 weeks (Figure 4h,i). This result aligns with the enhanced blood glucose responsiveness of the IA group at early timepoints (Figure 3) and suggests that FC thickness is an important contributor to initial improvements in macromolecular transport.

By 8 weeks, however, capsule thickness had equalized between the IA and control groups (Figure 4i). This result suggests that additional mechanisms are responsible for the sustained improvement in functional response to insulin in the IA group at later timepoints (Figure 3). To investigate this further, we examined capsule vascularity, density, and the maturity of the collagen fibres. However, we did not observe differences that would account for improvements in macromolecular transport and functional effect (Supplementary Figure 7, Supplementary Note 1). By optical coherence analysis of polarised light microscopy images, we found that collagen fibres exhibited higher alignment in the IA group compared to control at week 8 (Figure 4j,k). IA appeared to increase collagen alignment over time from 2 weeks to 8 weeks, whereas there was no temporal change in alignment observed in the control group (Figure 4k). We posit that the lower degree of alignment, and therefore greater degree of fibre entanglement, in the control group creates steric hindrance which potentially slows or immobilizes transport of macromolecules through the collagenous matrix^{1,2}.

Finally, we examined the ability of IA to protect against cellular invasion and blockage of the porous membrane of STAR. Scanning electron microscopy demonstrated clear differences in cellular infiltration between the control and IA group at the 8-week timepoint (Figure 4l). This effect could be attributed to convective flow generated by STAR upon actuation (Supplementary Figure 3g). These capsular analyses reveal the pleiotropic role of IA in modulating the FBR and highlight cellular and structural changes that lead to improved transport of macromolecular therapy.

Fig. 4 | Multiphasic temporal effects of intermittent actuation (IA). **a**, Timeline of multiphasic cellular and fibrous capsule (FC) changes induced by IA. **b**, Representative fluorescent images of the FC stained with Ly-6G+ (green) and DAPI (blue). Scale bars are 20 μm. **c**, Quantification of neutrophils present within FC +/- IA at day 3 and 5. **d**, Representative fluorescent images of the FC stained with α-SMA (green) and CD31 (red). Scale bars are 50 μm. **e**, Quantification of myofibroblasts present within FC +/- IA at 2 weeks. **f**, Representative histologic images of the FC

stained with haematoxylin and eosin. Scale bars are 20 μm . **g**, Quantification of total cells/capsular area +/- IA at day 3, day 5, and 2 weeks. **h**, Representative topographical reconstructions of μCT images showing the differences in FC thickness +/- IA at 2 weeks. **i**, Average FC thickness of the control and actuated groups at day 3, day 5, 2 weeks, and 8 weeks with two measurements taken per animal. **j**, Representative polarised light microscopy images of the FC obtained after picosirius red staining at 8 weeks. Scale bars are 100 μm . **k**, Quantification of the FC collagen fibre orientation by optical coherency with 60 ROIs per animal. **l**, Representative SEM images demonstrating reduced cellular invasion with actuation at the 8-week timepoint. Scale bars are 500 μm . $n = 2-6$ animals per group; data are means \pm standard error of mean; * $p < 0.05$, ** $p < 0.01$, *** $p < 0.001$, **** $p < 0.0001$.

Discussion (page 24, line 15 – page 26, line 22)

IA is the first element in STAR's armamentarium. By inducing strain at the tissue-contacting membrane and perturbing peri-device fluid flow, STAR acts as an oscillating mechanical shield against the invading cellular FBR. These localised mechanical effects create a favourable environment for the long-term transport of macromolecules. IA is able to preserve the functional effect of STAR over 8 weeks at the same level as that seen immediately following implantation, i.e., prior to formation of a significant inhibitory FC (Figure 3c). In sharp contrast, the insulin responsiveness of the control group decreased with longer implantation time, until near complete FC isolation and implant failure. The extension of therapeutic effect using a simple 5 min, twice daily actuation regimen represents an attractive and innovative strategy for mitigating the FBR.

By combining ITT blood glucose results with a range of *ex vivo* capsular analytical techniques, we were able to unravel the multiphasic, temporal effects of IA on cell infiltration and capsule formation. When examining the inflammatory response in the IA and control groups, we detected significant differences in distinct cell populations at times of expected peak infiltration, despite no difference in total pericapsular cell number (Figure 4).

We demonstrated that IA produces a localised immunomodulatory effect by clearing neutrophils from the pericapsular site (Figure 4b,c). The infiltration of neutrophils is an important first step in the FBR, initiating and propagating the inflammatory process, which causes the subsequent recruitment of cell populations known to develop the FC (e.g., macrophages). Early modulation of the neutrophil response may have important long-term consequences on the FBR. Indeed, we observed a trend towards better functional effect in the 3W IA group compared to control even after stopping actuation. Though this effect did not reach statistical significance, there may be some long-term benefits in therapy transport even with initial periods of actuation; however, it is clear that maximal benefit comes with continued actuation (Figure 3f). A recent study by Seo et al.³ corroborates the mechanosensitive nature of neutrophil cell populations following dynamic loading for skeletal muscle regeneration. The authors postulated that mechanical flushing of chemoattractants was responsible for the reported decrease in neutrophils³. In this context, our work motivates further study on the effect of IA on pro-inflammatory chemoattractant gradients such as interleukin 1, 6 and 8 versus direct mechanical effects on cell attachment, orientation, and function.

We noted a significant decrease in myofibroblast cell number with IA compared to control (Figure 4d,e). The activation of matrix-producing cells into a myofibroblast phenotype, characterised by αSMA expression, is a critical step in fibrosis progression. Increased expression leads to heightened contractile activity, formation of stress fibres, and synthesis of extra-cellular

matrix. Furthermore, activated fibrogenic cells can produce cytokines responsible for additional cell recruitment and propagation of the deleterious fibrotic response⁴. It is becoming increasingly evident that stiffness precedes, or is an important contributor to fibrosis⁵. Thus, reducing myofibroblast expression and its effect on matrix stiffness may be a key strategy to modifying this self-perpetuating fibrotic effect.

In addition to multiphasic cellular changes with time, we also observed multiphasic differences in macro-capsular architectural evolution, with distinct differences between the IA and control groups. At early timepoints (2 weeks) following implantation, we observed differences in capsule thickness between groups (Figure 4h,i), which aligns well with improved diffusion-based transport at these timepoints (Figure 3b,f). FC thickness equalizes between groups after 8 weeks of implantation, indicating that other mechanisms are responsible for the improved transport at later timepoints. We rejected several relevant hypotheses for improved transport in the IA group including vascularity, capsule density, and collagen maturity (Supplementary Figure 7). However, we noted differences in collagen architecture and cellular infiltration into the device reservoir which may contribute to late differences in transport and functional effects between the groups (Figure 4j-l). Further study will be necessary to fully elucidate and understand the multiple mechanisms at play. Note that 2 out of 6 mice in the control group had to be withdrawn from the study at later time points due to self-inflicted damage to their dorsal tissue and subcutaneous implant. Interestingly, this did not occur in any IA group mice. Capsular contracture mediated by excessive FBR is a painful condition⁶ that could rationalize this group difference, and future work could investigate this observation further.

R1C2: What are the optimal loading parameters for reducing FBR? In the introduction the authors mention that previous studies exceeded the threshold for reducing FBR. This is an interesting aspect that is not fully addressed. What are the ideal forces, strain, or stresses to significantly reduce FBR, and how did this analysis shape the experiments performed here?

Response: We thank the reviewer for highlighting this point.

Historically, mechanical loading has been demonstrated to propagate the FBR⁷⁻¹². However, the literature also demonstrates that loading can be anti-inflammatory in some contexts, and thus potentially beneficial in mitigating the foreign body response. For example, low magnitude, dynamic loading has demonstrated anti-inflammatory effects on tendon fibroblasts¹³. Furthermore, low magnitude loading of injured skeletal muscle has been recently shown to modulate the acute inflammatory response, reduce fibrosis, and aid skeletal muscle regeneration³. Importantly, both studies indicated the presence of a therapeutic threshold, beyond which tissue damage and inflammation occurs.

With reference to these studies, we propose that an ideal strategy for mitigating the invading FBR creates dynamic convective flow and atraumatic tissue strain. Thus, we optimized STAR device design and pressure parameters using finite element computation modelling so that actuation-induced tissue strain did not exceed a reported threshold of 40%³ known to cause damage and inflammation in a mouse model (Figure 1f and Supplementary Figure 3e,f).

We have made the following changes to the manuscript to address the reviewer's comment.

1. We have further expanded the introduction to better elucidate the ideal loading parameters for reducing the FBR:

Introduction (page 5, line 5 – 17)

Dynamically altering the local biomechanical environment at the implant site is one such promising, yet underexplored, drug-free approach¹⁴. Cells in our body are exquisitely sensitive to their mechanical environment, with loading playing a pivotal role in cell functions such as differentiation¹⁵, proliferation¹⁶, and migration¹⁷. Historically, studies have observed biomechanical stress as a pro-fibrotic or regenerative stimulus¹⁸, demonstrating that application of stretch⁷⁻⁹, fluid flow^{10,11}, or compression¹² to cells can lead to increased deposition of collagenous matrix. Accordingly, many anti-FBR strategies have focused on minimizing the mechanical mismatch, interfacial stress, and movement between the implant and local tissue. Our research seeks to challenge this status quo and reveals the potential for a dynamic mechanotherapeutic that uses low-magnitude, atraumatic, tissue strain and convective flow as a defence mechanism against the invading cellular FBR. Interestingly, some studies have observed that small magnitude, dynamic loading has anti-inflammatory effects, and indicated the presence of a therapeutic threshold, beyond which tissue damage and inflammation occurs^{3,13}.

2. We performed finite element computational modelling and have added data elucidating the effect of membrane deflection on the underlying tissue strain to Supplementary Figure 3e,f.

Results (page 8, line 1 – 7)

As part of device design and optimisation, we performed finite element (FE) simulations to understand the biomechanical changes mediated by actuation, particularly the relationships between membrane deflection, convective flow, and tissue strain (Figure 1e,f; Supplementary Figure 3). Based on recently reported results³, we designed our soft robotic actuation strategy to induce tissue strain that would fall within the atraumatic range (<40%), and hypothesise that this regimen would mitigate the FBR by creating convective flow disruptive to the cellular immune response.

Supplementary Fig. 3e,f | Membrane deflection in STAR. e, Finite element (FE) model showing tissue deflection caused by STAR membrane at 1, 2, and 3 psi pressure input. f, FE model showing tissue strain caused by STAR membrane at 1, 2, and 3 psi pressure input.

R1C3: The manuscript fails to provide justification as to why 2psi, 1Hz, and 12 hours/day were chosen as preconditioning parameters. These parameters also appear to be distinct from authors' previous papers (Dolan et al. and Cezar et al.) which seemed to use 2psi at 1Hz for 5 minutes every 12 hours. Some additional experiments may be necessary here to at least show what the effect of varying these parameters may be.

Response: We thank the reviewer for pointing out this potential source of confusion. The actuation parameters of 2psi, 1 Hz for 5 minutes every 12 hours (rather than 12 hours per day) were used in this study, similar to the above-mentioned studies. We have amended the text to make this clearer.

Results (page 12, lines 19 – 21)

In two experimental groups, we performed STAR-enabled IA with cyclic pressure input of 2 psi at 1 Hz for 5 min every 12 hours using a custom-made pneumatic control system (Supplementary Figure 6).

Methods (page 36, lines 5 – 7)

The device was then cyclically actuated at a controlled input pressure of 2 psi at 1 Hz for 5 min every 12 hours as previously described^{14,19}.

Furthermore, as discussed in comment R1C2 and Supplementary Figure 3 above, we have added computational modelling data exploring the effect of varying pressure input on device deflection and tissue strain. A 2 psi input elicits tissue strain within the range of a previously reported immunomodulatory mechanotherapeutic (10-40%)³ and does not exceed the threshold for causing inflammation and damage in a mouse model.

R1C4: The animal experiments showing insulin delivery through burst actuation are unclear. Were both groups given food? If so, why did the experimental group see their BG rise more than control. If not, why not?

Response: We thank the reviewer for this comment. To clarify, only the group subjected to actuation-mediated rapid release (RR; experimental group) received food at 120 minutes, causing their BG to rise more than the control. We gave food for animal welfare reasons, to prevent potentially severe hypoglycaemia in response to actuation-mediated rapid release of additional insulin.

We have made the following changes to add clarity:

Results (page 20, lines 3 – 9)

In a final example, we demonstrated enhanced mass transport and downstream functional effect using RR in our ITT model (Figure 5i). Insulin was delivered through STAR devices 2 weeks after implantation in mice. Passive diffusion of insulin led to a drop in blood glucose over 120 min in all animals. At this point, food was given to one group to allow recovery of blood glucose levels towards baseline. At 150 min, this group was subjected to actuation-mediated RR, causing a

significant reduction in blood glucose levels over 15 min due to rapid release of insulin (Figure 5i).

R1C5: The authors do not mention any effects of burst actuation on FC formation. The model proposed and the experimental data suggest that the FC is ‘expanded’ by the burst actuation – does this affect FC growth over time?

Response: We thank the reviewer for this comment.

In hindsight, ‘burst actuation’ could potentially be a misleading term because it could imply that this type of actuation damages the physical structure of STAR or the FC, which was not our intention. We have changed all instances of the term ‘burst actuation’ to ‘actuation-mediated rapid release (RR)’ to better convey the intended message. We have also removed the confusing schematic in old Figure 5c.

As seen in Supplementary Video 2, the device returns to its pre-inflation state after actuation-mediated RR. The magnitude of actuation is identical (2 psi at 1 Hz) for RR and IA regimens. There is a significant difference in the number of cycles (1-5 cycles for RR, 300 cycles twice daily for IA).

We envision that this RR strategy will be used only when rapid on-demand drug delivery is required, therefore we do not anticipate that RR will have a long-term effect on fibrous capsule formation or expansion.

We have added clarification to the manuscript:

Results (page 19, lines 10 – 15)

On demand, actuation-mediated rapid release of drug using STAR. In addition to intermittent immunomodulatory actuation, we demonstrate another actuation-based mechanism of augmenting drug transport. Actuation-mediated rapid release (RR) of a drug loaded STAR device consists of a few (~1–5) cycles of on-demand actuation at the same magnitude as IA (2 psi, 1 Hz). This strategy can accelerate mass transport of drug from the device reservoir (Figure 5a, Supplementary Video 2) into surrounding tissue (Figure 5b, Supplementary Video 3).

R1C6: I commend the authors on developing a clinical context and implementation for their device. However, what is lacking is a convincing explanation of how preconditioning will be applied in a clinical context. Would a pump be worn by the patient, or could it be implanted?

Response: We thank the reviewer for raising an important question. We have amended our manuscript to include a more thorough discussion of the clinical use case, including powering considerations. In the envisioned use cases, STAR or its embodiments is connected to an external wearable pump similar to those being used in existing insulin pumps. With current technology, we favour this approach over a fully implanted enclosed pump as this minimizes the invasiveness of delivery, maximizes patient comfort, and avoids issues with battery or pump failure necessitating

corrective surgery. However, with future developments in device and control system miniaturization, we do envision a future embodiment to be fully implantable.

Discussion (page 27, line 9 – page 28, line 20)

We envision several clinical use scenarios for IA and RR technologies demonstrated in this work. RR could be used independently of IA, such as for the rapid, on-demand delivery of therapeutics in response to a clinical emergency. Some relevant examples include the delivery of adrenaline for the treatment of anaphylaxis or glucagon for the treatment of hypoglycaemic coma. In both cases, impedance of delivery due to FC formation would have grave consequences.

An optimal FC mitigating strategy could also combine both IA and RR. For example, short daily bursts of IA (without drug) could attenuate the FBR to extend device lifespan and improve performance in long-term implantation. A temporally controlled and modular RR actuation regime could then be used to make rapid, precise dosage adjustments in accordance with the patient-specific clinical scenario and/or FBR severity. IA and RR are complementary strategies that elegantly make use of a single device design and pump, making STAR well suited to a variety of clinical scenarios.

The management of type 1 diabetes is one relevant clinical area where STAR could provide synergistic benefit. For example, IA could be applied to extend the lifespan of an artificial pancreas²⁰, preventing unnecessary FBR-mediated blockages, linked hyperglycaemic events, and ultimately simplifying the dosing regimen and patient experience. In synergy, actuation-mediated RR could make rapid insulin adjustments and maintain blood glucose levels in the narrow window necessary to prevent long-term complications²¹. Looking further into the future, application of STAR could enable translation of next-generation bioartificial technologies utilizing human-derived insulin-producing islet cells by modifying the transport-limiting FC, which has been a major barrier to the viability of cell-based therapeutics^{22–25}. Considering the need for daily IA for long-term efficacy, it is likely that embodiments will require a wearable pump, similar to those used in existing insulin pumps²⁶.

STAR's soft material endows it with biocompatibility advantages over rigid implantable drug delivery systems²⁷, and lends itself to minimally invasive catheter implantation. In a step towards clinical translation, we scaled-up the STAR device and developed a bespoke delivery tool as well as a minimally invasive procedure that is congruent with conventional interventional radiology techniques (Figure 6). We demonstrated delivery of STAR to a clinically accessible intermuscular space in the anterior abdominal wall in a human cadaver model. This approach allowed for a short procedure time (<20 min) for implantation of a human scale device through a modified sheath using ultrasound guidance in the hands of an experienced interventional radiologist. Additional design features demonstrated correct deployment and adhesive delivery to maintain device position in the tissue plane. Thus, STAR can be quickly implanted by interventionalists in an outpatient setting under local anaesthesia using an established imaging modality.

R1C7: The authors do not include any limitations or drawbacks of the experiments here in the discussion. Please add and include what future steps would be necessary for clinical translation.

Response: We thank the reviewer for this important comment. We have included a more thorough analysis of our results, including potential future directions and questions that remain unanswered. Furthermore, we have added a limitations paragraph, discussing differential anatomic and

microenvironmental factors in animal models that may be barriers to clinical translation. The relevant sections are copied below in red.

Discussion (page 25, lines 6 – 20)

We demonstrated that IA produces a localised immunomodulatory effect by clearing neutrophils from the pericapsular site (Figure 4b,c). The infiltration of neutrophils is an important first step in the FBR, initiating and propagating the inflammatory process, which causes the subsequent recruitment of cell populations known to develop the FC (e.g., macrophages). Early modulation of the neutrophil response may have important long-term consequences on the FBR. Indeed, we observed a trend towards better functional effect in the 3W IA group compared to control even after stopping actuation. Though this effect did not reach statistical significance, there may be some long-term benefits in therapy transport even with initial periods of actuation; however, it is clear that maximal benefit comes with continued actuation (Figure 3f). A recent study by Seo et al.³ corroborates the mechanosensitive nature of neutrophil cell populations following dynamic loading for skeletal muscle regeneration. The authors postulated that mechanical flushing of chemoattractants was responsible for the reported decrease in neutrophils³. In this context, our work motivates further study on the effect of IA on pro-inflammatory chemoattractant gradients such as interleukin 1, 6 and 8 versus direct mechanical effects on cell attachment, orientation, and function.

Discussion (page 26, lines 7 – 22)

In addition to multiphasic cellular changes with time, we also observed multiphasic differences in macro-capsular architectural evolution, with distinct differences between the IA and control groups. At early timepoints (2 weeks) following implantation, we observed differences in capsule thickness between groups (Figure 4h,i), which aligns well with improved diffusion-based transport at these timepoints (Figure 3b,f). FC thickness equalizes between groups after 8 weeks of implantation, indicating that other mechanisms are responsible for the improved transport at later timepoints. We rejected several relevant hypotheses for improved transport in the IA group including vascularity, capsule density, and collagen maturity (Supplementary Figure 7). However, we noted differences in collagen architecture and cellular infiltration into the device reservoir which may contribute to late differences in transport and functional effects between the groups (Figure 4j-l). Further study will be necessary to fully elucidate and understand the multiple mechanisms at play. Note that 2 out of 6 mice in the control group had to be withdrawn from the study at later time points due to self-inflicted damage to their dorsal tissue and subcutaneous implant. Interestingly, this did not occur in any IA group mice. Capsular contracture mediated by excessive FBR is a painful condition⁶ that could rationalize this group difference, and future work could investigate this observation further.

Discussion (page 28, line 21 – page 29, line 10)

Though we have demonstrated robust preclinical results in a long-term mouse model, there are several limitations and barriers to clinical translation. Our findings from implantation in the dorsal subcutaneous space of mice may not directly predict similar results in humans at different anatomical locations (e.g., abdominal intermuscular space), and further work is needed to understand how these anatomic and microenvironmental differences impact the effects of STAR. Although rodent models have been extensively used to study the FBR^{28,29}, rodents have been shown to have different tissue collagen content in the subcutaneous space surrounding an implant

and different metabolites in the interstitial fluid at implant interfaces, as compared to humans³⁰. Moreover, differences in rodent skin, fur, and behaviours may subject implanted devices to different biomechanical forces than in humans³¹. Encouragingly, thus far, we have observed similar FBR mitigating effects with an IA regimen that is agnostic of species and device design¹⁴. In addition to this, the presence of conserved inflammatory pathways in FC formation across species³² suggest that STAR may have a similar benefit in prolonging implant lifespan in humans.

R1C8: Finally, the pairing of preconditioning and burst actuation strategies is unclear. These two strategies are presented as distinct by the authors, but in fact the same device could be used to achieve both. Is there an optimal FC-reduction strategy that combines both of them, and what is it? What is the ultimate framework or guidance that other groups seeking to implement these techniques in their implants should follow to best reduce FC?

Response: We thank the reviewer for this comment. We have amended our discussion to present a framework and specific clinical examples for both individual and synergistic use of STAR's actuation technologies. The appropriate amended text is reproduced below.

Discussion (page 27, line 9 – page 28, line 20)

We envision several clinical use scenarios for IA and RR technologies demonstrated in this work. RR could be used independently of IA, such as for the rapid, on-demand delivery of therapeutics in response to a clinical emergency. Some relevant examples include the delivery of adrenaline for the treatment of anaphylaxis or glucagon for the treatment of hypoglycaemic coma. In both cases, impedance of delivery due to FC formation would have grave consequences.

An optimal FC mitigating strategy could also combine both IA and RR. For example, short daily bursts of IA (without drug) could attenuate the FBR to extend device lifespan and improve performance in long-term implantation. A temporally controlled and modular RR actuation regime could then be used to make rapid, precise dosage adjustments in accordance with the patient-specific clinical scenario and/or FBR severity. IA and RR are complementary strategies that elegantly make use of a single device design and pump, making STAR well suited to a variety of clinical scenarios.

The management of type 1 diabetes is one relevant clinical area where STAR could provide synergistic benefit. For example, IA could be applied to extend the lifespan of an artificial pancreas²⁰, preventing unnecessary FBR-mediated blockages, linked hyperglycaemic events, and ultimately simplifying the dosing regimen and patient experience. In synergy, actuation-mediated RR could make rapid insulin adjustments and maintain blood glucose levels in the narrow window necessary to prevent long-term complications²¹. Looking further into the future, application of STAR could enable translation of next-generation bioartificial technologies utilizing human-derived insulin-producing islet cells by modifying the transport-limiting FC, which has been a major barrier to the viability of cell-based therapeutics²²⁻²⁵. Considering the need for daily IA for long-term efficacy, it is likely that embodiments will require a wearable pump, similar to those used in existing insulin pumps²⁶.

Minor Comments:

R1C9: Given the focus on insulin and reference to closed loop management of T1DM, it would be helpful to discuss FC formation impeding current chronic insulin pump functionality.

Response: We thank the reviewer for this suggestion. We have modified the introduction to address the impact of FC formation on the current functionality of implanted insulin pumps.

Introduction (page 3 line 15 – page 4, line 8)

One pertinent example is the management of type 1 diabetes, a chronic disease affecting 18 million people worldwide, with an annual economic burden of greater than \$90 billion USD³³. Successful implementation and clinical adoption of an artificial pancreas combining continuous glucose monitoring with the rapid, responsive, release of insulin (or glucagon) would vastly improve outcomes and quality of life for this patient population. The development of a fully automated closed loop insulin delivery system would reduce user burden, remove the need for multiple daily injections, and increase time spent in the optimal blood glucose range, which is imperative for the prevention of long-term diabetic complications. Unfortunately, current efforts at developing such a device have been hindered by the dynamic and unpredictable FBR, leading to glucose sensing inaccuracy, inhibition of insulin release, and gradual loss of functionality in the weeks to months following implantation^{34–37}. Looking towards the future, living implants containing stem cell derived pancreatic β -cells represent a potential cure for diabetes. However, the attenuation of oxygen and molecular transport due to the FC barrier still constitutes a major hurdle to successful clinical translation of these implants^{25,29,38,39}. It is evident that a method to (i) mitigate the FBR or (ii) improve transport across the FC could transform the management of this pervasive disease. Furthermore, such a method could have broader implications for a range of diseases and device-based treatments affected by the FBR.

R1C10: In the introduction the authors mention that past work has not identified clear design rules for physical properties of implanted devices to modify FBR. This is not entirely true. For example, Veisheh et al. does make conclusions about shape and size. Also, it is unclear if the authors in this paper provide any fundamental design rules either.

Response: We agree completely with this summation. We have modified the introduction accordingly to include previously reported design rules. We acknowledge that providing fundamental material design rules was not a primary goal of our study, however we hope that the insights presented in our revised submission will be informative and instructive to the field.

Introduction (page 4, lines 9 – 13)

Conventional strategies to mitigate the FBR have focused on changing the attributes of the implant material itself, such as its size, shape, topography, and surface coating^{28,40–47}, or involved the concomitant delivery of FBR modifying drugs, such as steroidal anti-inflammatory, anti-fibrotic, and anti-proliferative agents⁴⁸. While these strategies have shown promise, they have not succeeded in completely disarming the FBR and possess several limitations.

R1C11: N values for various experiments unclear. Please clarify where plots show n greater than number of animals and specify how many values are taken per animal. For example, in figure 4c it is not clear how many values came from each animal, and how many animals were used in each group.

Response: We thank the reviewer for this helpful comment. We have added clarification on technical replicates to the figure captions and methods and have provided Supplementary Note 1, which summarizes the n numbers for all statistical comparisons made. We feel that this clarifies the figures and data substantially.

Reviewer #2:

Summary Comment: This paper describes a soft, transport augmenting reservoir (STAR) that can modulate and adapt to the foreign body response and can improve passive macromolecular transport. This advanced soft robotic anti-FBR strategy may attract research in the field of implantable materials and drug delivery devices. This is a great work in terms of novelty and overall quality. I would recommend acceptance of this manuscript for publication in Nature Communications after necessary revision.

Response: We thank the reviewer for the excellent summary and for their constructive comments that have helped us significantly expand and strengthen this work.

R2C1: In the introduction part, the author should elaborate the differences and advantages of STAR compared to recent advanced milliscale dynamic soft reservoir (Sci. Robot. 2019. 4, eaax7043.) that was reported by the author.

Response: We thank the reviewer for this comment and have adapted the introduction to elaborate the differences and advantages of STAR compared to the dynamic soft reservoir previously reported by our lab. The new text is copied below.

Introduction (page 5, line 18 – page 6, line 4)

Preceding work from our group demonstrated a fibrosis attenuating effect elicited by a dynamic soft reservoir following acute implantation¹⁴. Here, we build on this work, and introduce a soft transport augmenting reservoir (STAR) which can persistently mitigate the dynamic FBR and maintain long-term, rapid, molecular communication with its tissue environment using two distinct and synergistic soft robotic actuation strategies: intermittent actuation (IA) and actuation-mediated rapid release (RR). Importantly, we shed light on the mechanistic underpinnings of IA and reveal an immunomodulatory effect in the acute phase of implantation, with a significant reduction in neutrophil infiltration at the pericapsular site, followed by multiphasic temporal capsular changes with chronic implantation. Lastly, in a step towards clinical translation, we demonstrate minimally invasive percutaneous delivery of a human-scale STAR device.

R2C2: Why STAR can modulate the FBR? It helps reader to understand the work better if the authors comment more on the mechanism.

Response: We thank the reviewer for this excellent comment. We have performed additional experiments to elucidate the mechanism by which STAR modulates the FBR.

We propose that STAR enabled intermittent actuation uses low-magnitude, atraumatic, tissue strain and convective flow as a defence mechanism against the invading cellular FBR, essentially acting as an oscillating shield. These dynamic biomechanical changes could directly affect cellular attachment and function, and/or promote the clearance of pro-inflammatory factors responsible for further cell recruitment and FBR propagation.

To support this hypothesis, we now demonstrate that intermittent actuation mediates local immunomodulation of the infiltrating cellular FBR (neutrophils; new Figure 4b-c) and induces multiphasic temporal fibrous capsule changes.

We have made the following changes to the manuscript to specifically address this comment.

1. We have added new data exploring the immunomodulatory effect of intermittent actuation in the acute phase of implantation followed by temporal, multiphasic capsular changes with chronic implantation. This data includes significant changes in distinct cell populations (neutrophils, myofibroblasts), collagen thickness, collagen coherency and intra-device cellular infiltration.

Results (page 16, line 7 – page 19, line 8)

Multiphasic temporal effects of intermittent actuation. Following the completion of our pre-clinical study, we next set out to analyse differences in FC composition at evolving timepoints to better understand the multiphasic cellular changes and key drivers of enhanced drug transport caused by IA (Figure 4a).

First, we investigated the initial, acute phase of the inflammatory FBR. Using immunofluorescent Ly-6G+ staining, we examined the pericapsular region for the presence of neutrophils, the first responders of the immune defence. We found that IA significantly reduces the presence of neutrophils at day 5 in comparison to the control (Figure 4b,c). This result indicates that the application of IA can mediate a localised immunomodulatory effect.

We next assessed the activation of matrix producing cells into a myofibroblast phenotype, a key contractile cell in fibrosis progression. The IA group exhibited a significant reduction in α SMA expression when compared to the control at 2 weeks (Figure 4d,e). Despite observing differences in individual cell populations (neutrophils, myofibroblasts), we did not detect differences in overall cell number at equivalent timepoints (Figure 4f,g).

Next, we investigated the macroscale capsular changes responsible for improved therapy transport. We examined evolving capsule thickness with longer periods of STAR implantation. IA mitigated capsule growth in the first 2 weeks following implantation, with a significant reduction in thickness observed w.r.t. the control at 2 weeks (Figure 4h,i). This result aligns with the enhanced blood glucose responsiveness of the IA group at early timepoints (Figure 3) and suggests that FC thickness is an important contributor to initial improvements in macromolecular transport.

By 8 weeks, however, capsule thickness had equalized between the IA and control groups (Figure 4i). This result suggests that additional mechanisms are responsible for the sustained improvement in functional response to insulin in the IA group at later timepoints (Figure 3). To investigate this further, we examined capsule vascularity, density, and the maturity of the collagen fibres. However, we did not observe differences that would account for improvements in macromolecular transport and functional effect (Supplementary Figure 7, Supplementary Note 1). By optical coherence analysis of polarised light microscopy images, we found that collagen fibres exhibited higher alignment in the IA group compared to control at week 8 (Figure 4j,k). IA appeared to increase collagen alignment over time from 2 weeks to 8 weeks, whereas there was no temporal change in alignment observed in the control group (Figure 4k). We posit that the lower degree of alignment, and therefore greater degree of fibre entanglement, in the control group creates steric hindrance which potentially slows or immobilizes transport of macromolecules through the collagenous matrix^{1,2}.

Finally, we examined the ability of IA to protect against cellular invasion and blockage of the porous membrane of STAR. Scanning electron microscopy demonstrated clear differences in cellular infiltration between the control and IA group at the 8-week timepoint (Figure 4l). This effect could be attributed to convective flow generated by STAR upon actuation (Supplementary Figure 3g). These capsular analyses reveal the pleiotropic role of IA in modulating the FBR and highlight cellular and structural changes that lead to improved transport of macromolecular therapy.

Fig. 4 | Multiphasic temporal effects of intermittent actuation (IA). **a**, Timeline of multiphasic cellular and fibrous capsule (FC) changes induced by IA. **b**, Representative fluorescent images of the FC stained with Ly-6G+ (green) and DAPI (blue). Scale bars are 20 μm . **c**, Quantification of neutrophils present within FC +/- IA at day 3 and 5. **d**, Representative fluorescent images of the FC stained with α -SMA (green) and CD31 (red). Scale bars are 50 μm . **e**, Quantification of myofibroblasts present within FC +/- IA at 2 weeks. **f**, Representative histologic images of the FC stained with haematoxylin and eosin. Scale bars are 20 μm . **g**, Quantification of total cells/capsular area +/- IA at day 3, day 5, and 2 weeks. **h**, Representative topographical reconstructions of μCT images showing the differences in FC thickness +/- IA at 2 weeks. **i**, Average FC thickness of the control and actuated groups at day 3, day 5, 2 weeks, and 8 weeks with two measurements taken per animal. **j**, Representative polarised light microscopy images of the FC obtained after picosirius red staining at 8 weeks. Scale bars are 100 μm . **k**, Quantification of the FC collagen fibre orientation by optical coherency with 60 ROIs per animal. **l**, Representative SEM images demonstrating reduced cellular invasion with actuation at the 8-week timepoint. Scale bars are 500 μm . $n = 2-6$ animals per group; data are means \pm standard error of mean; * $p < 0.05$, ** $p < 0.01$, *** $p < 0.001$, **** $p < 0.0001$. See Supplementary Note 1 for detailed statistical analyses.

2. We demonstrate that STAR enabled intermittent actuation causes deflection and strain at the tissue-contacting membrane and promotes convective fluid flow in proximity to the device (Figure 1e,f). Furthermore, we have added data exploring the relationship between membrane deflection and tissue strain (Supplementary Figure 3).

Results (page 8, line 1 – 7)

As part of device design and optimisation, we performed finite element (FE) simulations to understand the biomechanical changes mediated by actuation, particularly the relationships between membrane deflection, convective flow, and tissue strain (Figure 1e,f; Supplementary Figure 3). Based on recently reported results³, we designed our soft robotic actuation strategy to induce tissue strain that would fall within the atraumatic range (<40%), and hypothesise that this regimen would mitigate the FBR by creating convective flow disruptive to the cellular immune response.

Fig. 1 | e, FE model showing peri-implant fluid velocity of convective flow during actuation. **f**, FE model estimating maximum principal tissue strain induced by actuation.

Supplementary Fig. 3 | Membrane deflection in STAR. **a**, Schematics of bi-chambered STAR device. Single pressure input allows for variable deflection in a single device. **b**, Prototype of bi-chambered STAR device manufactured by the process described in Supplementary Figure 2. **c**, Schematic depicting lateral view of bi-chambered device, and differential deflection height of each chamber upon uniform pressurization. **d**, Maximum membrane deflection for the 3 mm and 6 mm chambers during actuation with a 1–9 psi pressure input. Pressure of 2 psi was chosen for the preclinical experiments. **e**, Finite element (FE) model showing tissue deflection caused by STAR membrane at 1, 2, and 3 psi pressure input. **f**, FE model showing tissue strain caused by STAR membrane at 1, 2, and 3 psi pressure input. **g**, FE model showing peri-implant fluid velocity of convective flow around the large and small chamber during actuation at 2 psi.

3. We have further expanded the introduction to better frame our mechanotherapeutic strategy:

Introduction (page 5, line 5 – 17)

Dynamically altering the local biomechanical environment at the implant site is one such promising, yet underexplored, drug-free approach¹⁴. Cells in our body are exquisitely sensitive to their mechanical environment, with loading playing a pivotal role in cell functions such as differentiation¹⁵, proliferation¹⁶, and migration¹⁷. Historically, studies have observed biomechanical stress as a pro-fibrotic or regenerative stimulus¹⁸, demonstrating that application of stretch⁷⁻⁹, fluid flow^{10,11}, or compression¹² to cells can lead to increased deposition of collagenous matrix. Accordingly, many anti-FBR strategies have focused on minimizing the mechanical mismatch, interfacial stress, and movement between the implant and local tissue. Our research seeks to challenge this status quo and reveals the potential for a dynamic mechanotherapeutic that uses low-magnitude, atraumatic, tissue strain and convective flow as a defence mechanism against the invading cellular FBR. Interestingly, some studies have observed that small magnitude, dynamic loading has anti-inflammatory effects, and indicated the presence of a therapeutic threshold, beyond which tissue damage and inflammation occurs^{3,13}.

4. We have added a schematic highlighting the proposed mechanism of action to Figure 1a, with accompanying text:

Results (page 7, lines 2 – 7)

Our lab previously demonstrated the fibrotic attenuating potential of a dynamic device during the initial stages of the FBR (2 weeks). Based on this foundational work, we propose that application of intermittent, cyclical, low amplitude actuation can act as an oscillating shield against the invading, multiphasic FBR, induce local immunomodulatory effects, and create a favourable environment for the rapid long-term transport of macromolecular drug therapy (Figure 1a).

Fig. 1 | Design of soft transport augmenting reservoir (STAR). a, Proposed mechanism of STAR: intermittent actuation shields against the foreign body response, creating a favourable environment for the rapid long-term transport of macromolecular drug therapy.

5. Finally, in the discussion, we comment on STAR's mechanically disruptive oscillation at a cellular level and discuss potential future avenues of explorative research.

Discussion (page 25, lines 14 – 20)

A recent study by Seo et al.³ corroborates the mechanosensitive nature of neutrophil cell populations following dynamic loading for skeletal muscle regeneration. The authors postulated that mechanical flushing of chemoattractants was responsible for the reported decrease in

neutrophils³. In this context, our work motivates further study on the effect of IA on pro-inflammatory chemoattractant gradients such as interleukin 1, 6 and 8 versus direct mechanical effects on cell attachment, orientation, and function.

R2C3: Several important and representative advances relevant to the material strategies to resist the FBR is missing. Related references should be added, such as those advances reported in *Nat. Commun.* 2019, 10 (1), 5262; *Nat. Commun.* 2021, 12 (1), 5327; *Angew. Chem. Int. Ed.* 2020, 59 (24), 9586; *Adv. Funct. Mater.* 2019, 29, 1900140.

Response: We thank the reviewer for highlighting this omission. We have included additional references to highlight important advances in material strategies to resist the FBR. These are copied in blue below.

Q. Liu, A. Chiu, L. H. Wang, D. An, M. Zhong, A. M. Smink, B. J. de Haan, P. de Vos, K. Keane, A. Vegge, et al., *Nat. Commun.* 2019 101 **2019**, 10, 1.

D. Zhang, Q. Chen, Y. Bi, H. Zhang, M. Chen, J. Wan, C. Shi, W. Zhang, J. Zhang, Z. Qiao, et al., *Nat. Commun.* 2021 121 **2021**, 12, 1.

D. Zhang, Q. Chen, W. Zhang, H. Liu, J. Wan, Y. Qian, B. Li, S. Tang, Y. Liu, S. Chen, et al., *Angew. Chemie* **2020**, 132, 9673.

J. Zhang, Y. Zhu, J. Song, T. Xu, J. Yang, Y. Du, L. Zhang, *Adv. Funct. Mater.* **2019**, 29, 1900140.

R2C4: Why are the n values of several different groups different in some experiments such as in Figure 3.

Response: We thank the reviewer for this comment. N numbers varied in our original submission for two reasons. First, animals that were assessed for immunohistochemistry and fibrous capsule (FC) thickness at intermediate timepoints (2 weeks after device implantation) were not included in subsequent insulin transport tests due to the terminal nature of our FC assessment. Second, due to the bi-chamber design of STAR, we used two technical replicates per animal for capsule thickness, one for each chamber.

In our revised longitudinal preclinical study, we have carefully included n numbers for each group at each time point (Figure 3a). Note that the baseline preclinical study was performed post-hoc using different animals, hence the increase in n from Day 3 to Week 2.

Additionally, we have clarified technical replicates where appropriate, in both the figure captions and methods. Coherency analysis for collagen alignment and fibrous capsule thickness are the only two analyses that make use of technical replicates.

Finally, we have created a Supplementary Note 1, which summarizes all the n numbers used for all statistical comparisons in the manuscript. The changes are summarized below.

Fig. 3 | Intermittent actuation (IA) improves long-term macromolecule delivery. **a**, Preclinical study timeline used to evaluate the effect of IA on insulin transport through a fibrous capsule. [...] Data are means \pm standard error of mean; * $p < 0.05$, ** $p < 0.01$, *** $p < 0.001$. † Baseline study was performed post-hoc with separate mice. # Control mice removed from study at intermediate time points due to self-inflicted device damage with subsequent decrease in n .

Methods (page 42, lines 12 – 17)

Analysis of collagen orientation. Quantification of the collagen content was performed using a previously reported technique^{5,35,75–77}. In ImageJ, six ROIs were manually selected for each image obtained by polarised light microscopy to completely capture all collagen fibres imaged while minimizing noise from background artifact. Using the OrientationJ plugin, the coherency of each ROI was calculated. Each device sample generated 60 total datapoints (6 ROIs per section, 10 sections per sample) (Figure 4k).

Methods (page 43, lines 8 – 11)

Wall thickness of the isolated FC under both the small and large chambers was measured (Supplementary Figure 10a,b), which demonstrated no difference. Similarly, thickness measurements were taken at device edges which did not differ significantly from thickness values underneath device chambers (Supplementary Figure 10c,d).

R2C5: How were the number of CD31+ blood vessel and volume of cells stained with α -SMA counting and quantification performed? What identification criteria, counting area were used?

We thank the reviewer for this comment and now clarify additional experimental details in the revised methods section, reproduced below.

Methods (page 41, line 11 – page 42, line 11)

Analysis of myofibroblasts and neutrophils. Ten random fields of view were acquired from two sections using confocal microscopy. The volume fraction of α SMA+ cells and Ly-6G+ cells within the FC were estimated by an unbiased stereological counting technique using ImageJ (Fiji version 2.0.0) software. The same unbiased stereological method was used to count both myofibroblasts and neutrophils. A random offset stereological square grid (10,000 cm^2) was superimposed onto the images to provide test points. To calculate area fraction, intersections falling on positively stained cells were counted and expressed as a ratio of total grid intersections within the FC. To estimate the relative volume of myofibroblast cells per FC and assess whether the presence of

myofibroblasts was perturbed by actuation, the volume fraction of α SMA+ cells was normalized to a volume defined by FC thickness multiplied by unit area using the following equation: Relative Volume of α SMA+ cells (mm^3) = Volume Fraction of α SMA+ cells / FC Thickness (mm) \times 1 mm \times 1 mm.

Numerical density of Blood Vessels per Unit Area. Quantification of vascularization of the FC was performed using a previously reported technique^{49–51}. A systemic random sampling strategy was used. From each tissue section, ten non-overlapping images were taken of the FC in the area of interest. Number of blood vessels per area (N_a) were calculated using an unbiased counting frame (grid size = 2000 cm^2). The numbers of points which coincided with blood vessels were counted. The volume fractions of blood vessels were then calculated by expressing the proportion of points hitting blood vessels as a fraction of the total number of points observed in the tissue. The forbidden line rule was followed to ensure blood vessels were only counted once. The application of this counting rule generates an unbiased estimate of the number of blood vessels per unit area ($N_a = C_N \times C_{\text{pts}} \times A$; taking C_N as cumulative number of blood vessels counted and C_{pts} as cumulative number of points in the area of interest).

R2C6: The examination of the inflammatory response in the early stage is important evaluation criteria to the FBR. I suggest that the author examine the inflammatory response after 7 days or 15 days of implantation.

Response: We really appreciate this excellent suggestion, which led us to perform new experiments that expanded our understanding of why intermittent actuation is effective.

We examined the neutrophil response at early timepoints (day 3 and day 5) following implantation. The actuation group significantly reduced the presence of neutrophils at day 5 in comparison to the control. This result indicates that the application of intermittent actuation can mediate a localized immunomodulatory effect.

Neutrophil infiltration following implantation is an important first step in the initiation and propagation of the inflammatory process, and the subsequent recruitment of cell populations known to develop the FBR. As the reviewer alluded to above, early resolution of this inflammatory process may have important long-term consequences on the FBR.

Results (page 16, line 7 – page 19, line 8)

Multiphasic temporal effects of intermittent actuation. Following the completion of our pre-clinical study, we next set out to analyse differences in FC composition at evolving timepoints to better understand the multiphasic cellular changes and key drivers of enhanced drug transport caused by IA (Figure 4a).

First, we investigated the initial, acute phase of the inflammatory FBR. Using immunofluorescent Ly-6G+ staining, we examined the pericapsular region for the presence of neutrophils, the first responders of the immune defence. We found that IA significantly reduces the presence of neutrophils at day 5 in comparison to the control (Figure 4b,c). This result indicates that the application of IA can mediate a localised immunomodulatory effect.

We next assessed the activation of matrix producing cells into a myofibroblast phenotype, a key contractile cell in fibrosis progression. The IA group exhibited a significant reduction in α SMA expression when compared to the control at 2 weeks (Figure 4d,e). Despite observing differences in individual cell populations (neutrophils, myofibroblasts), we did not detect differences in overall cell number at equivalent timepoints (Figure 4f,g).

Fig. 4 | Multiphasic temporal effects of intermittent actuation (IA). **a**, Timeline of multiphasic cellular and fibrous capsule (FC) changes induced by IA. **b**, Representative fluorescent images of the FC stained with Ly-6G+ (green) and DAPI (blue). Scale bars are 20 μm . **c**, Quantification of neutrophils present within FC +/- IA at day 3 and 5. **d**, Representative fluorescent images of the FC stained with α -SMA (green) and CD31 (red). Scale bars are 50 μm . **e**, Quantification of myofibroblasts present within FC +/- IA at 2 weeks. **f**, Representative histologic images of the FC stained with haematoxylin and eosin. Scale bars are 20 μm . **g**, Quantification of total cells/capsular area +/- IA at day 3, day 5, and 2 weeks. **h**, Representative topographical reconstructions of μCT images showing the differences in FC thickness +/- IA at 2 weeks. **i**, Average FC thickness of the control and actuated groups at day 3, day 5, 2 weeks, and 8 weeks with two measurements taken per animal. **j**, Representative polarised light microscopy images of the FC obtained after picosirius red staining at 8 weeks. Scale bars are 100 μm . **k**, Quantification of the FC collagen fibre orientation by optical coherency with 60 ROIs per animal. **l**, Representative SEM images demonstrating reduced cellular invasion with actuation at the 8-week timepoint. Scale bars are 500 μm . $n = 2-6$ animals per group; data are means \pm standard error of mean; * $p < 0.05$, ** $p < 0.01$, *** $p < 0.001$, **** $p < 0.0001$. See Supplementary Note 1 for detailed statistical analyses.

Reviewer #3:

Summary Comment: The authors describe their soft, transport augmenting reservoir (STAR) implant that builds upon recent evidence of reduction of FBR using a dynamic implant motion (see Ref 44, 45). STAR is an actuatable, implantable drug delivery pouch that uses externally percutaneously applied pneumatic pressure to induce convective fluid flow to modulate the FBR, reduce fibrosis, and improve the transport of small- and macromolecular therapeutics.

Unique and impacting features of this report include:

1. They leverage a fibrosis-attenuating externally activated percutaneous oscillatory inflation/deflation implant preconditioning regimen to alter the foreign body capsule (FC), and improve passive macromolecular transport in rodent subcutaneous models.
2. They show a direct inverse relationship between STAR drug release and FC capsule thickness. Two discrete preconditioning deflection regimens (0.58 mm and 1.3 mm, respectively) are applied to affect local FC formation.
3. They further demonstrate that STAR can adapt to increasing capsule thicknesses at later timepoints by utilizing bursts of STAR inflatable membrane external percutaneous actuation to drive convective drug transport through the FC at the implant site in vivo.
4. Importantly, in both these drug transport-augmenting strategies, they claim to demonstrate functionality by delivering an active therapeutic (insulin) that elicits a known therapeutic effect clinically.
5. They demonstrate how such a device is feasibly implanted in human abdominal locations using a human cadaver surgical implant procedure.

Notably, STAR preconditioning-mediated differences observed in macromolecular transport through the FC are not attributed to changes in FC vascularity or FC collagen content or organization. They did not observe any within-group differences for FC thicknesses between the smaller and larger chamber wall implant in situ membrane deflections.

The approach is validated by experimental design; the quality results/data follow logically from the experiments, and their study conclusions are reasonable from the reported results. While well-written with an important creative and impacting message to the medical device and implant community, the following considerations would improve the paper's clarity and delivery.

Response: We thank the reviewer for this excellent and eloquent summary. Furthermore, we appreciate the constructive and insightful comments below, which have greatly aided STAR technology development.

R3C0: Is the same mechanism also operable in human sub-muscular abdominal compartment as proposed for translation with the same membrane deflection amplitudes and timings?

Response: We thank the reviewer for this important comment. Translating pre-clinical findings to the clinic is exceedingly difficult and unpredictable. We have thus tempered our results with a limitations section discussing the differential anatomic and microenvironmental factors in animal models that may be barriers to clinical translation.

The relevant section is copied below in blue.

Discussion (page 28, line 21 – page 29, line 10)

Though we have demonstrated robust preclinical results in a long-term mouse model, there are several limitations and barriers to clinical translation. Our findings from implantation in the dorsal subcutaneous space of mice may not directly predict similar results in humans at different anatomical locations (e.g., abdominal intermuscular space), and further work is needed to understand how these anatomic and microenvironmental differences impact the effects of STAR. Although rodent models have been extensively used to study the FBR^{28,29}, rodents have been shown to have different tissue collagen content in the subcutaneous space surrounding an implant and different metabolites in the interstitial fluid at implant interfaces, as compared to humans³⁰. Moreover, differences in rodent skin, fur, and behaviours may subject implanted devices to different biomechanical forces than in humans³¹. Encouragingly, thus far, we have observed similar FBR mitigating effects with an IA regimen that is agnostic of species and device design¹⁴. In addition to this, the presence of conserved inflammatory pathways in FC formation across species³² suggest that STAR may have a similar benefit in prolonging implant lifespan in humans.

Introduction: please consider these clarifications/details:

R3C1: Authors write: “heart valves⁷, pacemakers⁸, artificial joints⁹, orthopedic implants¹⁰, hernia meshes¹¹, and stents¹².”: but CGMs represent a relevant additional implant class for the diabetes theme of this manuscript. (see FBR-relevant CGM implant issues in book: Implantable Glucose Sensing, eds. J. Stenken, D. Cunningham, Wiley, NY, USA, ISBN 978-0470112960, 2009.)

Response: We have modified the introduction accordingly to focus on CGMs and insulin pumps more directly. We thank the reviewer for this excellent point.

Introduction (page 3, line 10 – page 4, line 8)

This fibrous barrier is particularly deleterious for biosensors, such as continuous glucose monitors, and controlled drug release devices, such as insulin pumps, which rely on interactive communication with their local tissue environment^{29,39,52}. In such cases, the formation of a hypopermeable capsule can impede transport of molecules, both to⁵³ and from^{54,55} the implant, and lead to therapy failure.

One pertinent example is the management of type 1 diabetes, a chronic disease affecting 18 million people worldwide, with an annual economic burden of greater than \$90 billion USD³³. Successful implementation and clinical adoption of an artificial pancreas combining continuous glucose monitoring with the rapid, responsive, release of insulin (or glucagon) would vastly improve outcomes and quality of life for this patient population. The development of a fully automated closed loop insulin delivery system would reduce user burden, remove the need for multiple daily injections, and increase time spent in the optimal blood glucose range, which is imperative for the prevention of long-term diabetic complications. Unfortunately, current efforts at developing such a device have been hindered by the dynamic and unpredictable FBR, leading to glucose sensing inaccuracy, inhibition of insulin release, and gradual loss of functionality in the weeks to months following implantation^{34–37}. Looking towards the future, living implants containing stem cell derived pancreatic β -cells represent a potential cure for diabetes. However, the attenuation of oxygen and molecular transport due to the FC barrier still constitutes a major

hurdle to successful clinical translation of these implants^{25,29,38,39}. It is evident that a method to (i) mitigate the FBR or (ii) improve transport across the FC could transform the management of this pervasive disease. Furthermore, such a method could have broader implications for a range of diseases and device-based treatments affected by the FBR.

R3C2: Authors write: “and delivery of anti-inflammatory drugs 30,31 have been developed to modify the FBR”: many different drugs have been delivered locally than merely anti-inflammatories (anti-histamines, antimicrobials, steroids, angiogenic agents, anti-fibrotics (this is reviewed in Avula and Grainger, “Addressing medical device challenges with drug/device combinations,” Chapter 3, in Drug-Device Combinations for Chronic Diseases, R. Siegel., S.P. Lyu, eds., Wiley, New York, USA ISBN: 978-1-118-12000-2, 280 pp., 2015.)

Response: We thank the reviewer for this helpful comment. We have broadened this discussion to include a wider range of therapeutics including steroidal anti-inflammatory, antifibrotic, and antiproliferative agents. Furthermore, we have added the reference that was helpfully provided to direct the reader to a more in-depth discussion of the topic.

The text now states the following:

Introduction (page 4, lines 9 – 12)

Conventional strategies to mitigate the FBR have focused on changing the attributes of the implant material itself, such as its size, shape, topography, and surface coating^{28,40–47}, or involved the concomitant delivery of FBR modifying drugs, such as steroidal anti-inflammatory, anti-fibrotic, and anti-proliferative agents⁴⁸.

R3C3: Authors write: “and need to be replenished for long term efficacy^{31–33}.”: the DES taught the community that long-term on-going drug release wasn't necessary if the initial acute host response was effectively attenuated. Reloading and continual release may not be necessary for the effective approach for all implants. This depends on the specific pharmacology.

Response: We thank the reviewer for pointing this out. We have amended the text to give a more expanded and nuanced discussion of the risks of FBR agents.

Introduction (page 4, line 17 – page 5, line 4)

Sustained, systemic delivery of therapeutics such as non-steroidal anti-inflammatories is associated with a range of toxicities in the liver, kidneys, heart, and gastrointestinal tract⁵⁶. Local targeted delivery can reduce off-target effects but may still adversely affect the underlying tissue or interfere with the mechanism of action of the implantable device. For example, local delivery of dexamethasone can mitigate the FBR, but not without suppressing underlying tissue regeneration⁵⁷. Furthermore, the effect of long-term immunosuppression on the behaviour and secretome of cell-based therapeutics is unclear. Finally, a local depot of drug is finite, often lasting 1-2 months, while many needs are lifelong⁵⁶. Thus, depending on drug pharmacology and clinical context, the immune and fibrotic response may rebound once the residual effects of drug inhibition dissipate. A long-term, drug-free method that can modulate and adapt to the FBR over time would, therefore, be highly desirable to address these limitations.

R3C4: “Altering the biomechanics at the implant site”: local tissue-implant biomechanics issues described in these papers ought to be considered here to fully describe the in situ situation: Helton et al., J Diabetes Sci Technol 2011;5(3):647-656; Helton et al., J Diabetes Sci Technol 2011;5(3):632-646.

Response: We thank the reviewer for suggesting that we consider these excellent articles. With reference to these papers, we have further expanded on the rationale for our study and device design.

1. We have further expanded the introduction to better elucidate the ideal loading parameters for reducing the FBR:

Introduction (page 5, lines 5 – 17)

Dynamically altering the local biomechanical environment at the implant site is one such promising, yet underexplored, drug-free approach¹⁴. Cells in our body are exquisitely sensitive to their mechanical environment, with loading playing a pivotal role in cell functions such as differentiation¹⁵, proliferation¹⁶, and migration¹⁷. Historically, studies have observed biomechanical stress as a pro-fibrotic or regenerative stimulus¹⁸, demonstrating that application of stretch⁷⁻⁹, fluid flow^{10,11}, or compression¹² to cells can lead to increased deposition of collagenous matrix. Accordingly, many anti-FBR strategies have focused on minimizing the mechanical mismatch, interfacial stress, and movement between the implant and local tissue. Our research seeks to challenge this status quo and reveals the potential for a dynamic mechanotherapeutic that uses low-magnitude, atraumatic, tissue strain and convective flow as a defence mechanism against the invading cellular FBR. Interestingly, some studies have observed that small magnitude, dynamic loading has anti-inflammatory effects, and indicated the presence of a therapeutic threshold, beyond which tissue damage and inflammation occurs^{3,13}.

2. We have further expanded the text associated with Figure 1 to explain key device design and material choices:

Results (page 7, lines 8 – 20)

To test this hypothesis, we first designed a reservoir suitable for long-term tissue implantation and the precise repeatable delivery of both drug and actuation therapy. Figure 1b shows the multi-layered composition of STAR, with a low-profile design that minimises the presence of sharp angles or edges which may exacerbate the FBR^{31,58}. A therapeutic chamber lies in direct contact with underlying tissue and is separated by a membrane with an array of 10 mm pores (Supplementary Figure 1). A connected indwelling catheter line allows for delivery of drug therapy with temporal control (Figure 1b,c). Superimposed on the therapeutic chamber is an actuation chamber that can be pressurized to elicit controlled oscillation of the porous, tissue-contacting membrane (Figure 1c,d; Supplementary Video 1). Imbalances between the mechanical properties of the implant and the surrounding tissue are also known to exacerbate FC formation, with stiffer implants eliciting a heightened immune response⁵⁸. For this reason, STAR was manufactured from thermoplastic polyurethane (TPU) with an elastic modulus of ~15 MPa (Figure 1d), similar to that of extracellular matrix^{59,60}.

Results: clarifications requested here:

R3C5: “by thickness measurements of the isolated FC under both the small and large chambers”: what is the specific physical location of the FC thickness measurements? STAR corners or edges or simply flat STAR surfaces? Corners and edges exacerbate FBR thickness.

Response: We thank the reviewer for seeking additional clarification on the location of FC thickness measurements.

FC thickness measurements were obtained from regions of interest directly below each of the porous therapy reservoirs of the STAR device. We have included a schematic below to demonstrate this visually (Supplementary Figure 10c). This region is of particular interest to us as the macromolecular therapy must travel through the capsule in this representative region to have a functional effect.

We appreciate that corners and edges can exacerbate fibrosis and have now performed additional thickness analysis on STAR edge regions (Supplementary Figure 10c). Our data suggests that there is no significant difference in FC thickness between the edge and reservoir regions in either control or IA groups (Supplementary Figure 10d). Interestingly the static control group exhibited higher variance.

We have added Supplementary Figure 10 to the manuscript, and the following text to the methods.

Methods (page 43, lines 8 – 11)

Wall thickness of the isolated FC under both the small and large chambers was measured (Supplementary Figure 10a,b), which demonstrated no difference. Similarly, thickness measurements were taken at device edges which did not differ significantly from thickness values underneath device chambers (Supplementary Figure 10c,d).

Supplementary Fig. 10 | Fibrous capsule thickness analysis using μ CT. **a**, 3D reconstruction of STAR by μ CT in the transverse plane, demonstrating small (3 mm) and large (6 mm) chambers of the device with fibrous encapsulation. Scale bar is 5 mm. **b**, Comparison of capsule thickness in large and small chambers at 2 weeks and 8 weeks. $n = 2-5$ with no technical replicates. **c**, Schematic showing the STAR device with bi-chambered design. Dashed boxes denote areas of FC measurement underneath the 6 mm large and 3 mm small chambers (white) and at reservoir edge (black). **d**, Box and whiskers plot showing difference in mean FC Thickness adjacent to reservoir and edge regions of STAR (difference = reservoir – edge) measured at 8 weeks with $n = 4$ mice per group and one thickness measurement per chamber (i.e., 2 technical replicates). Box represents mean (horizontal line) \pm standard error of mean (height of box). Whiskers represent the range.

R3C6: “and the downstream clinical effects” and “elicit a clinical effect”: no clinical effects are shown since no human therapeutic treatments are shown. Please modify.

Response: We thank the reviewer for this comment. We have replaced ‘clinical effect’ with ‘functional effect’ where appropriate in the manuscript.

R3C7: “used to improve distribution of chemotherapy in deep brain tumour targets²⁸⁻³¹”: though therapeutic outcome in this case is relatively modest, even perhaps clinically insignificant.

Response: We thank the reviewer for highlighting this. To address the comment, we have added an additional qualifying statement. The text now reads:

Discussion (page 27, lines 6 – 8)

Convection-enhanced delivery has been successfully used to improve distribution of chemotherapy in deep brain tumour targets, albeit with modest clinical outcomes^{28,46,56,61}.

Methods: improved details are requested a specified here:

R3C8: “The STAR device was then inserted”: dimensions of this implanted device?

Response: We thank the reviewer for this comment.

We have added more details on the device dimensions to Supplementary Figure 3 (copied below), and we have referenced this figure when describing the device in the methods.

Methods (page 35, lines 4 – 5)

The 15 × 18 × 2 mm STAR device (Supplementary Figure 3a,b) was then inserted under the skin *via* the superior incision site and tunneled inferiorly into position.

Subset of Supplementary Figure 3 | Membrane deflection in STAR. a, Schematics of bi-chambered STAR device. Single pressure input allows for variable deflection in a single device. **b,** Prototype of bi-chambered STAR device manufactured by the process described in Supplementary Figure 2.

R3C9: “controlled input pressure of 2 psi at 1 Hz every 12 hours for 14 days”: what STAR wall deflection amplitude does this create in the STAR device wall against local tissue?

Response: We thank the reviewer for this comment.

In vitro measurements taken from an unconstrained STAR device shows a membrane deflection amplitude of ~1.3mm at 2 psi input pressure (Supplementary Figure 3c,d). Because the device is constrained by the skin and subcutaneous tissue *in vivo*, we expect the constrained STAR wall deflection amplitude to be lower than its unconstrained amplitude. Therefore, we performed finite element simulations to shed light on the actual *in vivo* membrane deflection (which is the local tissue deflection) induced by STAR actuation. At 2 psi, we found the tissue deflection to be 0.73 mm, which is approximately half of the unconstrained amplitude.

This full analysis is included in Supplementary Figure 3.

Supplementary Fig. 3 | Membrane deflection in STAR. **c**, Schematic depicting lateral view of bi-chambered device, and differential deflection height of each chamber upon uniform pressurization. **d**, Maximum membrane deflection for the 3 mm and 6 mm chambers during actuation with a 1–9 psi pressure input. Pressure of 2 psi was chosen for the preclinical experiments. **e**, Finite element (FE) model showing tissue deflection caused by STAR membrane at 1, 2, and 3 psi pressure input. **f**, FE model showing tissue strain caused by STAR membrane at 1, 2, and 3 psi pressure input.

R3C10: “pneumatically activated using a pressure input of 2 psi.”: duration of this activation? constant isobaric stimulus, no cycling?

Response: We thank the reviewer for highlighting this omission. We have further clarified the actuation-mediated rapid release regimen in the methods.

Methods (page 36, lines 18 – 21)

The STAR actuation reservoir was connected to a custom-made pneumatic control unit, *via* the transcutaneous self-sealing access port, using a PNP3M connected (Instech) and pneumatically activated for 5 cycles of 2 psi cyclical pressure at 1 Hz. The duration of activation and evacuation were equivalent.

R3C11: “and actuation catheter of each STAR device.”: dimensions of this implanted device?

Response: We thank the reviewer for this comment and have amended the text to include dimensions.

Methods (page 31, lines 2 – 3)

Final devices measured 12 mm in length with the semi-circular reservoir measuring 3.9 mm in height and 3.5 mm in diameter, with variable lengths of 3Fr TPU catheter tubing.

R3C12: “Genhance 750”: chemical description of this substance should be briefly included.

Response: Genhance 750 is a small molecule fluorescent drug analogue designed for preclinical use in large and small animals. It has been used as a reporter for drug delivery approaches. We have contacted Genhance’s manufacturer (Perkin Elmer) for its chemical formulation, however due to the proprietary nature of the product, this was not made available to us. We have provided further clarification in the text.

Results (page 19, lines 25 – 27)

On day 24 following implantation, we monitored the distribution area of a fluorescent small molecule drug analogue (Genhance 750) using an *in vivo* imaging system (IVIS) (Figure 5g).

Limitations:

R3C13: Some expert comments on limitations of the SD Rat and C57BL6 mice STAR dorsal pocket implant fibrosis models reported in predicting actual human FBR and STAR implant fibrotic responses in the different anatomical location (abdominal sub-muscular/intermuscular) would help balance the discussion.

Response: We thank the reviewer for raising this important point. As mentioned in response to comment R3C0, there are certainly significant differences in the dorsal subcutaneous space in rodents compared to the human intermuscular compartment which may prove to be a barrier to clinical translation.

We have added a limitations paragraph to our discussion to specifically address this point.

Discussion (page 28, line 21 – page 29, line 10)

Though we have demonstrated robust preclinical results in a long-term mouse model, there are several limitations and barriers to clinical translation. Our findings from implantation in the dorsal subcutaneous space of mice may not directly predict similar results in humans at different anatomical locations (e.g., abdominal intermuscular space), and further work is needed to

understand how these anatomic and microenvironmental differences impact the effects of STAR. Although rodent models have been extensively used to study the FBR^{28,29}, rodents have been shown to have different tissue collagen content in the subcutaneous space surrounding an implant and different metabolites in the interstitial fluid at implant interfaces, as compared to humans³⁰. Moreover, differences in rodent skin, fur, and behaviours may subject implanted devices to different biomechanical forces than in humans³¹. Encouragingly, thus far, we have observed similar FBR mitigating effects with an IA regimen that is agnostic of species and device design¹⁴. In addition to this, the presence of conserved inflammatory pathways in FC formation across species³² suggest that STAR may have a similar benefit in prolonging implant lifespan in humans.

Weakness:

R3C14: Mechanism for how the FC around the implant is modulated by exogenously induced wall deflections is missing. This limitation is mentioned but remains to be addressed. As such then, the reporting is empirical and observational, producing intriguing effects but without a rational explanation. Phenomenologically, the cause-effect description is intriguing as described.

Response: We thank the reviewer for this insightful comment and look to address this weakness with the additions of significant new data and insights.

We propose that STAR enabled intermittent actuation uses low-magnitude, atraumatic, tissue strain and convective flow as a defence mechanism against the invading cellular FBR, essentially acting as an oscillating shield. These dynamic biomechanical changes could directly affect cellular attachment and function, and/or promote the clearance of pro-inflammatory factors such as interleukin 1,6, and 8, responsible for further cell recruitment, and FBR propagation.

To support this hypothesis, we demonstrate that intermittent actuation mediates local immunomodulation of the infiltrating cellular FBR (neutrophils) and induces multiphasic temporal fibrous capsule changes.

We have made the following changes to the manuscript to specifically address this comment.

1. We have added new data exploring the immunomodulatory effect of intermittent actuation in the acute phase of implantation followed by temporal, multiphasic capsular changes with chronic implantation. This data includes significant changes in distinct cell populations (neutrophils, myofibroblasts), collagen thickness, collagen coherency and intra-device cellular infiltration.

Results (page 16, line 7 – page 19, line 8)

Multiphasic temporal effects of intermittent actuation. Following the completion of our pre-clinical study, we next set out to analyse differences in FC composition at evolving timepoints to better understand the multiphasic cellular changes and key drivers of enhanced drug transport caused by IA (Figure 4a).

First, we investigated the initial, acute phase of the inflammatory FBR. Using immunofluorescent Ly-6G+ staining, we examined the pericapsular region for the presence of neutrophils, the first responders of the immune defence. We found that IA significantly reduces

the presence of neutrophils at day 5 in comparison to the control (Figure 4b,c). This result indicates that the application of IA can mediate a localised immunomodulatory effect.

We next assessed the activation of matrix producing cells into a myofibroblast phenotype, a key contractile cell in fibrosis progression. The IA group exhibited a significant reduction in α SMA expression when compared to the control at 2 weeks (Figure 4d,e). Despite observing differences in individual cell populations (neutrophils, myofibroblasts), we did not detect differences in overall cell number at equivalent timepoints (Figure 4f,g).

Next, we investigated the macroscale capsular changes responsible for improved therapy transport. We examined evolving capsule thickness with longer periods of STAR implantation. IA mitigated capsule growth in the first 2 weeks following implantation, with a significant reduction in thickness observed w.r.t. the control at 2 weeks (Figure 4h,i). This result aligns with the enhanced blood glucose responsiveness of the IA group at early timepoints (Figure 3) and suggests that FC thickness is an important contributor to initial improvements in macromolecular transport.

By 8 weeks, however, capsule thickness had equalized between the IA and control groups (Figure 4i). This result suggests that additional mechanisms are responsible for the sustained improvement in functional response to insulin in the IA group at later timepoints (Figure 3). To investigate this further, we examined capsule vascularity, density, and the maturity of the collagen fibres. However, we did not observe differences that would account for improvements in macromolecular transport and functional effect (Supplementary Figure 7, Supplementary Note 1). By optical coherence analysis of polarised light microscopy images, we found that collagen fibres exhibited higher alignment in the IA group compared to control at week 8 (Figure 4j,k). IA appeared to increase collagen alignment over time from 2 weeks to 8 weeks, whereas there was no temporal change in alignment observed in the control group (Figure 4k). We posit that the lower degree of alignment, and therefore greater degree of fibre entanglement, in the control group creates steric hindrance which potentially slows or immobilizes transport of macromolecules through the collagenous matrix^{1,2}.

Finally, we examined the ability of IA to protect against cellular invasion and blockage of the porous membrane of STAR. Scanning electron microscopy demonstrated clear differences in cellular infiltration between the control and IA group at the 8-week timepoint (Figure 4l). This effect could be attributed to convective flow generated by STAR upon actuation (Supplementary Figure 3g). These capsular analyses reveal the pleiotropic role of IA in modulating the FBR and highlight cellular and structural changes that lead to improved transport of macromolecular therapy.

Fig. 4 | Multiphasic temporal effects of intermittent actuation (IA). **a**, Timeline of multiphasic cellular and fibrous capsule (FC) changes induced by IA. **b**, Representative fluorescent images of the FC stained with Ly-6G+ (green) and DAPI (blue). Scale bars are 20 μ m. **c**, Quantification of neutrophils present within FC +/- IA at day 3 and 5. **d**, Representative fluorescent images of the FC stained with α -SMA (green) and CD31 (red). Scale bars are 50 μ m. **e**, Quantification of myofibroblasts present within FC +/- IA at 2 weeks. **f**, Representative histologic images of the FC stained with haematoxylin and eosin. Scale bars are 20 μ m. **g**, Quantification of total cells/capsular

area +/- IA at day 3, day 5, and 2 weeks. **h**, Representative topographical reconstructions of μ CT images showing the differences in FC thickness +/- IA at 2 weeks. **i**, Average FC thickness of the control and actuated groups at day 3, day 5, 2 weeks, and 8 weeks with two measurements taken per animal. **j**, Representative polarised light microscopy images of the FC obtained after picosirius red staining at 8 weeks. Scale bars are 100 μ m. **k**, Quantification of the FC collagen fibre orientation by optical coherency with 60 ROIs per animal. **l**, Representative SEM images demonstrating reduced cellular invasion with actuation at the 8-week timepoint. Scale bars are 500 μ m. $n = 2-6$ animals per group; data are means \pm standard error of mean; * $p < 0.05$, ** $p < 0.01$, *** $p < 0.001$, **** $p < 0.0001$. See Supplementary Note 1 for detailed statistical analyses.

2. We demonstrate that STAR enabled intermittent actuation causes deflection and strain at the tissue-contacting membrane and promotes convective fluid flow in proximity to the device (Figure 1e,f). Furthermore, we have added data exploring the relationship between membrane deflection and tissue strain (Supplementary Figure 3).

Results (page 8, line 1 – 7)

As part of device design and optimisation, we performed finite element (FE) simulations to understand the biomechanical changes mediated by actuation, particularly the relationships between membrane deflection, convective flow, and tissue strain (Figure 1e,f; Supplementary Figure 3). Based on recently reported results³, we designed our soft robotic actuation strategy to induce tissue strain that would fall within the atraumatic range (<40%), and hypothesise that this regimen would mitigate the FBR by creating convective flow disruptive to the cellular immune response.

Fig. 1 | e, FE model showing peri-implant fluid velocity of convective flow during actuation. **f**, FE model estimating maximum principal tissue strain induced by actuation.

Supplementary Fig. 3 | Membrane deflection in STAR. **a**, Schematics of bi-chambered STAR device. Single pressure input allows for variable deflection in a single device. **b**, Prototype of bi-chambered STAR device manufactured by the process described in Supplementary Figure 2. **c**, Schematic depicting lateral view of bi-chambered device, and differential deflection height of each chamber upon uniform pressurization. **d**, Maximum membrane deflection for the 3 mm and 6 mm chambers during actuation with a 1–9 psi pressure input. Pressure of 2 psi was chosen for the preclinical experiments. **e**, Finite element (FE) model showing tissue deflection caused by STAR membrane at 1, 2, and 3 psi pressure input. **f**, FE model showing tissue strain caused by STAR membrane at 1, 2, and 3 psi pressure input. **g**, FE model showing peri-implant fluid velocity of convective flow around the large and small chamber during actuation at 2 psi.

3. We have further expanded the introduction to better frame our mechanotherapeutic strategy:

Introduction (page 5, line 5 – 17)

Dynamically altering the local biomechanical environment at the implant site is one such promising, yet underexplored, drug-free approach¹⁴. Cells in our body are exquisitely sensitive to their mechanical environment, with loading playing a pivotal role in cell functions such as differentiation¹⁵, proliferation¹⁶, and migration¹⁷. Historically, studies have observed biomechanical stress as a pro-fibrotic or regenerative stimulus¹⁸, demonstrating that application of stretch⁷⁻⁹, fluid flow^{10,11}, or compression¹² to cells can lead to increased deposition of collagenous matrix. Accordingly, many anti-FBR strategies have focused on minimizing the mechanical mismatch, interfacial stress, and movement between the implant and local tissue. Our research seeks to challenge this status quo and reveals the potential for a dynamic mechanotherapeutic that uses low-magnitude, atraumatic, tissue strain and convective flow as a defence mechanism against the invading cellular FBR. Interestingly, some studies have observed that small magnitude, dynamic loading has anti-inflammatory effects, and indicated the presence of a therapeutic threshold, beyond which tissue damage and inflammation occurs^{3,13}.

4. We have added a schematic highlighting the proposed mechanism of action to Figure 1a, with accompanying text:

Results (page 7, lines 2 – 7)

Our lab previously demonstrated the fibrotic attenuating potential of a dynamic device during the initial stages of the FBR (2 weeks). Based on this foundational work, we propose that application of intermittent, cyclical, low amplitude actuation can act as an oscillating shield against the invading, multiphasic FBR, induce local immunomodulatory effects, and create a favourable environment for the rapid long-term transport of macromolecular drug therapy (Figure 1a).

Fig. 1 | Design of soft transport augmenting reservoir (STAR). a, Proposed mechanism of STAR: intermittent actuation shields against the foreign body response, creating a favourable environment for the rapid long-term transport of macromolecular drug therapy.

5. Finally, in the discussion, we comment on STAR's mechanically disruptive oscillation at a cellular level and discuss potential future avenues of explorative research.

Discussion (page 25, lines 14 – 20)

A recent study by Seo et al.³ corroborates the mechanosensitive nature of neutrophil cell populations following dynamic loading for skeletal muscle regeneration. The authors postulated that mechanical flushing of chemoattractants was responsible for the reported decrease in neutrophils³. In this context, our work motivates further study on the effect of IA on pro-

inflammatory chemoattractant gradients such as interleukin 1, 6 and 8 versus direct mechanical effects on cell attachment, orientation, and function.

Typos to revise:

R3C15:

“This potential space it is well”

“Devices was pneumatically”

“baseline. 300 µl of the”: shouldn’t start a sentence with a numeral.

“heated pad. 3 mL of warm”: shouldn’t start a sentence with a numeral.

“Data is represented”

Response: We thank the reviewer for pointing out these grammatical errors, and we have amended them in the text.

References

1. Li, J. & Mooney, D. J. Designing hydrogels for controlled drug delivery. *Nat. Rev. Mater.* **1**, 16071 (2016).
2. Shirazi, R. N. *et al.* Multiscale Computational Modeling: Multiscale Experimental and Computational Modeling Approaches to Characterize Therapy Delivery to the Heart from an Implantable Epicardial Biomaterial Reservoir (Adv. Healthcare Mater. 16/2019). *Adv. Healthc. Mater.* **8**, 2019 (2019).
3. Seo, B. R. *et al.* Skeletal muscle regeneration with robotic actuation–mediated clearance of neutrophils. *Sci. Transl. Med.* **13**, (2021).
4. Carver, W. & Goldsmith, E. C. Regulation of tissue fibrosis by the biomechanical environment. *Biomed Res. Int.* **2013**, (2013).
5. PC, G. *et al.* Increased stiffness of the rat liver precedes matrix deposition: implications for fibrosis. *Am. J. Physiol. Gastrointest. Liver Physiol.* **293**, (2007).
6. Jabaley, M. E. & Das, S. K. Late Breast Pain Following Reconstruction with Polyurethane-Covered Implants. *Plast. Reconstr. Surg.* **78**, 390–395 (1986).
7. Jalil, J. E. *et al.* Fibrillar collagen and myocardial stiffness in the intact hypertrophied rat left ventricle. *Circ. Res.* **64**, 1041–1050 (1989).
8. Imsirovic, J. *et al.* A novel device to stretch multiple tissue samples with variable patterns: application for mRNA regulation in tissue-engineered constructs. *Biomatter* **3**, (2013).
9. Leung, D. Y. M., Glagov, S. & Mathews, M. B. Cyclic stretching stimulates synthesis of matrix components by arterial smooth muscle cells in vitro. *Science (80-)*. **191**, 475–477 (1976).
10. Tan, H. *et al.* Fluid flow forces and rhoA regulate fibrous development of the atrioventricular valves. *Dev. Biol.* **374**, 345–356 (2013).
11. RodrÃ­guez, I. & GonzÃ¡lez, M. Physiological mechanisms of vascular response induced by shear stress and effect of exercise in systemic and placental circulation. *Front. Pharmacol.* **5**, (2014).
12. Regulation of Tissue Fibrosis by the Biomechanical Environment.
13. Yang, G., Im, H. J. & Wang, J. H. C. Repetitive mechanical stretching modulates IL-1 β induced COX-2, MMP-1 expression, and PGE2 production in human patellar tendon fibroblasts. *Gene* **363**, 166 (2005).
14. Dolan, E. B. *et al.* An actuatable soft reservoir modulates host foreign body response. *Sci. Robot.* **4**, eaax7043 (2019).

15. Engler, A. J., Rehfeldt, F., Sen, S. & Discher, D. E. Microtissue Elasticity: Measurements by Atomic Force Microscopy and Its Influence on Cell Differentiation. *Methods in Cell Biology* **83**, 521–545 (2007).
16. Wang, H. B., Dembo, M. & Wang, Y. L. Substrate flexibility regulates growth and apoptosis of normal but not transformed cells. *Am. J. Physiol. - Cell Physiol.* **279**, (2000).
17. Peyton, S. R. & Putnam, A. J. Extracellular matrix rigidity governs smooth muscle cell motility in a biphasic fashion. *J. Cell. Physiol.* **204**, 198–209 (2005).
18. Corradetti, B. *The immune response to implanted materials and devices: The impact of the immune system on the success of an implant. The Immune Response to Implanted Materials and Devices: The Impact of the Immune System on the Success of an Implant* (Springer International Publishing, 2016). doi:10.1007/978-3-319-45433-7
19. Cezar, C. A. *et al.* Biologic-free mechanically induced muscle regeneration. *Proc. Natl. Acad. Sci. U. S. A.* **113**, 1534–1539 (2016).
20. Freckmann, G. *et al.* Randomized Cross-Over Study Comparing Two Infusion Sets for CSII in Daily Life. *J. Diabetes Sci. Technol.* **11**, 253 (2017).
21. Kharroubi, A. T. Diabetes mellitus: The epidemic of the century. *World J. Diabetes* **6**, 850 (2015).
22. Hwang, P. T. J. *et al.* Progress and challenges of the bioartificial pancreas. *Nano Convergence* **3**, 28 (2016).
23. Vaithilingam, V., Bal, S. & Tuch, B. E. Encapsulated islet transplantation: Where do we stand? *Review of Diabetic Studies* **14**, 51–78 (2017).
24. Chick, W. L. *et al.* Artificial pancreas using living beta cells: Effects on glucose homeostasis in diabetic rats. *Science (80-.)*. **197**, 780–782 (1977).
25. Goswami, D. *et al.* Design Considerations for Macroencapsulation Devices for Stem Cell Derived Islets for the Treatment of Type 1 Diabetes. *Adv. Sci.* 2100820 (2021). doi:10.1002/ADVS.202100820
26. Alsaleh, F. M., Smith, F. J., Keady, S. & Taylor, K. M. G. Insulin pumps: from inception to the present and toward the future. *J. Clin. Pharm. Ther.* **35**, 127–138 (2010).
27. Farra, R. *et al.* First-in-human testing of a wirelessly controlled drug delivery microchip. *Sci. Transl. Med.* **4**, 122ra21 (2012).
28. Vegas, A. J. *et al.* Long-term glycemic control using polymer-encapsulated human stem cell-derived beta cells in immune-competent mice. *Nat. Med.* **22**, 306–311 (2016).
29. Veisheh, O. & Vegas, A. J. Domesticating the foreign body response: Recent advances and applications. *Advanced Drug Delivery Reviews* **144**, 148–161 (2019).

30. Wisniewski, N. *et al.* Analyte flux through chronically implanted subcutaneous polyamide membranes differs in humans and rats. *Am. J. Physiol. Metab.* **282**, E1316–E1323 (2002).
31. Helton, K. L., Ratner, B. D. & Wisniewski, N. A. Biomechanics of the sensor-tissue interface - Effects of motion, pressure, and design on sensor performance and the foreign body response - Part I: Theoretical framework. *J. Diabetes Sci. Technol.* **5**, 632–646 (2011).
32. Chung, L. *et al.* Interleukin 17 and senescent cells regulate the foreign body response to synthetic material implants in mice and humans. *Sci. Transl. Med.* **12**, (2020).
33. Study: Disease-Modifying Therapies Needed to Offset Costs of Type 1 Diabetes - JDRF.
34. Kharbikar, B. N., Chendke, G. S. & Desai, T. A. Modulating the foreign body response of implants for diabetes treatment. *Adv. Drug Deliv. Rev.* **174**, 87–113 (2021).
35. Edman, C. & Drinan, D. A review of the management of implanted medical devices for diabetes: Trends and directions. in *Journal of Diabetes Science and Technology* **2**, 995–1002 (SAGE Publications Inc., 2008).
36. Photiadis, S. J., Gologorsky, R. C. & Sarode, D. The Current Status of Bioartificial Pancreas Devices. *ASAIO J.* 370–381 (2021). doi:10.1097/MAT.0000000000001252
37. Ward, W. K. A review of the foreign-body response to subcutaneously-implanted devices: The role of Macrophages and cytokines in biofouling and fibrosis. in *Journal of Diabetes Science and Technology* **2**, 768–777 (SAGE Publications Inc., 2008).
38. Bose, S. *et al.* A retrievable implant for the long-term encapsulation and survival of therapeutic xenogeneic cells. *Nat. Biomed. Eng.* **4**, 814–826 (2020).
39. Zhang, D. *et al.* Dealing with the Foreign-Body Response to Implanted Biomaterials: Strategies and Applications of New Materials. *Advanced Functional Materials* **31**, (2021).
40. Liu, Q. *et al.* Zwitterionically modified alginates mitigate cellular overgrowth for cell encapsulation. *Nat. Commun.* 2019 101 **10**, 1–14 (2019).
41. Zhang, D. *et al.* Bio-inspired poly-DL-serine materials resist the foreign-body response. *Nat. Commun.* 2021 121 **12**, 1–12 (2021).
42. Zhang, D. *et al.* Silk-Inspired β -Peptide Materials Resist Fouling and the Foreign-Body Response. *Angew. Chemie* **132**, 9673–9680 (2020).
43. Zhang, J. *et al.* Rapid and Long-Term Glycemic Regulation with a Balanced Charged Immune-Evasive Hydrogel in T1DM Mice. *Adv. Funct. Mater.* **29**, 1900140 (2019).
44. Zhang, L. *et al.* Zwitterionic hydrogels implanted in mice resist the foreign-body reaction. *Nat. Biotechnol.* **31**, 553–556 (2013).

45. Vegas, A. J. *et al.* Combinatorial hydrogel library enables identification of materials that mitigate the foreign body response in primates. *Nat. Biotechnol.* **34**, 345–352 (2016).
46. Bochenek, M. A. *et al.* Alginate encapsulation as long-term immune protection of allogeneic pancreatic islet cells transplanted into the omental bursa of macaques. *Nat. Biomed. Eng.* **2**, 810–821 (2018).
47. Veisoh, O. *et al.* Size- and shape-dependent foreign body immune response to materials implanted in rodents and non-human primates. *Nat. Mater.* **14**, 643–651 (2015).
48. Avula, M. N. & Grainger, D. W. Addressing Medical Device Challenges with Drug–Device Combinations. *Drug-Device Comb. Chronic Dis.* 1–38 (2015). doi:10.1002/9781119002956.CH01
49. Coulter, F. B. *et al.* Additive Manufacturing of Multi-Scale Porous Soft Tissue Implants That Encourage Vascularization and Tissue Ingrowth. *Adv. Healthc. Mater.* 2100229 (2021). doi:10.1002/adhm.202100229
50. Dockery, P. & Fraher, J. The quantification of vascular beds: A stereological approach. *Exp. Mol. Pathol.* **82**, 110–120 (2007).
51. Madigan, N. N. *et al.* Comparison of cellular architecture, axonal growth, and blood vessel formation through cell-loaded polymer scaffolds in the transected rat spinal cord. *Tissue Eng. Part A* **20**, 2985–2997
52. Ward, W. K. & Duman, H. M. In Vivo Glucose Sensing. in *In Vivo Glucose Sensing* (eds. Cunningham, D. D. & Stenken, J. A.) 59–85 (Wiley, 2010).
53. Sharkawy, A. A., Klitzman, B., Truskey, G. A. & Reichert, W. M. Engineering the tissue which encapsulates subcutaneous implants. I. Diffusion properties. *J. Biomed. Mater. Res.* **37**, 401–412 (1997).
54. Ratner, B. D. Reducing capsular thickness and enhancing angiogenesis around implant drug release systems. in *Journal of Controlled Release* **78**, 211–218 (2002).
55. Blanco, E. *et al.* Effect of fibrous capsule formation on doxorubicin distribution in radiofrequency ablated rat livers. *J. Biomed. Mater. Res. - Part A* **69**, 398–406 (2004).
56. Farah, S. *et al.* Long-term implant fibrosis prevention in rodents and non-human primates using crystallized drug formulations. *Nat. Mater.* **18**, 892–904 (2019).
57. Barone, D. G. *et al.* Prevention of the foreign body response to implantable medical devices by inflammasome inhibition. *Proc. Natl. Acad. Sci.* **119**, (2022).
58. Helton, K. L., Ratner, B. D., Wisniewski, N. A. & Wisniewski, N. Biomechanics of the Sensor-Tissue Interface-Effects of Motion, Pressure, and Design on Sensor Performance and Foreign Body Response-Part II: Examples and Application. *J. Diabetes Sci. Technol.* **5**, 647–656 (2011).

59. Guimarães, C. F., Gasperini, L., Marques, A. P. & Reis, R. L. The stiffness of living tissues and its implications for tissue engineering. *Nat. Rev. Mater.* 2020 55 **5**, 351–370 (2020).
60. The effects of substrate stiffness on the in vitro activation of macrophages and in vivo host response to poly(ethylene glycol)-based hydrogels - PubMed.
61. Dang, T. T. *et al.* Spatiotemporal effects of a controlled-release anti-inflammatory drug on the cellular dynamics of host response. *Biomaterials* **32**, 4464–4470 (2011).

REVIEWER COMMENTS

Reviewer #1 (Remarks to the Author):

I am pleased with the additional data that the authors have included in this revision. This significantly enhances the manuscript and addresses various points raised in my first review. I do not think the manuscript requires any further data or major revisions to be accepted into Nature Communications. However, I would request the authors address the following minor revisions and comments:

- Comparison with previous study parameters: Various times in the introduction the authors refer to prior works that either induce FBR or 'smaller magnitude' actuation (Page 3, line 16) that is anti-inflammatory. Either in the introduction or discussion, I would recommend the authors include a brief quantitative comparison of these parameters for technical readers (including values for magnitude and frequency) who may wish to implement these findings.
- Page 7, line 3: add reference.
- The contrast raised between diffusion and convection delivery is a crucial part of the manuscript. The authors may want to include a calculation and discussion of Peclet number values to quantitatively characterize this.
- I would suggest the authors add a few lines early on explaining how IA and RR differ mechanistically. It took this reviewer a few reads to fully grasp that in IA experiments, devices did not have drug but were only actuated, and that in RR actuation was simultaneous with drug infusion into the device. One other interpretation that readers may have is that the drug is infused into the device at high speed to cause large convective flow when released through the membrane.
- I remain with some doubts about the extent of RR impact as shown by Figure 5i. Looking at the drop in BG of the blue (RR) group following RR, this seems similar in slope BG drop following the first diffusion delivery slope (~20% reduction in 15min). While the other panels in figure 5 do distinguish RR from diffusion transport, it is worth calling out in the discussion. If my interpretation is incorrect, then this difference should be characterized and explicitly highlighted. If not, then some discussion as to why the BG drops are so similar following different release strategies should be mentioned.

Reviewer #2 (Remarks to the Author):

The authors have addressed all questions and concerns with new experiments and discussions. I am satisfied by the revised version.

Reviewer #3 (Remarks to the Author):

The authors have done a comprehensive and thorough job in responding to the numerous prior critiques. The manuscript has been revised to accommodate prior recommendations and with new data and extensive revisions that strengthen the manuscript as requested.

As a result the revised manuscript is highly improved and sound. I have no further concerns.

We thank the reviewers once again for re-assessing our manuscript and for their positive feedback. In the current revision, we have addressed the remaining minor comments from Reviewer #1.

To facilitate reassessment of the manuscript, we have provided a point-by-point response to all the reviewers' comments in the following pages. Our responses are in green text, and the resultant changes in the manuscript noted in blue text.

Reviewer #1:

Summary Comment: I am pleased with the additional data that the authors have included in this revision. This significantly enhances the manuscript and addresses various points raised in my first review. I do not think the manuscript requires any further data or major revisions to be accepted into Nature Communications. However, I would request the authors address the following minor revisions and comments.

Response: We thank the reviewer for their insightful comments on the initial version of our manuscript which had motivated us to conduct several additional experiments to more comprehensively study the effect of our actuation strategies on drug delivery and the underlying mechanism of action. We are glad that the reviewer found the revised manuscript to be significantly improved. Below, we have addressed their remaining minor comments.

RIC1: Comparison with previous study parameters: Various times in the introduction the authors refer to prior works that either induce FBR or 'smaller magnitude' actuation (Page 3, line 16) that is anti-inflammatory. Either in the introduction or discussion, I would recommend the authors include a brief quantitative comparison of these parameters for technical readers (including values for magnitude and frequency) who may wish to implement these findings.

Response:

We thank the reviewer for this suggestion. There have been a number of prior studies that examine the effect of mechanical loading on inflammation and tissue regeneration, however there is significant heterogeneity with regards to actuation methods, regimens, resulting deformations, target tissue, and animal models used. Lee et al. [*Nature* 2000, 408, 998-1000; *Adv. Mater.* 2001, 13 (11), 837-839] previously demonstrated that externally applied mechanical compression of implanted VEGF-laden hydrogels at 25-50% strain for 3 cycles of 2 minutes for 7 days led to increased vascularity and thicker granulation tissue. Cezar et al. [*PNAS* 2016, 113(6), 1534-1539] have shown that magnetic actuation of ferrogel scaffolds at 1 Hz for 5 minutes every 12 hours for 2 weeks (force of 2N/g) reduced fibrous capsule thickness and inflammatory infiltrate at the site of muscle injury in mice. Seo et al. [*Sci. Transl. Med.* 2021, 13(614), eabe8868] recently demonstrated that a similar regimen of robotic actuation at 1 Hz for 5 minutes every 12 hours for 2 weeks with strains ranging from 10-40% has an immunomodulatory effect, with improved clearance of neutrophils resulting in better functional recovery. This study also demonstrated a 40% strain threshold beyond which tissue damage occurs. We acknowledge that this threshold may differ with the type of mechanical stimulus applied, and the type of tissue, as Yang et al. [*Gene* 2005, 363, 166-172] demonstrated *in vitro* that cyclic uniaxial stretching of human patellar tendon fibroblasts at lower strains (4% at 0.5Hz for 4 hours) produced an anti-inflammatory gene expression profile, whereas a higher strain (8%) produced the opposite pro-inflammatory profile.

We have included a brief summary of these relevant quantitative parameters in our introduction and discussion, reproduced below:

Introduction

page 5, lines 15-23:

Interestingly, some studies have observed that small magnitude, dynamic loading has anti-inflammatory and pro-regenerative effects. Previous work applying dynamic loading to tissue has used daily mechanical, pneumatic, or magnetic stimuli, either internally or externally, to apply cyclic loads, inducing strains ranging from 4-50%. with each cycle lasting between 1 second and 10 minutes.⁴⁵⁻⁵¹ These studies have demonstrated beneficial effects in terms of vascularisation^{45,46,51}, functional tissue regeneration^{47,50}, and anti-inflammatory gene expression⁴⁸. These prior works on mechanical loading have indicated the presence of a therapeutic threshold, beyond which tissue damage and inflammation occurs^{48,50}.

Discussion

page 29, lines 11-17:

Although there have been a number of prior studies that examine the effect of mechanical loading on inflammation and tissue regeneration, there is significant heterogeneity with regards to actuation methods, regimens, resulting deformations, target tissues, and animal models used⁴⁵⁻⁵¹. Only a few studies have attempted to address the effect of varying tissue strain^{48,50,51} and loading frequency⁵¹; therefore, significant work is needed to define the optimal loading parameters that maximise the anti-inflammatory effects of mechanical actuation, which may differ with the type of tissue and mechanical stimulus.

RIC2: Page 7, line 3: add reference.

Response: We have added the relevant reference to the text.

RIC3: The contrast raised between diffusion and convection delivery is a crucial part of the manuscript. The authors may want to include a calculation and discussion of Peclet number values to quantitatively characterize this.

Response: We thank the reviewer for this pertinent suggestion. We have now included a discussion on representative Peclet number values to quantitatively compare drug delivery by the two modalities – passive diffusion only and actuation-mediated rapid release.

The Peclet number for mass transfer, for a characteristic length L , is defined as $Pe_L = uL/D$, where u is the local flow velocity and D is the diffusivity. For our representative calculations, we assume $L = 1$ mm. This assumption is based on our experimental and computational data showing that the maximum device membrane deflection is ~ 1.5 mm and the estimated tissue deflection is ~ 0.73 mm (Supplementary Fig. 3). We use a diffusivity $D = 855 \mu\text{m}^2/\text{s}$ adopted from [*Buchwald, P. A local glucose-and oxygen concentration-based insulin secretion model for pancreatic islets. Theor. Biol. Med. Model. 8, 20 (2011)*], which is the same value used in our COMSOL Multiphysics models. For the passive diffusion scenario, a reasonable estimate for u is the velocity of interstitial fluid, which has been reported widely to be in the range $0.1 - 2 \mu\text{m}/\text{s}$ [<https://doi.org/10.1016/B978-0-08-087780-8.00040-1>]. For the purposes of this comparison, the most conservative estimate is $2 \mu\text{m}/\text{s}$. Our COMSOL simulations reveal a flow velocity of 0.06 mm/s adjacent to the porous membrane immediately following actuation. Using these estimates, we

get $Pe = 2.35$ for passive diffusion and $Pe = 70.18$ for actuation-mediated RR. This establishes, quantitatively, that for a given dose of drug, the time taken for purely passive drug delivery (diffusion dominated) far exceeds that for actuated-mediated drug delivery (convection dominated).

The changes made to the manuscript to address this comment are summarized below:

Results:

page 19, lines 22-26:

Péclet number (Pe) calculations⁵⁷ estimate $Pe = 2.35$ for passive diffusion and $Pe = 70.18$ for actuation-mediated RR, suggesting that for a given dose of drug, the time required for passive drug delivery through a diffusion dominated process far exceeds that of actuation-mediated drug delivery, which is convection dominated.

Methods:

page 34, line 18 - page 35, line 4:

Péclet number calculations. The Péclet number for mass transfer, for a characteristic length L , is defined as $Pe_L = uL/D$, where u is the local flow velocity and D is the diffusivity⁵⁷. For representative calculations, $L = 1$ mm was assumed. This assumption is based on experimental and computational data showing that the maximum device membrane deflection is ~ 1.5 mm and the estimated tissue deflection is ~ 0.73 mm (Supplementary Fig. 3). A diffusivity of $D = 855 \mu\text{m}^2/\text{s}$, which is the same value used in the COMSOL Multiphysics models above, was used⁷⁴. For the passive diffusion scenario, a reasonable estimate for u is the velocity of interstitial fluid, which has been reported widely to be in the range 0.1 - $2 \mu\text{m}/\text{s}$ ⁷⁵. A flow velocity of 0.06 mm/s adjacent to the porous membrane immediately following actuation was calculated from the COMSOL simulations above.

RIC4: I would suggest the authors add a few lines early on explaining how IA and RR differ mechanistically. It took this reviewer a few reads to fully grasp that in IA experiments, devices did not have drug but were only actuated, and that in RR actuation was simultaneous with drug infusion into the device. One other interpretation that readers may have is that the drug is infused into the device at high speed to cause large convective flow when released through the membrane.

Response: We thank the reviewer for pointing out this potential source of confusion, especially regarding another possible interpretation of actuation-mediated rapid release. As the reviewer has correctly pointed out, intermittent actuation (IA) occurs without any drug present in the devices. With actuation-mediated rapid release (RR), however, the drug is injected into the device and then actuation is performed. In fact, actuation does not necessarily need to happen simultaneously or immediately after drug infusion. The drug can be infused first and then actuation performed at any later point to expel any drug remaining inside the device after a period of diffusion, allowing for on-demand release. This is the basis for our experiments shown in Figure 5i.

Although the two strategies, IA and RR, utilize the same actuation parameters of cyclic pressure input of 2 psi at 1 Hz, they have fundamentally different goals: IA is a mechanotherapeutic method to mitigate the foreign body response, whereas RR is a means to perform on-demand drug delivery.

We have added additional text in the Results and Methods to better distinguish the difference between IA and RR. The changes are summarized below:

Results

page 12, lines 18-27:

We implanted STAR devices (without drug) on the dorsal subcutaneous aspect of 3 groups of mice (Figure 3a). In two experimental groups, we performed STAR-enabled IA with cyclic pressure input of 2 psi at 1 Hz for 5 min every 12 hours using a custom-made pneumatic control system (Supplementary Figure 6). One group (8W IA) was intermittently actuated for the total study duration of 8 weeks, while the second group (3W IA) received 3 weeks of IA followed by no actuation for the remainder of the study. A third group which did not receive IA served as the control. We then injected short-acting human insulin (2 IU/kg) into the device at various time points post-implantation: 2, 3, 4, 5, and 8 weeks as well as day 3, which served as a baseline (BL). We monitored passive, diffusion-based transport across the formed FC and into the bloodstream via serial blood glucose measurements at these timepoints.

page 20, lines 6-14:

In a final example, we demonstrated enhanced mass transport and downstream functional effect using RR in our ITT model at 2 weeks after STAR implantation (Figure 5i). Passive diffusion of insulin led to a drop in blood glucose over 120 min in all animals. At this point, food was given to one group to allow recovery of blood glucose levels towards baseline. At 150 min, this group was subjected to 5 cycles of actuation (with the same parameters of 2 psi at 1 Hz). No additional insulin was administered after the initial dose given at the start of the ITT. Despite a reduced insulin concentration gradient across the device and attenuated insulin sensitivity in the post-prandial animals, actuation-mediated RR led to a significant reduction in blood glucose levels over 15 min due to augmented release of insulin from STAR (Figure 5i).

Methods

page 36, lines 19-23:

Intermittent actuation: Intermittent actuation was performed by connecting a custom-made electropneumatic actuation and control system (described above) to the self-sealing transcutaneous actuation port using a PNP3M connector (Instech) (Supplementary Figure 4d). The device was then cyclically actuated at a controlled input pressure of 2 psi at 1 Hz for 5 min every 12 hours as previously described^{35,48}. No drug was present in the device during IA throughout the study.

page 37, lines 8-12:

Actuation-mediated rapid release of insulin: At 2 weeks following implantation, serial blood glucose measurements were performed over 120 min after insulin injection as described above. Food was given to the RR group at 120 min to allow recovery of blood glucose levels. Actuation-mediated RR was then performed by actuating the STAR device at 150 min. Note that no additional insulin was administered after the initial dose given at time = 0 min.

RIC5: I remain with some doubts about the extent of RR impact as shown by Figure 5i. Looking at the drop in BG of the blue (RR) group following RR, this seems similar in slope BG drop following the first diffusion delivery slope (~20% reduction in 15min). While the other panels in figure 5 do distinguish RR from diffusion transport, it is worth calling out in the discussion. If my interpretation is incorrect, then this

difference should be characterized and explicitly highlighted. If not, then some discussion as to why the BG drops are so similar following different release strategies should be mentioned.

Response: We thank the reviewer for this insightful comment. We agree with the reviewer that the slope of BG drop in the RR (blue) group in Figure 5i during the first 15 minutes of the experiment is similar to that immediately following RR. The reviewer has also correctly pointed out that, for example in Figure 5h, the slope representing drug delivered by RR is substantially higher than drug delivered by passive diffusion only. To remove this potential source of confusion, we would like to clarify the differences between the two experimental conditions (e.g., Figure 5h and 5i) and explain why the slopes in Figure 5i are similar.

In the experiment reported in Figure 5i, a single device in the animal is filled with insulin at $t = 0$ and the drug is then allowed to diffuse passively for 150 min. At $t = 150$ min, in the RR (blue) group, the device is actuated for 5 cycles (i.e., at 1 Hz for 5 seconds) – no additional insulin was injected at $t = 150$ min. Direct comparison between the slopes of the BG curve at $t = 0^+$ and $t = 150^+$ min is therefore not warranted for the following reasons: (1) The concentration of the insulin delivered at $t = 150$ min (by actuation) is lower than at $t = 0$ due to diffusion occurring for these 150 min (and thus lowering the concentration gradient across the device membrane). We highlight again that no additional insulin was injected at $t = 150$ min. (2) The two instances, $t = 0$ and $t = 150$ min, represent different physiologic states of the animal. Prior to all insulin transport tests, including the one in Figure 5i, the animals were fasted for 4 hours. Sensitivity to insulin strongly depends on how long the animal has been fasted prior to administering insulin. Thus, it is expected that animals would respond faster to insulin at $t = 0$ (after 4 hours of fasting) than at $t = 150$ (just 30 mins after feeding) at similar starting BG levels. In light of this, we believe that the result in Figure 5i is all the more impressive, as the slopes at $t = 0^+$ and $t = 150^+$ are similar even though actuation-mediated RR is applied in the context of decreased postprandial insulin sensitivity and a depleted insulin concentration inside the device.

In contrast, in the experiment in Figure 5h, each animal was implanted with 2 devices, which were simultaneously injected with the drug analogue. Only one of these devices were subjected to actuation (RR) at $t = 15$ min. Thus, we can directly compare the slopes of the curves at $t = 15^+$ min, finding that the slope in the RR (blue) group is substantially higher.

We have detailed this in the revised manuscript, and have also updated the methods to clarify the experimental procedure. The changes are summarized below:

Results

page 20, lines 6-14:

In a final example, we demonstrated enhanced mass transport and downstream functional effect using RR in our ITT model at 2 weeks after STAR implantation (Figure 5i). Passive diffusion of insulin led to a drop in blood glucose over 120 min in all animals. At this point, food was given to one group to allow recovery of blood glucose levels towards baseline. At 150 min, this group was subjected to 5 cycles of actuation (with the same parameters of 2 psi at 1 Hz). No additional insulin was administered after the initial dose given at the start of the ITT. Despite a reduced insulin concentration gradient across the device and attenuated insulin sensitivity in the post-prandial animals, actuation-mediated RR led to a significant reduction in blood glucose levels over 15 min due to augmented release of insulin from STAR (Figure 5i).

Methods

page 36, lines 6-7:

Insulin transport test (ITT): [...] Animals were fasted for 4 hours prior to the start of the ITT, and were kept in a clean cage without food and bedding for the duration of the test.

page 37, lines 8-12:

Actuation-mediated rapid release of insulin: At 2 weeks following implantation, serial blood glucose measurements were performed over 120 min after insulin injection as described above. Food was given to the RR group at 120 min to allow recovery of blood glucose levels. Actuation-mediated RR was then performed by actuating the STAR device at 150 min. Note that no additional insulin was administered after the initial dose given at time = 0 min.

Reviewer #2:

The authors have addressed all questions and concerns with new experiments and discussions. I am satisfied by the revised version.

We thank the reviewer for their important comments on our initial submission that motivated our additional experiments which have strengthened this work.

Reviewer #3:

The authors have done a comprehensive and thorough job in responding to the numerous prior critiques. The manuscript has been revised to accommodate prior recommendations and with new data and extensive revisions that strengthen the manuscript as requested.

As a result the revised manuscript is highly improved and sound. I have no further concerns.

We thank the reviewer for their detailed review and insightful comments on the initial submission that led to a significant improvement of this work.